# TRPV1 inhibition overcomes cisplatin resistance by blocking autophagy-mediated hyperactivation of EGFR signaling pathway

Se Jin Oh [1,2], Ji Yeon Lim [1,3], Min Kyu Son[1,2], Jun Hyeok Ahn[1,2], Kwon-Ho Song[4], Hyo-Jung Lee[1,2], Suyeon Kim[1,2], Eun Ho Cho [1,2], Joon-Yong Chung [5], Hanbyoul Cho[6], Hyosun Kim[6], Jae-Hoon Kim[6], Jooyoung Park [7], Jungmin Choi [7], Sun Wook Hwang [1,3] & Tae Woo Kim[1,2,8] ✉

Cisplatin resistance along with chemotherapy-induced neuropathic pain is an important cause of treatment failure for many cancer types and represents an unmet clinical need. Therefore, future studies should provide evidence regarding the mechanisms of potential targets that can overcome the resistance as well as alleviate pain. Here, we show that the emergence of cisplatin resistance is highly associated with EGFR hyperactivation, and that EGFR hyperactivation is arisen by a transcriptional increase in the pain-generating channel, TRPV1, via NANOG. Furthermore, TRPV1 promotes autophagy-mediated EGF secretion via $Ca^{2+}$ influx, which activates the EGFR-AKT signaling and, consequentially, the acquisition of cisplatin resistance. Importantly, TRPV1 inhibition renders tumors susceptible to cisplatin. Thus, our findings indicate a link among cisplatin resistance, EGFR hyperactivation, and TRPV1-mediated autophagic secretion, and implicate that TRPV1 could be a crucial drug target that could not only overcome cisplatin resistance but also alleviate pain in NANOG+ cisplatin-resistant cancer.

Cisplatin is one of the most intensive chemotherapeutic agents widely used for treating tumors, such as cervical, lung, head and neck, gastric and bladder cancer[1,2]. However, repeated cisplatin treatment frequently results in the acquisition of resistance and severe side effects, such as chemotherapy-induced neuropathic pain (CIPN)[1,3–8]. Cisplatin resistance along with the CIPN is an important contributor to cancer treatment failure and represents an unmet clinical need[8–10]. Therefore, to fulfill this unmet clinical need, future studies should provide definitive evidence regarding the mechanisms of novel drug targets that can not only overcome

the cisplatin resistance but also lessen the pain caused by cisplatin.

The cisplatin resistance in cervical cancer are complex and associated with the following features: (1) reduction in the intracellular accumulation of the platinum compounds; (2) increase in DNA damage repair; (3) inactivation of apoptosis; (4) activation of EMT; (5) alteration in DNA methylation, microRNA profile, cancer stem cell characteristics, and expression of stress-response chaperones[11]. Among the diverse molecular mechanisms, the epidermal growth factor receptor (EGFR), an oncogenic receptor tyrosine kinase, hyperactivation has

[1]BK21 Graduate Program, Department of Biomedical Sciences, Korea University College of Medicine, Seoul 02841, Republic of Korea. [2]Department of Biochemistry and Molecular Biology, Korea University College of Medicine, Seoul 02841, Republic of Korea. [3]Department of Physiology, Korea University College of Medicine, Seoul 02841, Republic of Korea. [4]Department of Cell biology, Daegu Catholic University School of Medicine, Daegu 42472, Republic of Korea. [5]Molecular Imaging Branch, Center for Cancer Research, National Cancer Institute, National Institutes of Health, Bethesda, MD 20892, USA. [6]Department of Obstetrics and Gynecology, Gangnam Severance Hospital, Yonsei University College of Medicine, Seoul 06273, Republic of Korea. [7]Department of Biomedical Sciences, Korea University College of Medicine, Seoul 02841, Republic of Korea. [8]NEX-I Inc., Seoul 05854, Republic of Korea. ✉ e-mail: twkim0421@korea.ac.kr

attracted attention, as it can trigger cisplatin resistance features above. In tumor cells, the EGFR hyperactivation increases drug efflux, DNA damage repair, anti-apoptosis, and stem-like properties[12–15]. Notably, hyperactivated EGFR signaling was found in various cancer types that acquire cisplatin resistance[16–18]. In this regard, results from recent studies have suggested that the hyperactivation of EGFR signaling could be one of major pathway that drives the adaptation of tumor cells to cisplatin, thereby contributing to the generation of cancer cells with better survival advantages. Therefore, it is important to identify the cause of EGFR hyperactivation to overcome the clinical limitations of cisplatin treatment in cervical cancer patient with hyperactivated EGFR signaling.

In an effort to elucidate the molecular mechanisms underlying dysregulation of EGFR signaling, we previously found that immunotherapy-mediated immune selection caused the enrichment of a subset of cells with high expression of NANOG[19,20]. NANOG+ cells exhibited stem-like, metastatic, anti-apoptotic properties, and resistance to multi-modal therapies, such as chemo-, radio-, and immunotherapy, by hyperactivation of the EGFR-AKT pathway, indicating the important role of NANOG signaling in these refractory phenotypes of tumor cells[21–27]. Notably, NANOG induces EGF secretion by promoting secretory autophagy rather than degradative autophagy, thereby hyperactivating the EGFR-AKT signaling pathway[14]. Thus, these results propose a rationale whereby strategies impeding NANOG-mediated secretory autophagy may overcome the limitations of cisplatin-based chemotherapy. Given that clinically available NANOG or secretory autophagy pharmacologic inhibitors have not yet been developed, it is necessary to identify additional pathways that can regulate NANOG-mediated secretory autophagy.

The transient receptor potential vanilloid 1 (TRPV1), a nonselective cationic channel, regulates $Ca^{2+}$ influx. TRPV1 is widely expressed in various tissues including the brain, kidney, bronchial epithelial cells, and keratinocytes in the epidermis and is not only involved in pain sensations but also other processes, such as autophagy, inflammation, and apoptosis[28,29]. Potential TRPV1 agonistic and antagonistic agents have been tested to treat migraines, osteoarthritis, overactive bladder, atopic dermatitis, and neuropathic pain[29,30]. Interestingly, TRPV1 was found to be associated with CIPN caused by different chemotherapeutic drugs[31,32]. In addition, TRPV1 has been applied as a potential target for cancer treatment because it was aberrantly expressed and promoted refractory phenotypes in the several tumor types[33,34]. In this regard, previous studies suggested that TRPV1 has a high value as an actionable target that could simultaneously reduce NANOG-mediated refractory phenotypes and CIPN. Although the importance of TRPV1 as a pharmacological target continues to grow, the potential relationship between TRPV1 and cisplatin resistance conferred by the NANOG axis is yet to be studied extensively.

Here, we demonstrate that cisplatin resistance can be caused by the enrichment of a subset of tumor cells with secretory autophagic phenotypes. The phenotypes of cisplatin-resistant tumor cells are dependent upon TRPV1, which is a direct transcriptional target of NANOG. We observe that TRPV1-mediated $Ca^{2+}$ influx increased autophagic EGF secretion, which activates the EGFR-AKT signaling pathway and, consequentially, the acquisition of cisplatin resistance. Importantly, TRPV1 inhibition using a small-molecule agent AMG9810, a potential pain-reliever, could sensitize resistant tumors to cisplatin by reversing anti-apoptotic properties. Thus, we provide proof-of-concept evidence that TRPV1 inhibition could be a plausible strategy to control NANOG[high] cisplatin-resistant cancer as well as CIPN.

## Results
### Hyperactivated EGFR confers cisplatin resistance in cervical cancer
Recent studies have been indicated that EGFR signaling is dysregulated in various cancer types that acquire cisplatin resistance[16]. To explore the potential relationship between EGFR signaling activity and cisplatin responsiveness in cervical cancer patients, we analyzed the transcriptome of The Cancer Genome Atlas (TCGA) cervical cancer patients. Either high or low EGFR signaling could be predicted by determining the expression signature scores of the gene sets that have been reported as indicators of EGFR signaling activity (hereafter referred to as EGFR activity score)[35]. Interestingly, we found that elevated EGFR activity scores were significantly associated with the poor overall survival of cisplatin-treated cervical cancer patients, but not cisplatin-untreated cervical cancer patients (Fig. 1a), suggesting hyperactivated EGFR signaling might be a crucial candidate that induces cisplatin resistance. We further questioned whether hyperactivated EGFR signaling could confer cisplatin resistance. By analyzing the EGFR activity scores in transcriptomic data on patients with TCGA cervical cancer patients classified as responders (R, complete or partial response) or non-responders (NR, stable or progressive disease) to cisplatin alone treatment, we found that EGFR activity scores were statistically higher in NR than in R (Fig. 1b). We also found that the above results were reproduced in TCGA cervical cancer patients classified as having low or high chemotherapy-resistant (CR) gene signature[36] (Supplementary Fig. 1). These results suggest that hyperactivated EGFR signaling could drive cisplatin resistance, thereby leading to poor clinical outcomes in patients with cervical cancer.

Previously, we established CaSki CR (a highly cisplatin-resistant cervical cancer cell line), which was generated from CaSki P (a cisplatin-susceptible parental cell line of CaSki), through three rounds of in vivo selection imposed by cisplatin[37]. Notably, the CaSki CR cells were resistant to apoptotic death by cisplatin, whereas the CaSki P cells remained sensitive[37]. To investigate whether hyperactivated EGFR signaling hampered cell death by cisplatin in the cisplatin-resistant tumor cells, we first performed Western blot analysis of lysates derived from CaSki P and CaSki CR cells. We found that EGFR was hyperactivated in CaSki CR cells compared to CaSki P cells (Fig. 1c). We previously reported that hyperactivated EGFR promoted resistance to multi-modal therapies, including cisplatin, γ-irradiation, as well as cognate cytotoxic T lymphocytes (CTLs), via anti-apoptotic protein MCL1 upregulation[14]. Consistently, the levels of MCL1 were increased in CaSki CR cells compared to CaSki P cells (Fig. 1c).

To determine the roles of hyperactivated EGFR in the cisplatin-resistant phenotype of CaSki CR cells, we inhibited EGFR in CaSki CR cells using small-molecule inhibitor of EGFR, such as gefitinib (Fig. 1d). Compared to the control cells, gefitinib-treated CaSki CR cells were more susceptible to apoptosis induced by cisplatin (Fig. 1e). Consistently, the knockdown of EGFR using two kinds of small interfering RNAs (siRNAs) (siEGFR #1 or #2) reversed the cisplatin-resistant phenotypes of CaSki CR cells (Fig. 1f, g). In contrast, EGFR inhibition did not alter susceptibility to cisplatin of CaSki P cells (Fig. 1d–g). To verify the reproducibility of the above results, we repeated experiments of Fig. 1c–g using two different types of cisplatin-resistant cervical cancer cell lines: SiHa CR and HeLa. SiHa CR is another newly established cisplatin-resistant tumor cell line, generated from SiHa P (a cisplatin-susceptible parental cell line of SiHa), and HeLa is commonly used in various drug resistance studies[38]. Consistent with the results from CaSki CR cells, SiHa CR displayed the resistance to cell death by cisplatin and the hyperactivated EGFR signaling compared to SiHa P cells (Supplementary Fig. 2a, b). Importantly, EGFR inhibition using gefitinib or siEGFR could reverse the cisplatin-resistant phenotypes of SiHa CR cells and HeLa cells (Supplementary Fig. 2c–f). Taken together, our data demonstrate that the hyperactivation of EGFR signaling pathway contributes to the cisplatin resistance in cervical cancer cells.

### Elevated autophagosome abundance confers cisplatin resistance of tumor cells through EGFR activation by EGF secretion
We next attempted to elucidate the underlying mechanism responsible for EGFR hyperactivation in cisplatin-resistant tumor cells.

Previously, we discovered that autophagy-mediated EGF secretion contributed to the hyperactivation of EGFR signaling, thereby triggering resistance to anti-cancer treatment[14]. Therefore, we hypothesized that autophagy-mediated EGF secretion might induce cisplatin resistance by activating EGFR signaling in cervical cancer. We first confirmed the level of autophagosome formation in cisplatin-resistant tumor cells. We found that autophagosome abundance was significantly increased in CaSki CR cells compared to CaSki P cells (Fig. 2a). Because the amount of LC3B-ll (lipidated form of LC3B-l) is clearly correlated with the number of autophagosome, we also assessed the absolute level of LC3B-ll (normalized by β-actin) by immunoblotting. Relative to CaSki P cells, CaSki CR cells exhibited escalated

LC3B-ll levels (Fig. 2b). Furthermore, transmission electron microscopy (TEM) analysis revealed that compared to CaSki P cells, CaSki CR cells exhibited an extensive accumulation of double and multi-membraned structures with a broad range of morphology types, presumably corresponding to stalled autophagosomes or autolysosomes (Fig. 2c). Interestingly, there was no significant alteration in autophagic flux in CaSki CR cells compared to CaSki P cells (Supplementary Fig. 3). Instead, the amount of LC3B-II escalated in parallel with increase in the amount of total cellular LC3B in CaSki CR cells (Fig. 2b). Therefore, our data indicate that cisplatin-resistant cervical cancer cells exhibit an increase in autophagosome abundance rather than autophagy flux.

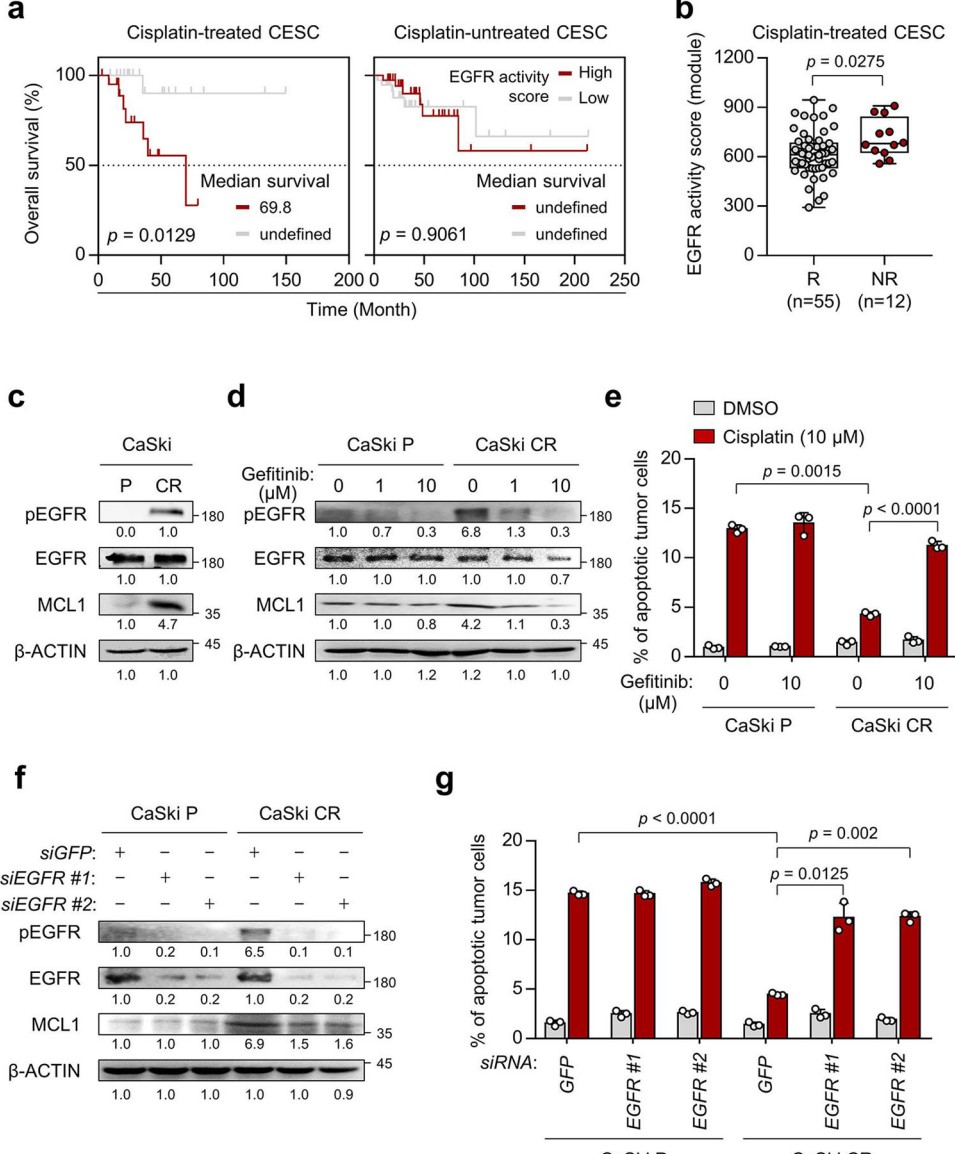

**Fig. 1 | Hyperactivated EGFR signaling promotes resistance to cisplatin, leading to poor survival in cervical cancer patients. a** Kaplan−Meier analysis of overall survival (calculated as months to death or months to last follow-up) in cisplatin-treated CESC or cisplatin-untreated CESC. The 30th and 70th percentiles were used as cutoffs values for the EGFR activity scores (EGFR activity score[high] > 70th; EGFR activity score[low] < 30th). **b** Comparison of the level of EGFR activity scores in responder (R, $n = 55$) and non-responder (NR, $n = 12$) of cisplatin-treated CESC cohort. **c** Western blot analysis of pEGFR, EGFR and MCL1 expression. **d, e** CaSki P and CR cells were treated with gefitinib as the indicated dose. **f, g** CaSki P and CR cells were transfected with siRNA targeting *GFP* or *EGFR*. **d, f** The protein levels of pEGFR, EGFR and MCL1 were determined by western blots. β-actin was used as an internal loading control. Numbers below the blot images indicate the expression as measured by fold-change. **e, g** Flow cytometry analysis of the frequency of apoptotic (active caspase 3[+]) cells after incubation with or without cisplatin for 24 h. All experiments were performed in triplicate. In the box plots, the top and bottom edges of boxes indicate the first and third quartiles, respectively; the center lines indicate the medians; and the ends of whiskers indicate the maximum and minimum values, respectively. The $p$ values by Gehan−Breslow−Wilcoxon test (**a**), two-tailed Student's $t$ test (**b**) or two-way ANOVA (**e, g**) are indicated. The data represent the mean ± SD. Source data are provided as a Source Data file.

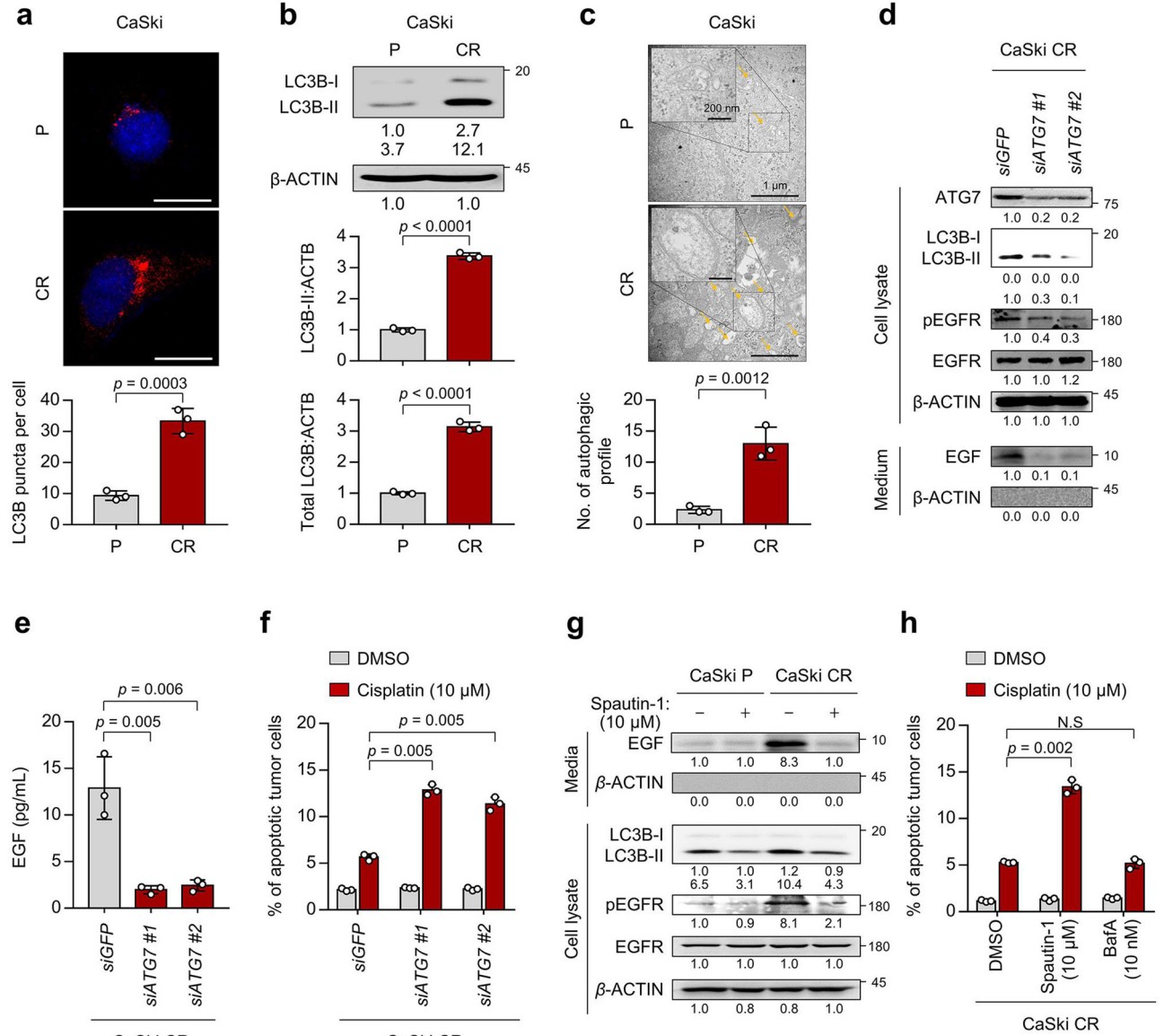

**Fig. 2 | Autophagy-mediated EGFR hyperactivation is required for cisplatin resistance. a** The cells were stained with anti-LC3B (red) antibodies and then visualized by confocal microscopy. DAPI was used to stain the nuclei. The images are representative of three separate experiments. Scale bar, 20 μm. The graph depicts the experimental quantitation of puncta. **b** The protein levels of LC3B were determined by western blot analysis. **c** CaSki P and CR cells starved for 24 hr in medium supplemented with 0.1% FBS were fixed and imaged by TEM. **d–f** CaSki CR cells were transfected with siRNA targeting *GFP* or *ATG7*. **d** The protein levels of secreted EGF and internal ATG7, LC3B, pEGFR, EGFR, and β-actin were confirmed by western blots. **e** The amount of EGF secreted into the media was measured by ELISA. **f** Flow cytometry analysis of the frequency of apoptotic (active caspase 3⁺)

cells after incubation with or without cisplatin for 24 h. **g** In the cells treated with or without spautin-1, the protein levels of secreted EGF and internal LC3B, pEGFR, EGFR, and β-actin were determined by western blots. **h** Flow cytometry analysis of the frequency of apoptotic (active caspase 3⁺) cells after incubation with or without cisplatin for 24 h. **b, c, g** β-actin was used as the internal loading control. Numbers below the blot images indicate the expression as measured by fold-change. All experiments were performed in triplicate. The *p* values by two-tailed Student's *t* test (**a–c**), one-way ANOVA (**e**), or two-way ANOVA (**f, h**) are indicated. The data represent the mean ± SD. Source data are provided as a Source Data file. (NS, not significant).

Since an increase of autophagosome abundance could trigger the secretion of intracellular molecule[39], we first investigated secretion levels of EGFR ligands. Previous studies indicated that EGFR activation is mediated by EGF and its related ligands, and that increased phosphorylation of EGFR through the autocrine growth factor loop plays an important role in human cancers[40]. In this regard, we assessed those levels of both intracellular and secreted EGFR ligands in CaSki cells. Among the EGFR ligands, only EGF secretion was increased in CaSki CR cells compared to CaSki P cells (Supplementary Fig. 4a, b). Notably, the neutralization of secreted EGF by its specific monoclonal antibody led to a significant decrease in the phosphorylation levels of EGFR and the

resistance to apoptotic cell death imposed by cisplatin in CaSki CR cells (Supplementary Fig. 4c, d), suggesting a direct role for EGF in the hyperactivated EGFR signaling-mediated cisplatin resistance. We next questioned whether autophagosome abundance could regulate EGF secretion and EGFR signaling-mediated cisplatin resistance. To directly link autophagosome abundance to the phenotype of CaSki CR cells, we silenced a core autophagosome gene *ATG7* in CaSki CR cells using siRNAs. The knockdown of *ATG7* not only abolished the formation of autophagosome shown by the markedly decreased LC3B-I to LC3B-II transition but also reduced EGF secretion and the level of pEGFR in CaSki CR cells (Fig. 2d, e). We also found that *ATG7* knockdown

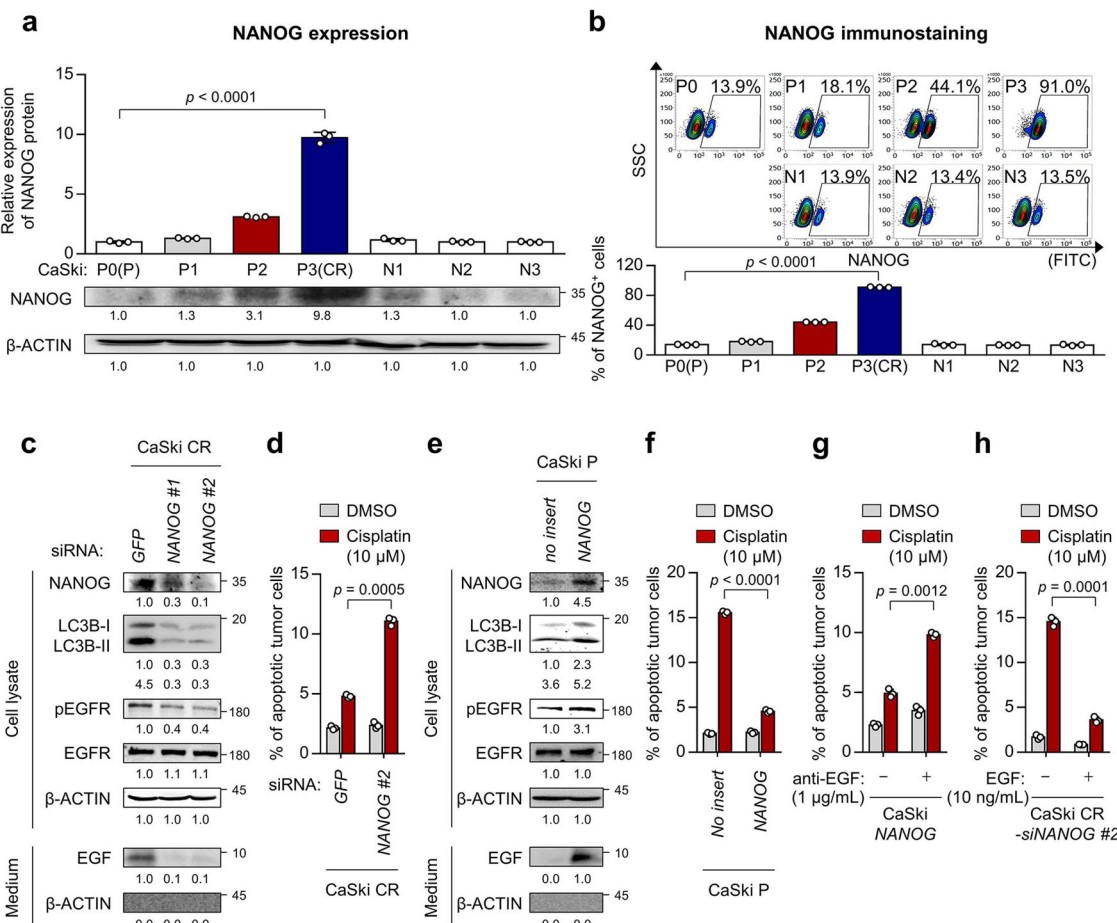

**Fig. 3 | NANOG promotes cisplatin resistance through autophagy-mediated EGFR activation. a** Top, quantification of NANOG expression in tumor cells at different stages of selection with cisplatin (P to CR). Parallel stages without selection are labeled as N1 to N3. Bottom, representative Western blot images. **b** Top, representative images of flow cytometry analysis of NANOG+ tumor cells. Bottom, quantification of the frequency of NANOG+ tumor cells. **c, d** CaSki CR cells were transfected with siRNA targeting *GFP* or *NANOG*. **e, f** CaSki P cells were transfected with empty vector (*no insert*) or *NANOG* wild-type. **c, e** The protein levels of secreted EGF and internal NANOG, LC3B, pEGFR, and EGFR were confirmed by western blots. β-actin was used as the internal loading control. Numbers below the blot images indicate the expression as measured by fold-change. **d, f** Flow cytometry analysis of the frequency of apoptotic (active caspase 3+) cells after incubation with or without cisplatin for 24 h. **g, h** In the cells treated with or without anti-EGF or EGF, the frequency of apoptotic (active caspase 3+) cells after incubation with or without cisplatin for 24 h were confirmed by flow cytometry. All experiments were performed in triplicate. The *p* values by one-way ANOVA (**a, b**) or two-way ANOVA (**d, f, g, h**) are indicated. The data represent the mean ± SD. Source data are provided as a Source Data file.

reduced the resistance of CaSki CR cells to apoptotic cell death by cisplatin (Fig. 2f). We further investigated which stages of the autophagy pathway is crucial for EGF secretion and EGFR signaling-mediated cisplatin resistance by using two other autophagy inhibitors, spautin-1 and bafilomycin A1 (BafA). Spautin-1 is a specific inhibitor of the PtdIns3K catalytic subunit (PIK3C3/VPS34) responsible for autophagosome formation, whereas BafA is an inhibitor of V-ATPase, which blocks the fusion of autophagosomes to lysosomes[41]. Interestingly, the elevated EGF secretion and the cisplatin resistance in CaSki CR cells were significantly reduced by the treatment with spautin-1 but not with BafA (Fig. 2g, h and Supplementary Fig. 5), indicating that autophagosome formation is crucial for EGF secretion and EGFR signaling-mediated cisplatin-resistant phenotypes in CaSki CR cells, but late autophagy components involved in autophagosome-lysosome fusion appear to be dispensable in the process. Colinearly, compared to SiHa P cells, SiHa CR cells also showed increase in autophagosome formation and EGF secretion (Supplementary Fig. 6a). Notably, the suppression of autophagosome formation by spautin-1 reduced EGF secretion, the level of pEGFR and cisplatin-resistant phenotypes in SiHa CR cells (Supplementary Fig. 6a, b). Given these data, we conclude that autophagosome-mediated EGF secretion induces the

hyperactivation of EGFR signaling, thereby promoting chemoresistance to cell death by cisplatin in cervical cancer cells.

## NANOG confer cisplatin resistance through autophagy-mediated EGFR activation

We next aimed to elucidate the underlying mechanism implicated in secretory autophagy-mediated EGFR activation in cisplatin-resistant tumor cells. Previously, we reported that NANOG was a crucial transcription factor driving the multi-refractory phenotypes of tumor cells, such as stem-like, metastatic and anti-apoptotic properties and resistance to multi-modal therapies via autophagy-mediated hyperactivation of the EGFR-AKT pathway[14]. To characterize the role of NANOG in the acquisition of cisplatin resistance by autophagy-mediated EGFR activation in cisplatin-resistant tumor cells, we first assessed the expression of NANOG in CaSki cells at different rounds of in vivo selection by cisplatin (CaSki P to CR) and found a stepwise increase in NANOG expression from CaSki P to CR (Fig. 3a). The overall increase in NANOG expression in the CaSki CR cells was likely due to the enrichment of NANOG+ cells, as opposed to the upregulation of NANOG expression itself, as the frequency of NANOG+ cells rose from around 13.9% in the CaSki P cells to around 91.0% in the CaSki CR cells (Fig. 3b).

In contrast, there was no significant increase in NANOG levels in tumor cells selected without cisplatin (N1 to N3; Fig. 3a, b). Consistently, the increased expression of NANOG was also observed in another cisplatin-resistant tumor model (SiHa CR) (Supplementary Fig. 7a). Thus, these results demonstrate that cisplatin depletes tumor cells lacking NANOG while enriching tumor cells containing NANOG, suggesting that NANOG expression in tumor cells could confer a survival advantage under the selection pressure imposed by cisplatin.

To further investigate whether NANOG expression in tumor cells is responsible for the acquisition of cisplatin resistance by autophagy-mediated EGFR activation, we silenced the *NANOG* gene in cisplatin-resistant tumor cells using siRNAs. The knockdown of *NANOG* in the cisplatin-resistant tumor cells led to a significant decrease in autophagosome abundance, EGF secretion and the level of pEGFR (Fig. 3c and Supplementary Fig. 7b), suggesting an important role of NANOG in regulating the autophagy-mediated EGFR activation of cisplatin-resistant tumor cells. Notably, we found that *NANOG* knockdown reduced the resistance of cisplatin-resistant tumor cells to apoptotic cell death by cisplatin (Fig. 3d and Supplementary Fig. 7c). Conversely, the *NANOG* gene-transduced parental cells (CaSki *NANOG*) were phenotypically similar to cisplatin-resistant cells (CaSki CR), displaying cisplatin resistance (Fig. 3e, f). We further questioned that the role of secreted EGF in NANOG-mediated cisplatin resistance. To verify the potential role of EGF in the NANOG-mediated cisplatin resistance in tumor cells, we neutralized EGF with a specific monoclonal antibody in CaSki *NANOG* cells. The EGF neutralization decreased the NANOG-mediated cisplatin-resistant phenotype (Fig. 3g). Conversely, the treatment of recombinant EGF to *NANOG*-silenced CaSki CR cells increased resistance to apoptotic cell death by cisplatin (Fig. 3h), suggesting an immediate role for EGF in the NANOG-mediated cisplatin resistance. Taken together, our results demonstrate that NANOG is a critical mediator that could confer cisplatin resistance by regulating secretory autophagy-mediated EGFR activation.

## NANOG directly regulates TRPV1 through promoter occupancy

We showed that NANOG conferred anti-apoptotic phenotypes in response to anti-cancer agents by promoting autophagic EGF secretion in therapeutic-resistant tumor cells[14]. Although the data presented in this study suggest that targeting NANOG-mediated autophagic secretion could be a promising approach for overcoming cisplatin resistance, pharmacologic inhibitors of these phenotypes are yet to be developed. To investigate potential targets for impeding NANOG-driven autophagy in cisplatin-resistant tumor cells, we compared the expression of genes involved in EGFR hyperactivation among the genes upregulated by NANOG and highly expressed in chemotherapy-resistant cervical cancer patients (Fig. 4a). Among the putative target genes, *ACSBG2*, *DUOX2*, and *TRPV1* were previously reported that could be involved in various chemotherapy-refractory phenotypes. For example, Acyl-CoA Synthetase Bubblegum Family Member 2 (ACSBG2) induces resistance to apoptotic cell death by increasing mitochondrial membrane lipids[42]. In addition, Dual Oxidase 2 (DUOX2) induces epithelial mesenchymal transition through production of reactive oxygen species[43]. Although these putative targets have been reported to be related to chemotherapy-refractory phenotypes, their clinically available agents have not been developed. On the other hand, TRPV1 is not only involved in various chemotherapy-refractory phenotypes, but also clinically available agents have been developed. Thus, we focused on TRPV1 as an available target to overcome NANOG-mediated autophagic secretion. TRPV1 is a Ca²⁺-permeable nonselective cation channel modulated by specific small-molecule ligands such as capsaicin and is not only involved in pain sensations but also in other processes such as inflammation, autophagy, and apoptosis. Indeed, recent studies suggested that TRPV1 is potentially an actionable target that could simultaneously overcome the refractory phenotypes of tumor cells and CIPN[30,44,45]. To investigate TRPV1 as a target for impeding

NANOG-driven autophagy in cisplatin-resistant tumor cells, we monitored TRPV1 expression in cisplatin-susceptible or -resistant tumor cells. Interestingly, TRPV1 was significantly overexpressed in cisplatin-resistant tumor cells compared to cisplatin-susceptible tumor cells (Fig. 4b and Supplementary Fig. 8a). To determine whether TRPV1 is functionally active in these cells, we examined TRPV1-induced electrical currents upon exposure to its agonist, capsaicin. TRPV1-specific inward currents were elicited by capsaicin and were abolished by the TRPV1-specific antagonist AMG9810 in CaSki CR cells, but not in CaSki P cells (Fig. 4c). Previous studies reported that TRPV1-mediated intracellular Ca²⁺ influx induced autophagy by promoting autophagosome formation[46,47]. Interestingly, TRPV1 inhibition reduced autophagosome formation, thereby impeding autophagy-driven EGFR signaling (Fig. 4d) as well as increasing the sensitivity of CaSki CR cells to cisplatin in a dose-dependent manner (Fig. 4e). Moreover, the above results were also reproduced in SiHa CR or HeLa cells (Supplementary Fig. 8b, c). Thus, our results suggest that TRPV1, which is functionally active as a Ca²⁺-permeable cation channel, could confer cisplatin resistance by regulating autophagosome formation in cisplatin-resistant tumor cells.

We further investigated whether NANOG regulates TRPV1 expression through its transcriptional functions. To address this, we first silenced NANOG expression in cisplatin-resistant tumor cells. Notably, the elevated level of *TRPV1* mRNA in cisplatin-resistant tumor cells was repressed by *NANOG* knockdown, indicating the NANOG-dependent expression of TRPV1 (Fig. 4f and Supplementary Fig. 9). For further confirmation of this possibility, we next used a mutant form of *NANOG* (*NANOG* MT) that was previously found to have weak transcriptional activity[20]. When we transduced CaSki and HEK293 cells with wild-type *NANOG* (*NANOG* WT), mRNA and protein of TRPV1 were significantly increased, whereas *NANOG* MT had no profound impact on either TRPV1 mRNA or protein levels (Fig. 4g, h), demonstrating that NANOG regulates TRPV1 expression through its transcriptional function. To further account for the underlying mechanism by which NANOG regulates TRPV1 transcription, we identified the *TRPV1* promoter region containing a putative NANOG-binding site (Fig. 4i), suggesting the possibility that NANOG was a direct transcriptional activator of TRPV1. Consistently, chromatin immunoprecipitation (ChIP) assays confirmed the direct binding of NANOG to the regulatory region of the *TRPV1* gene (Fig. 4j). Thus, our results clearly demonstrate that NANOG upregulates TRPV1 transcription by directly binding to its promoter region.

## TRPV1 promotes NANOG-mediated cisplatin resistance through the Ca²⁺-AMPK pathway

We then questioned whether TRPV1 potentiates the refractory phenotypes of NANOG⁺ tumor cells. To directly link TRPV1 to these phenotypes, we silenced TRPV1 expression in CaSki *NANOG* and HEK293 *NANOG* cells (Fig. 5a and Supplementary Fig. 10a). Notably, *TRPV1* knockdown decreased autophagy-mediated activation of EGFR-AKT pathway (Fig. 5a and Supplementary Fig. 10a) and increased sensitivity to cisplatin in CaSki *NANOG* and HEK293 *NANOG* cells (Fig. 5b and Supplementary Fig. 10b). Consistent with the above results, we found that the NANOG-mediated phenotypes were dampened by TRPV1 inhibition using AMG9810 (Fig. 5c, d and Supplementary Fig. 10c, d), indicating TRPV1 plays a crucial role in NANOG-mediated cisplatin resistance. Previous reports indicated that TRP channels could promote autophagosome formation and multi-refractory phenotypes through the Ca²⁺-AMPK pathway[48]. Interestingly, TRPV1 activation by capsaicin elicited increase in intracellular Ca²⁺ influx, which were abolished by TRPV1-specific inhibitor AMG9810, in CaSki *NANOG* cells, whereas no influence on Ca²⁺ influx in CaSki *no insert* cells was observed (Fig. 5e, f). Indeed, intracellular Ca²⁺ signaling blockade by *CaMKKβ* knockdown reduced the NANOG-mediated phenotypes (Fig. 5g, h and Supplementary Fig. 11). Taken together, our findings

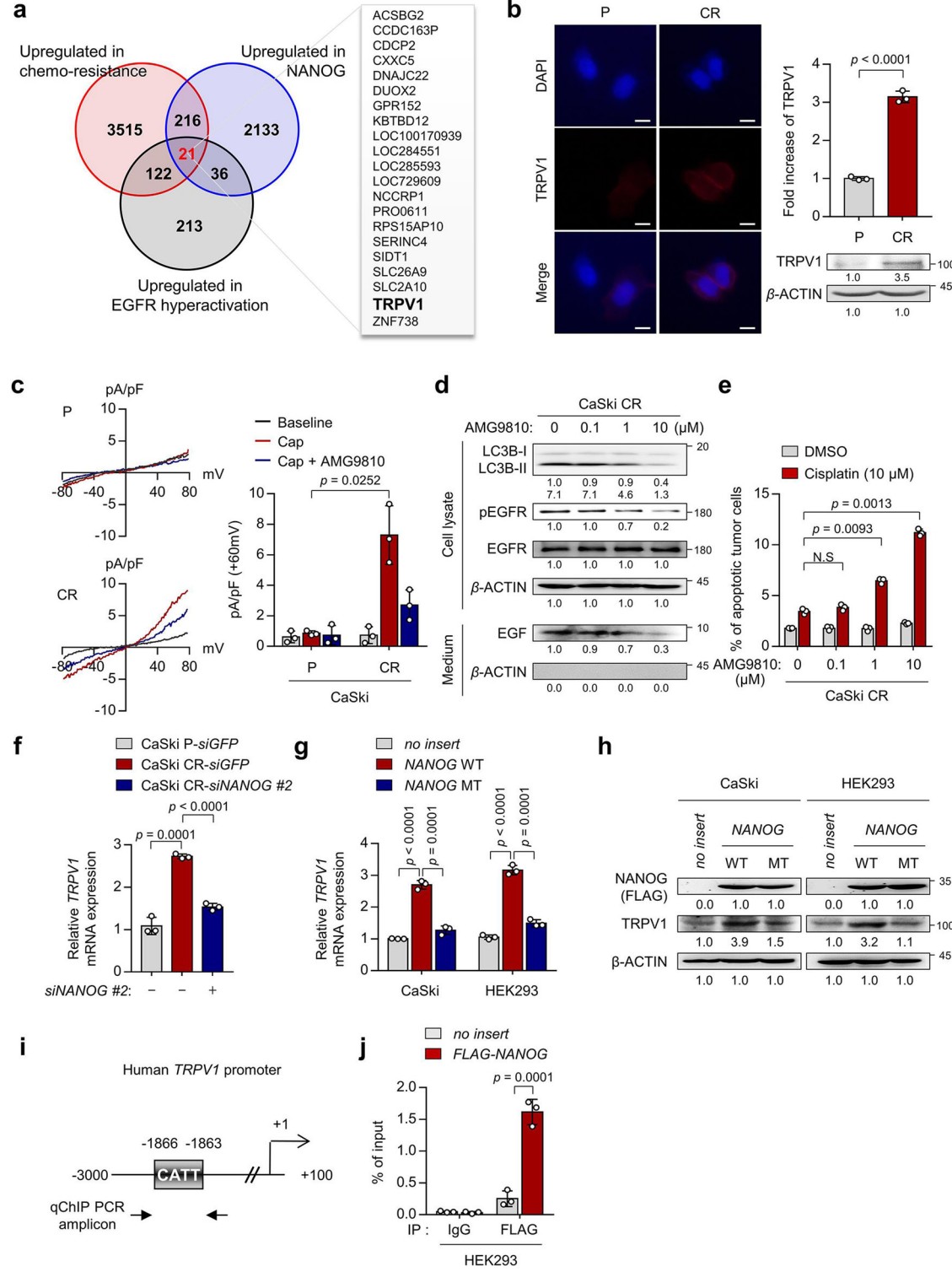

indicate that the TRPV1 channel plays an important role in the secretory autophagy-dependent refractory phenotypes of NANOG+ tumor cells.

## The NANOG-TRPV1-pEGFR axis in human cervical neoplasia correlates with resistance to chemoradiation therapy and prolonged survival rates

We previously revealed that the high expression of NANOG and pEGFR correlated with the poor outcomes of patients with cervical cancer[14,20,23]. Given that NANOG and pEGFR have been defined as poor prognostic factors in patients with cervical cancer, we further assessed the potential association between NANOG-TRPV1-pEGFR expression and chemoradiation resistance in human cervical cancer by immunohistochemistry (Fig. 6a). The high expression of NANOG and TRPV1 was correlated with chemoradiation resistant tumors ($p = 0.035$ and $p = 0.013$, respectively), whereas the high expression of pEGFR tended to correlate with chemoradiation resistance (Fig. 6b). The combinations of NANOG+/TRPV1+ ($\chi^2$ test; $p < 0.001$), NANOG+/pEGFR+ ($\chi^2$ test; $p = 0.003$), TRPV1+/pEGFR+ ($\chi^2$ test; $p = 0.002$), and NANOG+/TRPV1+/pEGFR+ ($\chi^2$ test; $p = 0.001$) were significantly higher in

**Fig. 4 | NANOG directly regulates TRPV1 through promoter occupancy. a** Venn diagram showing the overlap of genes upregulated in high versus low chemotherapy-resistant cervical cancer patients (red), CaSki *NANOG* versus CaSki *no insert* (blue), and cervical cancer patients from the TCGA cohort classified as having high or low EGFR activity score (dark). **b** The cells were stained with anti-TRPV1 (red) antibodies and then visualized by confocal microscopy. DAPI was used to stain the nuclei. Scale bar, 10 μm. The graph depicts the experimental quantitation of TRPV1. The protein levels of TRPV1 and β-actin were confirmed by western blots. **c** Current-voltage relationships from capsaicin (Cap, red line) or AMG9810 (AMG, blue line)-induced whole-cell current in CaSki P versus CR cells. Summary of average outward current at +60 mV during Cap exposure in CaSki P versus CR cells. **d** The protein levels of secreted EGF and internal LC3B, pEGFR, and EGFR were confirmed by western blots. **e** Flow cytometry analysis of the frequency of apoptotic (active caspase 3+) cells after incubation with or without cisplatin for 24 h.

**f** mRNA expression of *TRPV1* was analyzed by qRT-PCR. **g** *TRPV1* mRNA levels were measured by qRT-PCR. **h** Levels of TRPV1 and FLAG-NANOG proteins were confirmed by western blot. **i** Diagram of the human TRPV1 promoter region containing the NANOG-binding site. The arrows indicate the qChIP-PCR amplicon. **j** The cross-linked chromatin from HEK293 cells transfected with empty vector or FLAG-*NANOG* was immunoprecipitated with rabbit IgG or anti-FLAG antibodies. Relative enrichment of FLAG-NANOG in the TRPV1 promoter region was assessed by qChIP-PCR analysis with primers that amplified the genomic region indicated above. The ChIP data values represent the relative ratio to the input. Numbers below the blot images indicate the expression as measured by fold-change. All experiments were performed in triplicate. The *p* values by two-tailed Student's *t* test (**b**), two-way ANOVA (**c**, **e**), or one-way ANOVA (**f**, **g**, **j**) are indicated. The data represent the mean ± SD. Source data are provided as a Source Data file. (NS, not significant).

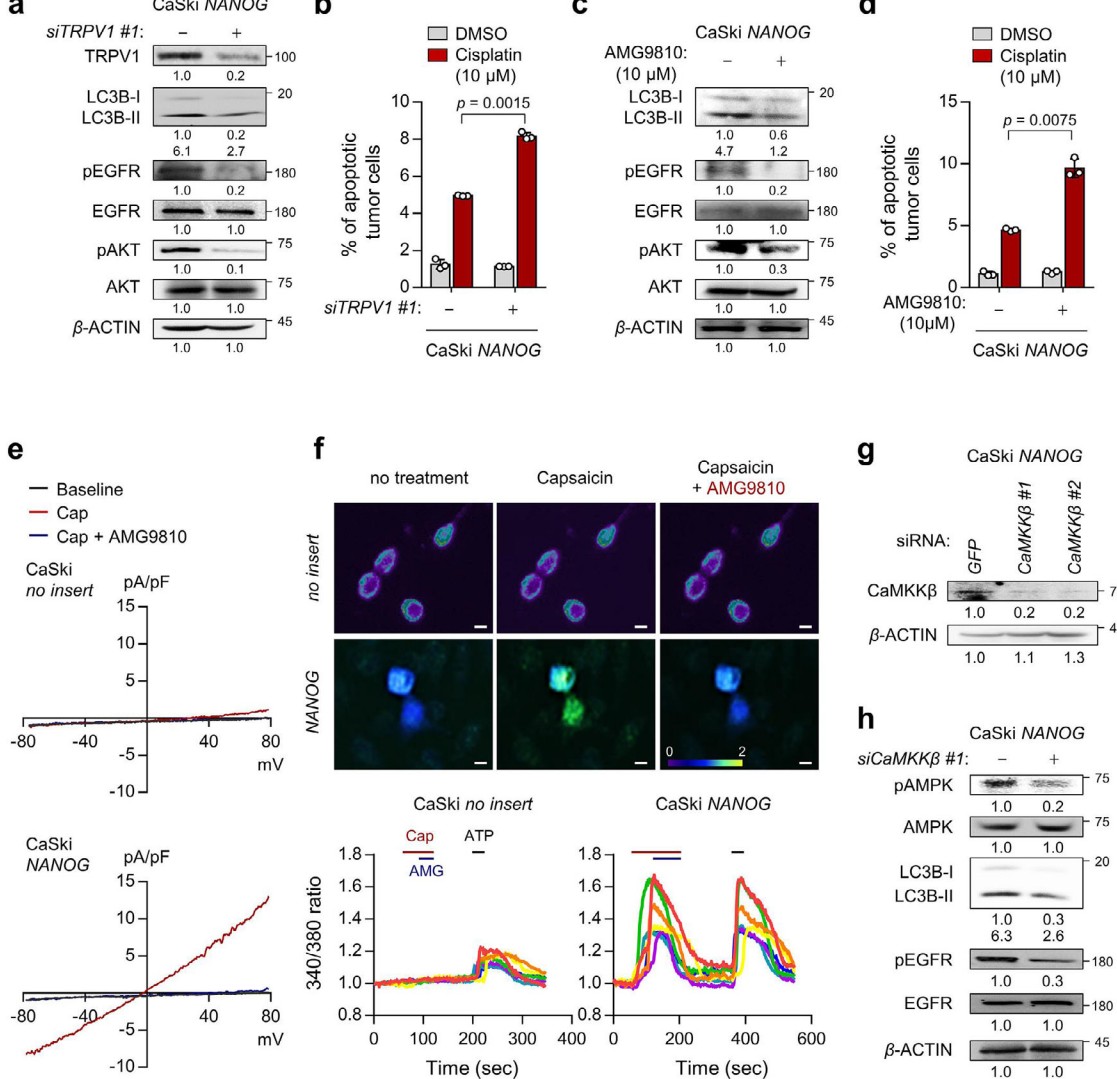

**Fig. 5 | TRPV1 promotes cisplatin resistance through Ca²⁺-AMPK pathway in NANOG⁺ tumor cells. a**, **b** CaSki *NANOG* cells were transfected with siRNA targeting *GFP* or *TRPV1*. **c**, **d** CaSki *NANOG* cells were treated with DMSO or AMG9810. **a**, **c** The levels of TRPV1, LC3B, pEGFR, EGFR, pAKT, and AKT proteins were confirmed by western blot analysis. **b**, **d** The frequency of apoptotic (active caspase 3+) cells were analyzed by flow cytometry. **e** Current-voltage relationships in capsaicin (Cap, red line) or AMG9810 (AMG, blue line)-induced whole-cell current in CaSki *no insert* versus CaSki *NANOG* cells. **f** Fluorescence calcium imaging by TRPV1 activity in tumor cells. Top, representative calcium images of FURA-2-AM-loaded cells. Bottom, changes in intracellular Ca²⁺ current over time in tumor cells. **g**, **h** CaSki *NANOG* cells were transfected with siRNA targeting *GFP* or *CaMKKβ*. The expression levels of indicated protein were confirmed by western blots. **a**, **c**, **g**, **h** β-actin was used as the internal loading control. Numbers below the blot images indicate the expression as measured by fold-change. All experiments were performed in triplicate. The *p* values by two-way ANOVA (**b**, **d**) are indicated. The data represent the mean ± SD. Source data are provided as a Source Data file.

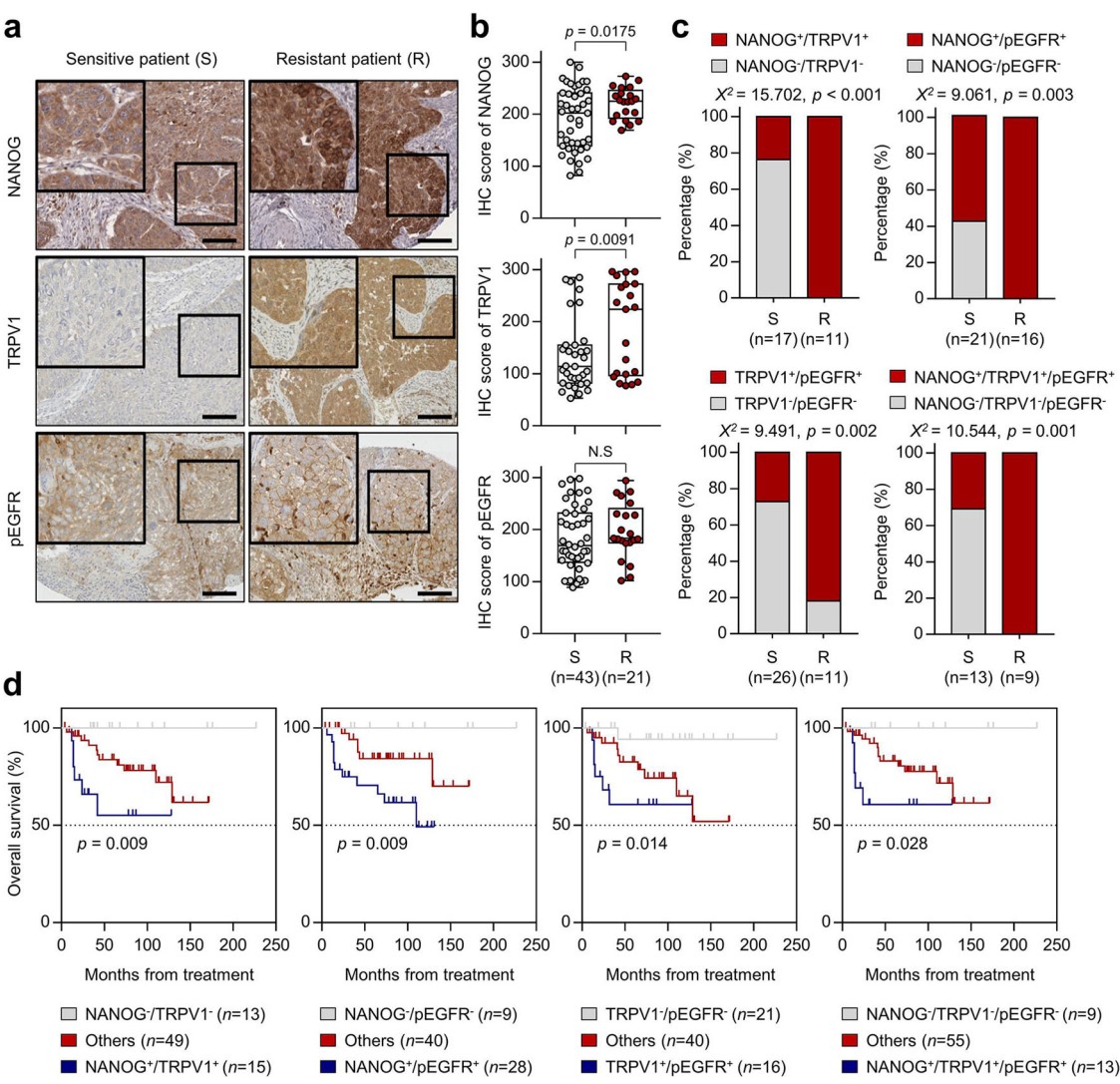

**Fig. 6 | NANOG, TRPV1, and pEGFR expression in human cervical cancer patients. a** Representative image of immunohistochemical staining in chemoradiosensitive and chemoradioresistant patient specimens. The boxed regions are displayed at high magnification in the inset. Scale bar, 100 μm. **b** Box plot depiction of NANOG, TRPV1, and pEGFR protein expression in chemoradiosensitive ($n = 43$) and chemoradioresistant ($n = 21$) cervical cancer patients. **c** Combinations of NANOG, TRPV1, and pEGFR expression were compared between chemoradiosensitive and chemoradioresistant cervical cancer cases. **d** Kaplan−Meier plots of overall survival according to combinations of NANOG, TRPV1, and pEGFR expression. In the box plots, the top and bottom edges of boxes indicate the first and third quartiles, respectively; the center lines indicate the medians; and the ends of whiskers indicate the maximum and minimum values, respectively. The $p$ values by two-tailed Student's $t$ test (**b**), Mann−Whitney $U$ test (**c**), or Gehan−Breslow−Wilcoxon test (**d**) are indicated. The data represent the mean ± SD. Source data are provided as a Source Data file. (NS, not significant).

chemoradioresistant cervical cancer patient tissues than in chemoradiosensitive cervical cancer patient specimens (Fig. 6c). Kaplan−Meier plots demonstrated that patients with NANOG⁺/TRPV1⁺ (overall survival (OS) rate, 60.0%), NANOG⁺/pEGFR⁺ (OS rate, 60.7%), TRPV1⁺/pEGFR⁺ (OS rate, 62.5%), or NANOG⁺/TRPV1⁺/pEGFR⁺ (OS rate, 61.5%) had worse overall survival than patients with NANOG⁻/TRPV1⁻ (OS rate, 100.0%), NANOG⁻/pEGFR⁻ (OS rate, 100.0%), TRPV1⁻/pEGFR⁻ (OS rate, 95.2%), or NANOG⁻/TRPV1⁻/pEGFR⁻ (OS rate, 100.0%) (Fig. 6d). Thus, our results indicate that the NANOG-TRPV1-pEGFR axis links a potential association with a chemoradiation responsiveness and a patient survival in human cervical cancer.

**TRPV1 promotes resistance to cell death by cisplatin via EGF-EGFR signaling pathway**
Given the important role the TRPV1 in cisplatin-resistant phenotypes of NANOG⁺ tumor cells, we questioned that TRPV1 expression alone could induce these phenotypes. We found that transfection of CaSki P cells with TRPV1 not only elevated autophagosome formation shown by the markedly increased LC3B-I to LC3B-II transition but also induced phosphorylation of EGFR (Fig. 7a). Furthermore, we found that TRPV1 overexpression in CaSki P cells induced a resistance to apoptotic cell death by cisplatin (Fig. 7b). Interestingly, we could not observe any changes in the levels of NANOG upon TRPV1 overexpression (Supplementary Fig. 12). These results suggest that ectopic expression of TRPV1 could induce the chemoresistance of tumor cells to cisplatin.

Notably, the inhibition of TRPV1 activity using AMG9810 failed to enhance the EGF secretion, activation of EGFR signaling as well as cisplatin-resistant phenotypes (Fig. 7a−c). These data indicate a key role of TRPV1 channel activity on those properties. Importantly, TRPV1-trasnfected cells had markedly higher levels of EGF secretion, compared with empty vector-transfected cells or AMG9810-treated CaSki TRPV1 cells (Fig. 7c). The neutralization of secreted EGF by antibody led to a significant decrease in the levels of pEGFR in CaSki TRPV1 cells (Fig. 7d), suggesting a direct role for EGF in the TRPV1-mediated EGFR

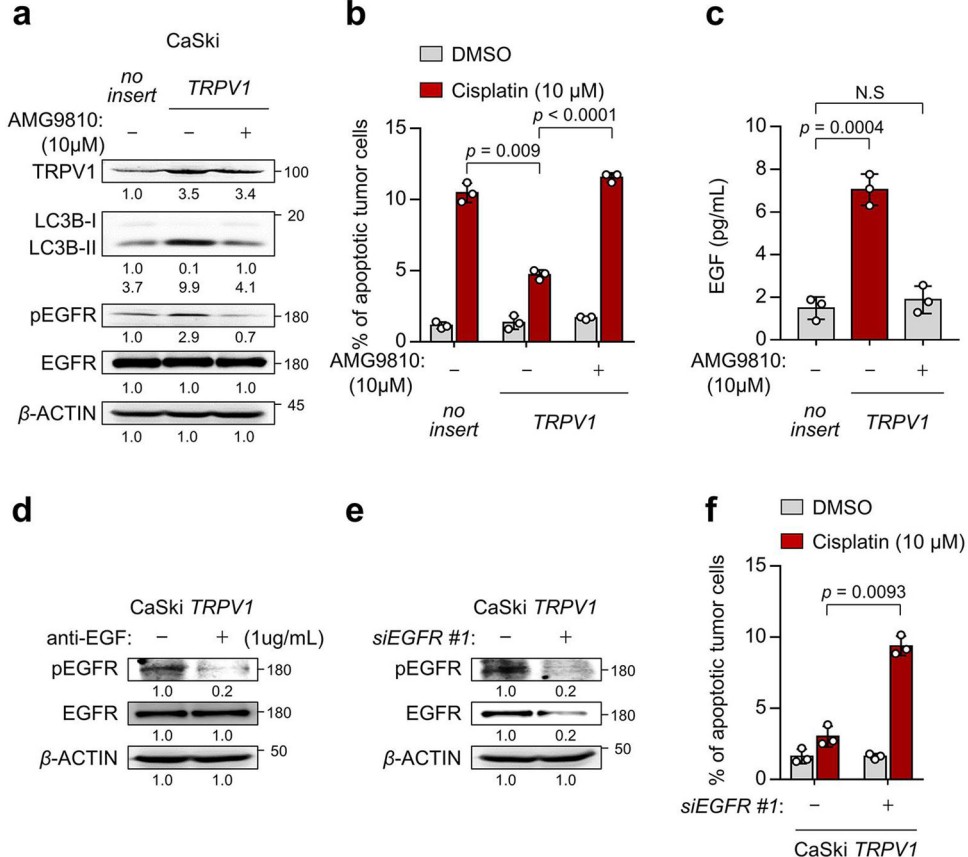

**Fig. 7 | TRPV1 promotes cisplatin resistance through EGF-EGFR signaling pathway. a–c** CaSki *no insert* and *TRPV1* cells were treated with DMSO or AMG9810, as indicated. **a** The protein levels of TRPV1, LC3B, pEGFR, and EGFR were confirmed by western blots. **b** The frequency of apoptotic (active caspase 3⁺) cells were analyzed by flow cytometry. **c** The amount of EGF secreted into the media was measured by ELISA. **d** CaSki *TRPV1* cells were treated with IgG or anti-EGF. Levels of pEGFR and EGFR were confirmed by western blots. **e** and **f** CaSki *TRPV1* cells were transfected with siRNA targeting *GFP* or *EGFR*. **e** The protein levels of pEGFR and

EGFR were confirmed by western blot analysis. **f** Flow cytometry analysis of the frequency of apoptotic (active caspase 3⁺) cells after incubation with or without cisplatin for 24 h. **a, d, e** β-actin was used as the internal loading control. Numbers below the blot images indicate the expression as measured by fold-change. All experiments were performed in triplicate. The *p* values by two-way ANOVA (**b, f**) or one-way ANOVA (**c**) are indicated. The data represent the mean ± SD. Source data are provided as a Source Data file. (NS, not significant).

signaling pathway. Furthermore, EGFR knockdown reduced the resistance of TRPV1-transfected tumor cells to cisplatin (Fig. 7e, f). Thus, our results indicate that TRPV1 by itself was sufficient to induce the cisplatin resistance of tumor cells via EGF-EGFR signaling pathway.

## TRPV1 is a therapeutic target for cisplatin-resistant NANOG⁺ cancer cells

To verify the functional effects of the NANOG-TRPV1-pEGFR axis in diverse types of cancer which cisplatin was used as the major chemotherapeutic agent, we focused on lung and gastric cancer among the various cancer types. We first mined lung and gastric cancer patients from the TCGA cohort. Interestingly, the NANOG signature, which is a gene set used to acquire a more reliable readout indicating NANOG functionality in tumor cells[24], was upregulated in chemotherapy-resistant LUAD (lung adenocarcinoma), LUSC (lung squamous cell carcinoma), STAD (stomach adenocarcinoma), and STES (stomach and esophageal carcinoma) cancer patients (Supplementary Fig. 13). These data suggest that the NANOG-TRPV1-pEGFR axis may confer cisplatin resistance in lung or gastric cancer patients. To further confirm whether a cisplatin resistance caused by the NANOG-TRPV1-pEGFR axis was conserved in NANOG^high lung or gastric cancer cells, we first tried to select NANOG^hgih lung or gastric cancer cell lines, respectively. In our previous studies, we profiled the levels of NANOG protein in a variety of human cancer cells[23,25]. Among lung cancer cells, we designated H1299 as a representative NANOG^high lung

cancer cell line. In the case of gastric cancer cells, we newly screened the levels of NANOG protein and selected MKN28 as a representative NANOG^high gastric cancer cell (Supplementary Fig. 14). Interestingly, *NANOG* knockdown reduced TRPV1 expression in those tumor cells (Supplementary Fig. 15), indicating that the NANOG axis was conserved in all tested cells. Notably, TRPV1 inhibition by AMG9810 robustly dampened EGFR or AKT phosphorylation and downregulated MCL1 expression in the tumor cells (Supplementary Fig. 16a). Furthermore, treatment with AMG9810 led to a significantly increase in apoptotic cell death caused by cisplatin (Supplementary Fig. 16b). Thus, our results demonstrate that the functional properties of the NANOG-TRPV1-pEGFR axis are conserved across NANOG^high lung and gastric cancer cells, and TRPV1 could be a potential target for controlling NANOG^high cisplatin-resistant cancer.

## TRPV1 inhibition sensitizes NANOG^high tumor cells to cisplatin

Given our observation in vitro, we reasoned that the in vivo administration of TRPV1 inhibitor, AMG9810, could overcome cisplatin resistance. To investigate the effect of AMG9810 on sensitizing NANOG^high tumor cells to cisplatin treatment, CaSki CR cells were co-treated with cisplatin and AMG9810 at various dose. Compared to cisplatin alone treatment, combined treatment of cisplatin with AMG9810 increased apoptotic cell death in dose-dependent manner (Fig. 8a). To quantify the synergistic effects of treatment of AMG9810 with cisplatin, a combination score was calculated based on changes in the percentage

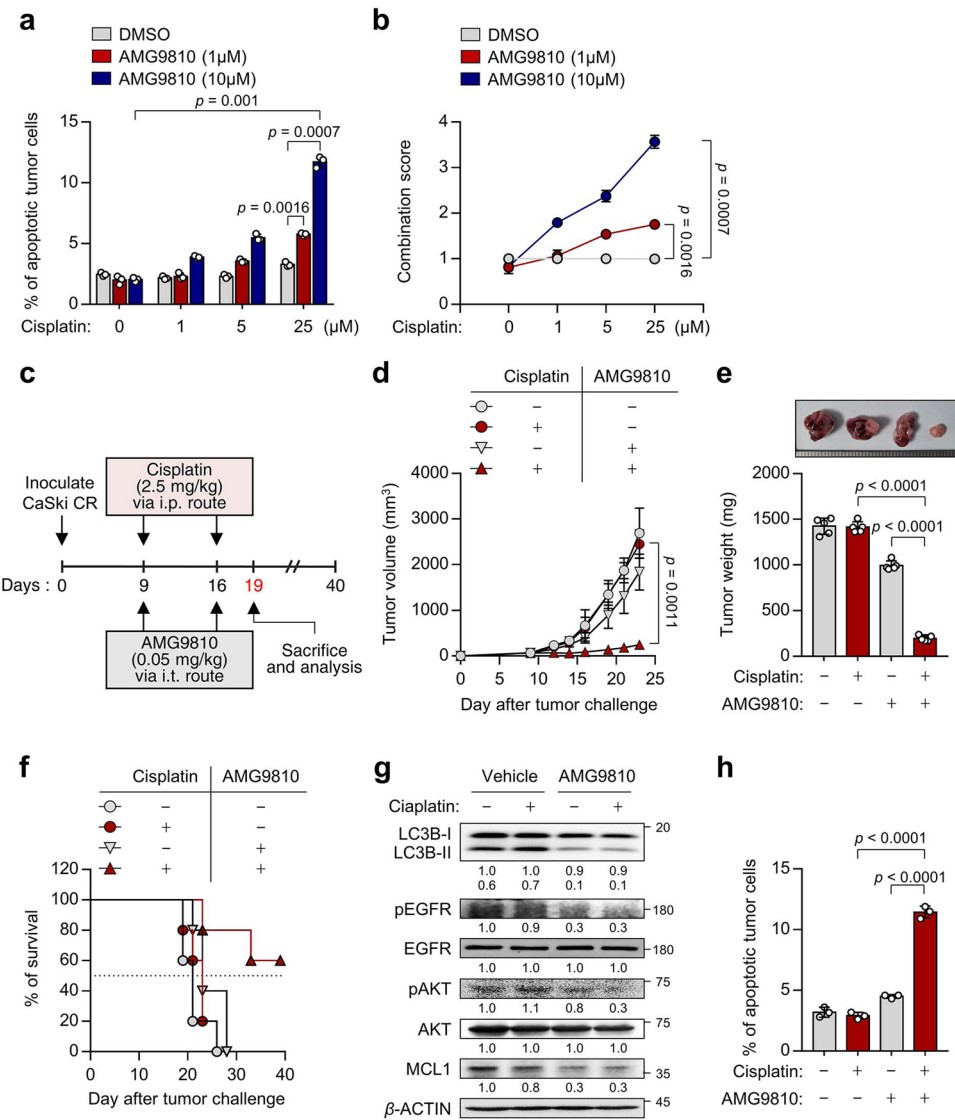

**Fig. 8 | In vivo targeting of autophagy by TRPV1 inhibition renders tumors susceptible to cisplatin in a preclinical model. a** CaSki CR cells were treated with cisplatin and AMG9810, as indicated. The frequency of apoptotic (active caspase 3⁺) cells was analyzed by flow cytometry. **b** The combination score was calculated based on changes in the percentage of apoptotic cells in cisplatin-treated tumor cells with or without AMG9810. Combination score = (% of active caspase 3⁺ tumor cells by cisplatin and AMG9810)/((% of active caspase 3⁺ tumor cells by cisplatin). **c** Schematic of the therapeutic regimen in mice implanted with CaSki CR cells. **d** tumor growth, **e** mass (at 19 d after challenge), and **f** survival of mice inoculated with CaSki CR cells treated with the indicated reagents. **g** The expression levels of LC3B, pEGFR, EGFR, pAKT, AKT, and MCL1 protein were confirmed by western blot analysis. β-actin was used as the internal loading control. Numbers below the blot images indicate the expression as measured by fold-change. **h** The frequency of apoptotic (active caspase 3⁺) cells was analyzed by flow cytometry in mice administrated DMSO or cisplatin, with or without AMG9810. For the in vivo experiments, five mice from each group were used. All experiments were performed in triplicate. The *p* values by two-way ANOVA (**a**, **b**, **d**) or one-way ANOVA (**e**, **h**) are indicated. The data represent the mean ± SD. Source data are provided as a Source Data file.

of apoptotic cell death in cisplatin-treated tumor cells with or without AMG9810. We found that treatment with a sublethal dose of AMG9810 enhanced the cisplatin-mediated cell death in a synergistic fashion (Fig. 8b). Given these data, we concluded that AMG9810 could be a plausible therapeutic strategy to overcome the cisplatin-resistant phenotypes of NANOG^high tumor cells.

To evaluate the preclinical therapeutic value of TRPV1 targeting, we inoculated the cisplatin-resistant CaSki CR cells into NOD-SCID mice. Nine days after the tumor challenge, the mice were treated with cisplatin along with chitosan hydrogel containing AMG9810 (Fig. 8c). While cisplatin treatment alone had no effect on the growth of cisplatin-resistant tumors, combined treatment with cisplatin and AMG9810 retarded tumor growth (Fig. 8d, e) and prolonged the survival of the mice (Fig. 8f). Consistent with our in vitro results, we

observed reduced levels of LC3B, pEGFR, pAKT and MCL1 in tumor tissue from AMG9810-treated mice compared to vehicle-treated mice (Fig. 8g). Notably, we found that the percentage of apoptotic tumor cells was increased by the combined treatment compared to either treatment alone (Fig. 8h). Taken together, we conclude that TRPV1 inhibition could overcome the cisplatin resistance of tumor cells and represent an attractive strategy for the control of human cancer as a synergistic agent with cisplatin treatment.

## Discussion

Chemotherapy remains the mainstay of treatment in the majority of solid and hematological malignancies[49]. However, the emergence of resistance or neuropathic pain caused by anti-cancer drug is the major cause of treatment cessation in patients undergoing chemotherapy[5,6].

Therefore, further studies need to construct a new proof-of-concept based on the mechanisms of novel therapeutic targets that can not only reverse chemotherapy resistance but also alleviate the pain caused by chemotherapy. In this respect, our study provided evidence to satisfy the unmet clinical need in cisplatin therapy. Here, we showed that the cisplatin resistance of tumor cells is closely linked to secretory autophagy-mediated EGFR hyperactivation via the transcriptional induction of TRPV1 by NANOG. Therefore, our study emphasized that targeting the NANOG-mediated refractory signal axis may overcome the obstacle that induce cisplatin resistance, thereby improving cisplatin efficacy.

Studies have reported that EGFR hyperactivation was associated with disease progression, metastatic potential, and the resistance to multi-modal therapies, including chemo-, radio-, and immunotherapy[50–52], and that autophagy-mediated EGF secretion was a crucial cause that provoked the hyperactivation of EGFR signaling[14]. Given that clinically available secretory autophagy pharmacologic inhibitors to overcome the chemoresistance of tumors have not yet been developed, it is necessary to define additional pathways that can regulate autophagy-mediated EGFR hyperactivation. Here, we found that the TRPV1 is overexpressed by NANOG in cisplatin-resistant tumor cells and mediates $Ca^{2+}$ influx and subsequent increases in autophagic EGF secretion, which activates the EGFR-AKT signaling pathway consequently contributing to cisplatin resistance. Importantly, the TRPV1 inhibition using its specific inhibitor AMG9810 renders the tumor vulnerable to cisplatin. Previous TRPV1 studies have repeatedly confirmed that neuronal TRPV1 activation aggravates pain and accordingly its inhibition is potentially among future therapeutic strategies for patients with neuropathic pain[30]. Thus, our results suggest that TRPV1 has profound value, as an actionable target that could simultaneously overcome the cisplatin resistance as well as CIPN in the clinic.

The regulatory mechanism of TRPV1, particularly in the course of acquiring cisplatin resistance, has not yet been extensively studied. In this study, we noted that the cause of TRPV1 overexpression in cisplatin-resistant cancer was due to direct transcriptional regulation by NANOG, but other possibilities may exist. NANOG also works together with other stemness factors, such as KLF4, MYC, OCT4, or SOX2, to control target genes that have important functions in embryonic stem cells and, plausibly, in tumor cells[27,53]. From this perspective, it is possible that TRPV1 expression by other stemness factors also contribute to tumor progression and therapeutic resistance. Additionally, TRPV1 was found to be aberrantly expressed by various factors in tumors, such as inflammation, tissue damage, and hypoxic conditions[33], suggesting that TRPV1 expression in tumor cells could be mediated by various tumor microenvironmental factors as well as several different stemness factors besides NANOG. Furthermore, we found that TRPV1 expression by itself was sufficient to induce cisplatin resistance in its channel activity-dependent manner. Notably, after *EGFR* knockdown, the cisplatin-resistant phenotypes of *TRPV1*-transduced P0 cell was almost entirely lost, highlighting that the EGFR pathway can function as a primary route through which TRPV1 promotes these phenotypes. Although it will be important in future studies to assess the precise underlying mechanisms by which TRPV1 regulates autophagy-dependent EGFR activation, this connection immediately hints at several potentially promising therapeutic targets to control cisplatin-resistant cancer in the clinic.

With regard to the cisplatin resistance induced by autophagy-dependent EGFR activation, this raises a question: Is EGFR alone capable of promoting these phenotypes? When EGFR binds to its ligand, such as EGF, it undergoes a conformational change that allows it to dimerize with other EGFR molecules or with other ErbB family members, such as ErbB2, ErbB3, or ErbB4. For example, the dimerization of EGFR with ErbB2 leads to the activation of both receptors, which results in initiation of intracellular signaling pathways that regulate cellular processes, such as proliferation, differentiation, and

survival[54,55]. Interestingly, Shiqi Ma et al. reported that EGFR/ErbB2 heterodimer can induce the resistance to tyrosine kinase inhibitors by preventing apoptosis[56]. Therefore, we believe that there is a possibility that TRPV1-mediated EGF secretion may induce cisplatin resistance through activation of other ErbB family members, and further exploration should be progressed to define the possibility in future studies.

Multi-drug resistance (MDR) is among the culprits of failure in chemotherapy since the cancer cells can efflux chemotherapeutic agents and, therefore, reduce intracellular drugs levels[57]. A well-established cause of MDR is related to the increased expression of members of the ATP-binding cassette (ABC) transporter superfamily[57]. Among various ABC transporters, ABCC2, ABCC5, and ABCC6 are known to be involved in cisplatin resistance[57]. Interestingly, results from previous studies indicated that hyperactivated EGFR signaling induces ABC transporter expression[58,59]. Here, we found that ABCC5 and ABCC6 was overexpressed in cisplatin-resistant CaSki CR cells compared to CaSki P cells and that ABCC5 was significantly regulated by TRPV1 (Supplementary Fig. 17). These results suggest that TRPV1 inhibition could restore the sensitivity to therapeutic agents by reversing the anti-apoptotic properties of tumor cells as well as by blocking the efflux of chemotherapeutic agents. Thus, targeting TRPV1 in chemotherapy-resistant cancer could be an actionable strategy in overcoming MDR.

Studies have demonstrated that autophagy plays a dual role in cancer by suppressing the growth of tumors or the progression of cancer development, which seems to be dependent on unknown characteristics of various cancer types[60]. The exact mechanisms underlying the paradoxical functions of autophagy in cancer cells are complex and not fully understood. Accumulating evidence has implicated that a combination of genetic and environmental factors determines the outcomes of autophagy activation in cancer cells[61]. Here, we unexpectedly found that cisplatin-resistant cancer cells exhibited an increase in autophagosome abundance which can accelerate autophagy flux. Interestingly, however, it also increased the secretion of EGF, which activated the EGFR-AKT signaling and, consequentially, the acquisition of cisplatin resistance. Notably, we found that the NANOG-TRPV1 axis could be a major molecular pathway inducing autophagosome abundance in cisplatin-resistant cancer cells. Although how paradoxical functions of autophagy can coexist in those cancer cells remain as a critical question to be answered in future studies, at least our results propose that the NANOG-TRPV1 axis could be used as one of molecular markers to predict the outcome of autophagosome abundance in cisplatin-resistant cancer cells.

Taken together, we propose that NANOG[+] tumor cells enriched by selective pressure imposed by cisplatin treatment preferentially expressed TRPV1 via transcriptional regulation. TRPV1 promoted anti-apoptotic phenotypes through the autophagy-mediated hyperactivation of EGFR signaling. Importantly, TRPV1 inhibition using the AMG9810, a potential pain-reliever, could sensitize resistant tumor to cisplatin by repressing anti-apoptotic properties. Thus, we provide proof-of-concept evidence that TRPV1 inhibition could be a clinically available strategy to control NANOG[high] cisplatin-resistant cancer and CIPN.

## Methods
### Ethics oversight
All mice were maintained and handled under the protocol approved by the Korea University Institutional Animal Care and Use Committee (KUIACUC) (KOREA-2021-0049). All animal procedures were performed in accordance with recommendations for the proper use and care of laboratory animals. Additionally, this study received approval from the Institutional Review Board of Gangnam Severance Hospital (Seoul, KOR), and informed consent was obtained from each patient after providing a detailed explanation of the study procedures,

including the risks and benefits of participating. Once the patient had a full understanding of the study, they provided written informed consent to participate. All procedures were conducted in accordance with the guidelines of the Declaration of Helsinki.

## Mice
Six- to eight-week-old female NOD-SCID mice were purchased from Central Lab Animal Inc. (Seoul, KOR).

## Cell lines
CaSki (ATCC, CRL-1550), HEK293 (ATCC, CRL-1573), HeLa (ATCC, CCL-2), H1299 (ATCC, CRL-5803), SiHa (KCLB, 30035), SNU719 (KCLB, 00719), AGS (KCLB, 21739), SNU668 (KCLB, 00668), and MKN28 (JCRB Cell Bank, 0253) cell lines were purchased from the American Type Culture Collection (ATCC, Manassas, VA, USA), the Korean Cell Line Bank (KCLB, Seoul, KOR) or the Japanese Collection of Research Bioresources Cell Bank (JCRB Cell Bank, Osaka, JPN). YCC2 were obtained from Cancer Metastasis Research Center (CMRC, Yonsei University College of Medicine, Seoul, KOR). All cell lines were obtained between 2010 and 2022 and tested for mycoplasma using the Mycoplasma Detection Kit (Thermo Fisher Scientific, San Jose, CA, USA). The identities of the cell line were confirmed by short tandem repeat (STR) profiling by IDEXX Laboratories Inc. and were used within six months for testing. The generation and maintenance of the cisplatin-resistant CaSki CR cell lines have been previously described[37]. Cisplatin-resistant SiHa cell line (SiHa CR) were generated from its parental cell line (SiHa P) by continuous exposure to cisplatin (Selleckchem, S1166) following initial dose-response studies of cisplatin (0.1–100 μM) over 72 h from that $IC_{50}$ values were acquired. Initially, SiHa P cells were treated with cisplatin ($IC_{50}$) for 72 h. The media was removed and cells were allowed to recover for a further 72 h. This development period was carried out for approximately 4 months, after which time $IC_{50}$ concentrations were re-assessed in SiHa CR cells. CR cells were maintained continuously in the presence of cisplatin at these new $IC_{50}$ concentrations. The CaSki NANOG or HEK293 NANOG cell lines have also been previously described[25]. All cells were grown at 37°C in a 5% $CO_2$ incubator with a humidified chamber.

## Chemical reagents
The following chemical reagents were used in this study: AMG9810 (Sigma-Aldrich, St. Louis, MO, USA, A2731), capsaicin (Sigma-Aldrich, M2028), gefitinib (Selleckchem, Houston, TX, USA, S1025), cisplatin (Selleckchem, S1166), spautin-1 (Selleckchem, S78880), and bafilomycin A1 (Sigma-Aldrich, B1793-2UG). The chemical reagents we used were dissolved in below solvents. AMG9810, gefitinib, spautin-1, or bafilomycin A1: Dimethyl sulfoxide (DMSO); Cisplatin: Dimethyl formamide (DMF); Capsaicin: ethanol.

## siRNA constructs
Synthetic small interfering RNAs (siRNAs) specific for GFP, EGFR, ATG7, NANOG, TRPV1, and CaMKKβ were purchased from Bioneer (Daejeon, KOR). The sequence were: non-specific GFP (green fluorescent protein), 5′-GCAUCAAGGUGAACUUCAA-3′ (sense), 5′-UUGAAGUUCAC CUUGAUGC-3′ (antisense); EGFR #1, 5′-GAUCCACAGGAACUGGAUA-3′ (sense), 5′-UAUCCAGUUCCUGUGGAUC-3′ (antisense); EGFR #2, 5′-UU AGAUAAGACUGCUAAGGCAUAGG-3′ (sense), 5′-CCUAUGCCUUAGCA GUCUUAUCUAA-3′ (antisense); ATG7 #1, 5′-CAGCUAUUGGAACACU-GUA-3′ (sense), 5′-UACAGUGUUCCAAUAGCUG-3′ (antisense); ATG7 #2, 5′-CAGCUAUUGGAACACUGUA-3′ (sense), 5′-UACAGUGUUCCAA UAGCUG-3′ (antisense); NANOG #1, 5′-GCAACCAGACCUGGAACAA-3′ (sense), 5′-UUGUUCCAGGUCUGGUUGC-3′ (antisense); NANOG #2, 5′-C UAAACUACUCCAUGAACA-3′ (sense), 5′-UGUUCAUGGAGUAGUUU AG-3′ (antisense); TRPV1 #1, 5′-GGAGACCUGUCUGCUGAAAUU-3′ (sense), 5′-AAUUUCAGCAGACAGGUCUUC-3′ (antisense); TRPV1 #2, 5′-GGAGUUCACCGAGAACUAU-3′ (sense), 5′-AUAGUUCUCGGUGAACU

CC-3′ (antisense); CaMKKβ #1, 5′-GUGAAGACCAUGAUACGUA-3′ (sense), 5′-UACGUAUCAUGGUCUUCAC-3′ (antisense); and CaMKKβ #2, 5′-GACCAUCUGUACAUGGUGU-3′, and 5′-ACACCAUGUACAGAU GGUC-3′ (antisense). For in vitro delivery, the cells were transfected with 100 pmol of synthesized siRNAs using Lipofectamine 2000 (Invitrogen, San Jose, CA, USA, 11668027) according to the manufacturer's instructions.

## DNA constructs
The pMSCV-FLAG-NANOG WT and pMSCV-FLAG-NANOG MT plasmids have been previously described[20]. Briefly, to generate pMSCV-NANOG, cDNA encoding human NANOG was amplified from pSIN-EF2-NANOG-Pur (Addgene, Watertown, MA, USA, 16578). The amplified cDNA was cloned into XhoI/EcoRI restriction sites of the pMSCV retroviral vector (Clontech, Mountain View, CA, USA). To generate pMSCV-NANOG MT, the QuikChange XL Site-Directed Mutagenesis Kit (Stratagene, San Diego, CA, USA, 200516) was used according to the manufacturer's instructions. The pcDNA 3.1-TRPV1 plasmids were purchased from Cosmogenetech (Seoul, KOR).

## Western blot analysis
Lysate extracted from a total of $1 \times 10^5$ cells was used to perform Western blots analysis. Primary antibodies against pEGFR (1:3000; Cell Signaling Technology, Danvers, MA, USA, 3777 S), EGFR (1:3000; Cell Signaling Technology, 4267 S), MCL1 (1:3000; Santa Cruz Biotechnology, Paso Robles, CA, USA, sc-819), ATG7 (1:3000; Cell Signaling Technology, 8558 S), LC3B (1:3000; Cell Signaling Technology, 2775 S), EGF (1:1000; Abcam, Cambridge, UK, ab206106), NANOG (1:3000; Bethyl Laboratories, Montgomery, TX, USA, A300-379A), TRPV1 (1:3000; Abcam, ab6166), FLAG (1:5000; Medical & Biological Laboratories, Nagoya, JPN, M185-3L), pAKT (1:3000; Cell Signaling Technology, 9271), AKT1 (1:3000; Cell Signaling Technology, 9272), and β-actin (1:5000; Medical & Biological Laboratories, M177-3) were used. Western blotting was followed by incubation with the appropriate secondary antibodies conjugated to horseradish peroxidase (HRP), anti-rabbit IgG-HRP (1:5000; Enzo, Farmingdale, NY, USA, ADI-SAB-300-J), and anti-mouse IgG-HRP (1:5000; Enzo, ADI-SAB-100-J). The immunoreactive bands were developed with the chemiluminescence ECL Detection System (GE Healthcare, Chicago, IL, USA), and signals were detected using a luminescent image analyzer (LAS-4000 Mini, Fujifilm, JPN).

## Real-time quantitative RT-PCR
Total RNA from the cells was purified using RNeasy Micro Kit (Qiagen, Valencia, CA, USA, 74004) and cDNA was synthesized by reverse transcriptase (RT) using an iScript cDNA synthesis kit (Bio-Rad, Hercules, CA, USA, 1708891) according to the manufacturer's recommended protocol. Real-time PCR was performed using IQ SYBR Green Super mix (Bio-Rad, 1708880) with the specific primers on a CFX96 real-time PCR detection system. Fold-change was calculated relative to the expression level of mRNA in the control cells. qPCR primers were purchased from Bioneer (Daejeon, KOR): TRPV1, 5′-GGCTTGCC TCCCTTTAAGATG-3′ (forward), 5′-CTGTCCACAAACAGGGTCTTC-3′ (reverse) and β-actin, 5′-CGACAGGATGCAGAAGGAG-3′ (forward), 5′-T AGAAGCATTTGCGGTGGAC-3′ (reverse). All real-time quantitative PCR experiments were performed triplicate and quantification cycle (Cq) values were determined using Bio-Rad CFX96 Manager 3.0 software (Bio-Rad, Hercules, CA, USA). Relative quantification of the mRNA levels was performed using the comparative Ct method with β-actin as the reference gene.

## Quantitative ChIP (qChIP) assays
The ChIP kit (Millipore, Burlington, MA, USA, 17-295) was employed according to the manufacturer's instructions and the ChIP assay was performed as described previously[25]. Briefly, cells ($1 \times 10^7$ per assay)

were bathed in 1% formaldehyde at 25°C for 10 min for cross-linking of proteins and DNA and then lysed in sodium dodecyl sulfate buffer containing protease inhibitors. DNA was sheared by sonication using a Sonic Dismembrator Model 500 (Fisher Scientific, Pittsburgh, PA, USA). Immunoprecipitation was carried out by incubating with 1 µg of anti-FLAG (Medical & Biological Laboratories, M185-3L), anti-NANOG (Bethyl Laboratories, A300-379A) antibodies or rabbit IgG (Millipore, Billerica, MA, USA, PP64) for 16 h and then the immunoprecipitated DNA was quantified by real-time qPCR using the following primer set: 5′-GCAGGACTTGAGAAGCCTCTC-3′ (forward) and 5′-GACGCAGCTG-CACACATG-3′ (reverse). Each sample was assayed in triplicate, and the amount of precipitated DNA was calculated as the percentage of the input sample.

## Immunofluorescence
Cells grown on four-well cell culture slides (SPL life science, Pochon, Kyonggi-do, KOR, 30404) were fixed with 4% paraformaldehyde (Wako, Richmond, VA, USA, 30525-89-4) in phosphate-buffered saline for 20 min, and then blocked with 5% normal goat serum (NGS, Thermo Fisher Scientific, 50062Z) in PBS for 1 h. The cells were incubated with primary antibodies (1:1000–2000 dilution) and appropriate secondary antibodies conjugated to Alexa Fluor 555 (1:1000; Invitrogen, A21429). Confocal fluorescence images were obtained using a Zeiss LSM700 confocal microscope (Carl Zeiss, Oberkochen, DEU). Image analysis was performed using ImageJ[14].

## Electron microscopy analysis
Cells were incubated in medium supplemented with 0.1% FBS for 48 h. The cells were washed with 0.1 M PBS (pH 7.0) and fixed with a buffer containing 2% paraformaldehyde and 2.5% glutaraldehyde (Electron microscopy sciences, Hatfield, PA, USA, 16310) in PBS at room temperature for 30 min. After washing with PBS, cells were post-fixed with 1% osmium tetroxide (Heraeus, Hanau, DEU, 89.740.219) for 90 min, dehydrated and then embedded with Epoxy resin mixture (Electron microscopy sciences, 14120). Thin sections of 70 nm were collected on 200-mesh nickel grid (Electron microscopy sciences, EMS200-Ni) and stained with uranyl acetate (Electron microscopy sciences, 22400) followed by lead citrate. Electron microscopy observation for general morphology was performed using a Hitachi H-7650 electron microscope (Hitachi High-Technology Co. Tokyo, JPN) with 80 kV acceleration voltage[14].

## Apoptotic cell death analysis
Tumor cells were treated with indicated agents in medium supplemented with 0.1% fetal bovine serum (FBS) for 24 h at 37°C. The cells were stained for active caspase 3 (1:500; BD Biosciences, Franklin Lakes, NJ, USA, 560626) as an indicator of apoptosis and examined by flow cytometry as shown gating strategy in Supplementary Fig. 18.

## EGF assessment of medium supernatants
The cells were grown in six-well plates and incubated with medium containing 0.1% FBS for 48 h at 37°C. The supernatants were collected and centrifuged to remove cell debris. The EGF levels in the supernatants were measured by the Quantikine Human EGF ELISA kit (R&D Systems, Minneapolis, MN, USA, DEG00). For western blot analysis, the supernatants were further concentrated 30× by Centricon Plus-70 centrifugal Filter Units-3kDa (Millipore, UFC800324).

## Electrophysiology
Whole-cell currents were measured using a MultiClamp 700B Microelectrode Amplifier (Molecular Devices, San Jose, CA, USA) at a holding potential of −60 mV at room temperature. Recording pipets were pulled from borosilicate glass to a final resistance of 2–4 MΩ and their tips were fire-polished. The cells with a seal resistance of over 3–10 GΩ were used for recording. After establishing a whole-cell configuration,

currents were recorded by applying 200 ms voltage-ramp pulses from −80 to +80 mV every 750 ms. The currents were digitized with a Digidata 1440 A Converter (Molecular Devices), filtered at 5 kHz, and analyzed using Clampfit 10.5 (Molecular Devices). The internal pipette solution contained 140 mM CsCl, 5 mM EGTA, and 10 mM HEPES (titrated to pH 7.2 with CsOH). The extracellular bath solution contained 140 mM NaCl, 5 mM KCl, 2 mM $CaCl_2$, 1 mM $MgCl_2$, 10 mM glucose and 10 mM HEPES (titrated to pH 7.3 with NaOH).

## Intracellular calcium imaging
Fluorescence intracellular calcium imaging was performed as described previously[62]. Briefly, glass coverslips (35 mm diameter) were coated with fibronectin human protein (Thermo Fisher Scientific) and then cells were seeded on it. The cells were incubated in a 5% $CO_2$ incubator at 37°C. After 24 h, 4 µM Fura-2 AM (Thermo Fisher Scientific, F1221) and 0.02% Pluronic F-127 (Thermo Fisher Scientific, P3000MP) were mixed in Opti-MEM medium (Gibco) and loaded in cells at 37°C. After 45 min, the coverslips were put in the recording bath chamber and intracellular fluorescence was measured at ratiometric wavelengths of 340/380 nm using a cooled charge-coupled device (CCD) camera (Retiga-SRV) and Metafluo software (Molecular Devices).

## Immunohistochemistry
Tissue microarray (TMA) were produced from formalin-fixed, paraffin-embedded tumor specimens, and matched nonadjacent normal specimens. A total of 77 surgically resected primary cervical cancer patients (53 chemoradiosensitive and 24 chemoradioresistant) were analyzed in this study. The patient's clinicopathological characteristics are summarized in Supplementary Table 1. Some of the paraffin blocks for the chemoradioresistant patients were provided by the Korea Gynecologic Cancer Bank through the Bio & Medical Technology Development Program of the Ministry of Education, Science and Technology, Korea (NRF-2017M3A9B8069610). Tissue samples and medical records were obtained with approval by the Institutional Review Board of Gangnam Severance Hospital (Seoul, KOR). Cancer patients who had a recurrence within one year after chemoradiotherapy were considered to have a "resistant response". Immunohistochemistry was performed on 5-µm sections of the TMA using a standard streptavidin-peroxidase method as described previously[37]. After deparaffinization and rehydration, heat-induced antigen retrieval was performed for 20 min using antigen retrieval at pH 6.0 (Dako, Carpinteria, CA, USA) in a pressure cooker. Endogenous peroxidase activity was quenched with 3% $H_2O_2$ for 10 min. The sections were incubated with rabbit polyclonal anti-TRPV1 antibodies (Alomone, Jerusalem, ISR, ACC-030) at 1:1000 dilution for 60 min. The antigen-antibody reaction was detected with EnVision⁺ Rb HRP-linked secondary antibody (Dako) and visualized with 3,3-diaminobenzidine (Dako), followed by hematoxylin counterstaining. Negative controls, which included immunoglobulin G (IgG) but no primary antibody was concurrently performed, and the TMA included the appropriate positive control tissues. NANOG and pEGFR protein expression were previously evaluated in the same cohort[14]. Sixty-four (83.1%, NANOG), 62 (80.5%, TRPV1), and 65 (84.4%, pEGFR) out of 77 cases were suitable for immunohistochemical (IHC) interpretation. For digital image analysis, all stained slides were scanned using the NanoZoomer 2.0 HT (Hamamatsu Photonics K.K., Japan) at ×40 objective magnification (0.23 µm per pixel resolution). Immunohistochemistry scoring was performed using Visiopharm Digital Image Analysis (DIA) software v6.2.0.2089 (Visiopharm, Hørsholm, DNK) as described previously[20]. Tumor nuclei were defined after training the system. Cytoplasm was further defined by outlining the defined nucleus. DAB intensity was obtained by using a predefined algorithm ranging from 0 (no signal) to 300 (fully saturated signal). The cutoff value for NANOG, TRPV1, and pEGFR was 160, 222, and 172, respectively.

## Tumor treatment experiments

NOD/SCID mice were inoculated subcutaneously with $1 \times 10^6$ CaSki CR cells per mouse. Nine days following the tumor challenge, AMG9810 (0.05 mg/kg)-loaded chitosan hydrogel or cisplatin (2.5 mg/kg) was administered intratumorally or intraperitoneally, respectively. The cisplatin is prepared in DMF, diluted in phosphate-buffered saline (PBS). This treatment regimen was repeated for two cycles. The mice were monitored for tumor burden and survival for 19 and 40 d after the challenge, respectively. Tumor size was measured before the tumors were smaller than, or at about 10% of mice body weight, the maximal tumor size/burden permitted by KUIACUC (Tumor size do not exceed 20 mm at the largest diameter). In some cases, this limit has been reached on the last day of tumor size measurement and the mice were immediately euthanized.

## TCGA data collection and analysis

Gene expression data for CESC, LUAD, LUSC, STAD, and STES samples profiled by TCGA were collected from the Firehose BROAD GDAC data repository (https://gdac.broadinstitute.org/). Clinical data were also retrieved from the same source. To determine the clinical relevance of interest gene signature, we analyzed the transcriptome data from CESC classified as responders (R) or non-responders (NR) to cisplatin. The patient's information is summarized in Supplementary Data 1. The EGFR activity score and chemotherapy-resistant gene expression signature were defined previously[35,36]. We used Seurat (v4.0.6)'s 'AddModuleScore' function to calculated EGFR activity score. This method measures the average expression of EGFR pathway genes, subtracted by the average expression of randomly chosen control genes. In addition, we used the single-sample Gene Set Enrichment analysis algorithm implemented in R package GSVA to calculate the autophagy, or chemotherapy-resistant gene expression signature for each sample. The default parameters from the GSVA package were used. The association between EGFR activity score and survival was evaluated by Cox regression and Kaplan–Meier analyses. The 30th and 70th percentiles were used as cutoff thresholds.

## Statistical analysis

All data are representative of at least three separate experiments. Statistical differences were calculated by the Student's $t$ test (two-tailed, unpaired), one-way ANOVA, or two-way ANOVA using GraphPad Prism 7 software (GraphPad Software, Inc). For tissue samples and IHC, statistical tests were performed using IBM SPSS statistics version 21.0 (IBM Corporation). The Kruskal-Wallis or Mann–Whitney $U$ test and $\chi^2$-test were used to compare the protein expression between each group. Survival curves were calculated using the Kaplan–Meier method and the differences between the survival curves were calculated by the Gehan–Breslow-Wilcoxon test. The Cox proportional hazards model was created to identify the independent predictors of survival. Results with two-tailed $p$ values of <0.05 were considered statistically significant.

## Reporting summary

Further information on research design is available in the Nature Portfolio Reporting Summary linked to this article.

## Data availability

The publicly available RNA sequencing and clinical data used in this study are publicly available in the Firehose BROAD GDAC data repository (https://gdac.broadinstitute.org/). The gating strategy is provided in Supplementary Fig. 18. All raw images for the immunoblots are provided in Supplementary Fig. 19. The remaining data are available within the Article, Supplementary Information or Source Data file. Source data are provided with this paper.

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

## Acknowledgements

This work was funded by the National Research Foundation of Korea (NRF-2021R1C1C2008422, to S.J. Oh; NRF-2020R1A2B5B03095410, to T.W. Kim; and NRF-2022R1A4A2000827, to T.W. Kim).

## Author contributions

Study concept and design: S.J.O. and T.W.K.; acquisition of data: S.J.O., J.Y.L., M.K.S., J.H.A., H.-J.L., Y.J.P., J.M.C., and E.H.C.; analysis and interpretation of the data: S.J.O., S.K.H., S.W.H., and T.W.K.; technical or other material support: J.-Y.C., H.B.C., H.S.K., and J.-H.K.; writing and review of the manuscript: S.J.O., J.Y.L., J.-Y.C., S.W.H., and T.W.K.

## Competing interests

The authors declare no competing interest.
