## [Peer Review File · Nature Communications]

Reviewers' Comments:

Reviewer #1:

Remarks to the Author:

The paper by Se Jin Oh and collaborators describes the role of an autophagy-mediated hyperactivation of the EGFR pathway in the regulation of cisplatin resistance through a transcriptional increase of the pain-generating channel TRPV1, mediated by NANOG.

The paper provides novel and intriguing information on a mechanism that could regulate at the same time resistance to cisplatin and chemotherapy-induced neuropathic pain (CIPN), thus representing a target for therapeutic intervention in cancer patients who receive cisplatin-containing chemotherapy.

However, there are several points that the authors need to address in order to improve their manuscript:

- 1) The paper must be carefully revised to eliminate typos and grammatical errors that I have found in several paragraphs
- 2) The majority of the experiments were performed on a single cell model of resistance to cisplatin (CaSki cells and their derivative cisplatin-resistant P3 clone). Data on additional cell lines/resistance models could reinforce the significance of the observations described.
- 3) Although the difference in EGFR activity score between low and high cisplatin resistant patients is statistically significant (fig 1b), there is a huge overlap in the values of the two cohorts. This suggests that the EGFR pathway might be hyperactivated only in a fraction of resistant tumors. This is true also per the autophagy signatures in figure 2a. This finding should be acknowledged by the authors in the results and in the discussion of the data. In addition, more details should be provided on the TCGA cohorts analyzed. Are patients in early or advanced stage of disease? Did they receive the same therapeutic regimen?
- 4) In figure 1c the quantification of the bands is not clear, all the bands are reported with a value of 1.0
- 5) In figure 1e and 1g the concentration of cisplatin is not shown. In addition, the method used for the measurement of apoptotic cell death should be reported in the figure legend.
- 6) The authors state that both siEGFR 1 and 2 modulated cisplatin resistance in P3 cells. However, data on siEGFR2 are not shown/described in the paper nor in the supplementary data.
- 7) The authors found that the increase in EGFR activation was due to increased autophagy-mediated release of EGF. In this respect, the quantification of EGF in the medium from P3 cells (fig 2e) should be performed by using a quantitative method such as ELISA or similar approaches. Even more importantly, the increase in EGFR activation could be due to the increased secretion of a number of different EGFR ligands that are frequently expressed in cancer cells. Therefore, the levels of secretion of other EGF-like peptides should also be assessed.
- 8) Similarly, the effects of NANOG silencing on different EGF-like peptides in addition to EGF should be assessed by using a quantitative approach (fig 3c).
- 9) The authors should discuss why they focused on TRPV1 among the 21 upregulated genes and briefly discuss the possible role of other genes not investigated in details in this paper.
- 10) TRPV1 levels in cisplatin resistant cells should also be assessed with western blotting (fig 4b)
- 11) The TRPV1 inhibitor AMG9810 has been used at quite high concentration in different experiments (10 micromolar). The authors should show the effects of lower doses of AMG on the activation of EGFR and secretion of EGF and other EGF-like peptides, in order to demonstrate a specific dose-dependent effect of the drug.
- 12) The effects of MUT NANOG on TRPV1 were more significant in HEK293 as compared with CaSki cells. Is this due to the higher basal levels of TRPV1? This finding needs to be discussed.
- 13) The cohort of cervical cancer patients analyzed is too small to drive any firm conclusion (fig 6). In particular, when combining the three biomarkers (NANOG/TRPV1/EGFR), a subgroup includes only 9 patients. The authors need to significantly increase the number of patients if they want to support their hypothesis with clinical findings. In addition, the authors must acknowledge that increased survival observed in this retrospective analysis could be due to a prognostic and not predictive value of the investigated biomarkers. In fact, the predictive value of a biomarker can be only assessed within a randomized clinical trial. Finally, more information on the clinical and pathologic features of these patients should be provided at least in the supplementary material.
- 14) The authors claim that the mechanism they identified could involve different types of cancer. However, only one gastric cancer cell line was found to overexpress NANOG (supplementary figure 2), whereas no data are shown for other cancer cell lines. The authors should at least acknowledge

that expression of NANOG is heterogeneous in these cancer types. In addition, additional information on the cohorts of patients included in this analyses should be provided (early vs advanced stage, therapeutic regimens etc).

15) The authors claim that TRPV1 inhibition and cisplatin treatment might have a synergistic effect. However, in order to demonstrate synergism, the authors need to perform in vitro and/or in vivo experiments with combinations of the two drugs at different concentrations, according to specific models evaluating the interaction between different drugs.

16) Finally, the authors conclude that TRPV1 inhibition is a clinically available strategy to target cisplatin resistance. However, the clinical utility of this approach needs to be addressed in prospective clinical trials before making any conclusions.

Reviewer #2:

Remarks to the Author:

The reasonably well written article presents results to link hyperactivation of EGFR, ion channel TRPV1, NANOG, autophagy-mediated EGF secretion via Ca²⁺ influx (the NANOG-TRPV1-pEGFR axis) in cisplatin resistance. Although abundant data is presented, the support for the somewhat complicated pathways is weak. The proposed links seem to be random rather than systematic, and likely not to be the main routes or mechanisms. The data presented are not of high quality or clarity. Although the experiments and rationales are based on previous publications by the authors, the current manuscript does not present a solid concept.

In breaking down the ideas:

1. Hyperactivated EGFR confers resistance to cisplatin in cervical cancer: other mechanisms are likely, not limiting to EGFR hyperactivation.
2. Autophagosome mediates EGF secretion: the data is not sufficient and the conclusion seems not convincing.
3. NANOG activates TRPV1, which then stimulates AMPK-Ca²⁺, and then Autophagy to promote EGF secretion: the pathway seems random and unusual, and the support from the results is not convincing.

The description of experiments/figures are overly brief in places preventing proper evaluation:

For examples:

Fig. 1: (b) CESC cohort: no information? What is it? (c) p-EGFR P0 P3, 1.0, 1.0 ? (e) The frequency of apoptotic cells ?

Many results present values of little differences?

For an example:

Fig. 1b: the values of the two groups are ;largely overlapping.

Reviewer #3:

Remarks to the Author:

The authors have shown that cisplatin resistance is highly associated with hyperactivation of EGFR on tumor cells and that EGFR hyperactivation rises by a transcriptional increase in TRPV1 via NANOG. In addition, the authors have shown that TRPV1 promoted autophagy-mediated EGF secretion with calcium influx, which activated EGFR/AKT signaling pathways. Furthermore, the authors have shown that TRPV1 inhibition using AMG9810 increased the effectiveness of cisplatin treatment.

I enjoyed reading the rationale and the introductory part of the manuscript. If there is a way to inhibit tumor growth and, at the same time, alleviate the chemotherapy-induced neuropathic pain, that will help the patients, as that is one of the major limitations of cisplatin treatment.

However, there are several places in the manuscript that need further clarification to improve the paper's overall significance.

1. Define secretory autophagy and degradative autophagy. It was unclear what the difference was and how it promoted EGF secretion by NANOG. The papers have many references to previous work, but it is not clear how it ties all together to the context of this paper.
2. How are low and high chemotherapy-resistant tumors defined? What is the cut-off mark? The gene signatures were generated from the breast cancer model. Are the same gene signatures apply to cervical cancer?
3. Please elaborate on why LC3B-I and LC3B-II are tested and how they are different. In some setting, the upper band (LC3B-I) is used to come up with a conclusion, but in some settings, it is the lower band (LC3B-II) that is used (i.e. Fig 5c).
4. The authors used a sub-section titled "TRPV1 promotes NANOG-mediated resistance to cisplatin through the Ca²⁺-AMPK pathway"; however, they have not tested any effect of TRPV1 overexpression in sensitive cells. It is not the resistance, but the sensitivity is being tested here.
5. The authors concluded that TRPV1 inhibition could reverse the cisplatin resistance (Figure 7). However, no experiment was done to directly prove that. To test reversal, the tumors have to form first, and then treat the tumors with TRPV1 inhibitors to reverse, meaning to shrink the tumor. What was actually done is the prevention of tumor progression, not the reversal of cisplatin resistance. The title is, therefore, misleading.

Minor comments:

1. Figure 1e. What is the concentration of cisplatin?
2. Please indicate in the Methods section how the cisplatin was dissolves. Depending on the solvent, the efficacy of cisplatin changes and will be beneficial to the readers.
3. Figure 1g. Data for siEGFR #2 is missing.
4. Figure 4. The authors indicate that "NANOG MT had no profound impact on either TRPV1 mRNA or protein levels", but it looks significantly downregulated. I do not see bands for TRPV1 in NANOG MT in HEK293 cells, but the value says 1.0. I think there is an error here.
5. Figure 5g. Please correct the wrong values.
6. Figure 6. The sentence "The high expression of NANOG and TRPV1 was correlated with chemoradiation resistant tumors, whereas the high expression of pEGFR tended to correlate with chemoradiation resistance" is confusing. Although Fig 6b shows that it is not significant, the immunohistochemical staining looks significant. Could you please clarify the discrepancies in the conclusion?
7. Supplementary Figure 5 is confusing. ABCC5 is usually upregulated in cisplatin-resistant cells but it was downregulated in P3. What was the rationale for silencing TRPV1 in P3 cells that has low ABCC5 and ABCC6 transporters?

Reviewer #4:

Remarks to the Author:

Oh et al have submitted a manuscript entitled "TRPV1 inhibition reverses cisplatin resistance by blocking autophagy-mediated hyperactivation of EGFR signal pathway" for consideration of publication. This study describes as novel mechanism where EGFR is activated through a autophagy mediated pathway leading to cisplatin resistance in cervical cancers. Even though the evidence in the cisplatin resistant cell line is convincing, there is a lack of subsequent results supporting this mechanism in cervical cancer. Thus, there is not enough results presented to warrant publication. List below are my specific comments.

1. The generation of a cisplatin resistant cell line (P3) gives evidence that EGFR activation and Mcl-1 levels are increased. This needs to be demonstrated in at least two independent cervical cancer cells. It might just be a selection for cisplatin resistant specific to only this cell line.
2. In figure 1, Gefitinib was used to show increased cell death in P3 cells. As a control, P0 cells needs to be shown.
3. In figure 2, the chemotherapy resistant signature is not convincing. The variability is very high and expression at the gene level often does not correlated with protein level. More evidence needs to be presented that cisplatin resistance patients have altered autophagy function.
4. Fig 2 B shows changes in LC3B punctation was demonstrated using anti-LC3B antibodies. This was not very convincing as the punctation is very weak in P3. Transfection of GFP-LC3B would be a better method to validate whether autophagy is altered in P3 cells.
5. In figure 2 E, EGF was shown in the medium which was reduced by knocking out ATG7. There

needs to be demonstrated that EGF release by cells lead to the hyperactivation of EGFR in P3 cells. This could be accomplished in several different ways such as immunodepleting EGF in the media or knocking down EGF in the cells.

6. Whenever demonstrating difference in LC3-II levels, autophagy flux needs to be determined. This requires the use of an autophagy inhibitor that prevents the formation of autolysosomes such as chloroquine.

7. In figure 3, the knockdown of NANOG reduces EGF in the media and increases cisplatin induced cell death in P3 cells. To demonstrate this is due to EGFR signaling, EGF should be added to the media and changes in cell death measured. In P0 cells overexpressing NANOG, EGF should be knocked down and effects on cisplatin induced cell death determined.

8. It would be important to use AMG9810 in combination with EGFR inhibitors to demonstrate whether these treatments would be synergistic?

9. Besides P3 cells, it would be important to show that upregulation of NANOG in another cell line sensitizes these tumors to combined cisplatin and AMG9810.

10. It is unclear whether TRPV1 leads to autophagy mediated secretion of EGF and how this will be different than blocking autophagy mediated cell survival function by providing cells with nutrients such as amino acids.

Dear Reviewer,

We appreciate the detailed and invaluable comments that have been raised in response to our manuscript (NCOMMS-21-51679) entitled "**TRPV1 inhibition overcomes cisplatin resistance by blocking autophagy-mediated hyperactivation of EGFR signaling pathway**". We appreciate the detailed and relevant comments from the reviewers. Your assistance has been invaluable and has helped us improve the quality of our manuscript. We have fully addressed each reviewer's questions and comments and amended the manuscript content accordingly. We have also provided point-by-point responses to the critiques raised by the reviewers. All the reviewer questions are in **bold and italic text** while our responses are in regular or bold text (if necessary). In the revised manuscript, the changes are marked in **RED**. Once again, we appreciate you for your time, consideration, and invaluable guidance.

Sincerely,

Tae Woo Kim, PhD
Professor, Department of Biomedical Science
Korea University College of Medicine
Phone: +82-2-2286-1301
Fax: +82-2-923-0480
E-mail: twkim0421@korea.ac.kr

Responses to reviewers:

Reviewer #1

The paper by Se Jin Oh and collaborators describes the role of an autophagy-mediated hyperactivation of the EGFR pathway in the regulation of cisplatin resistance through a transcriptional increase of the pain-generating channel TRPV1, mediated by NANOG.

The paper provides novel and intriguing information on a mechanism that could regulate at the same time resistance to cisplatin and chemotherapy-induced neuropathic pain (CIPN), thus representing a target for therapeutic intervention in cancer patients who receive cisplatin-containing chemotherapy. However, there are several points that the authors need to address in order to improve their manuscript.

1) *The paper must be carefully revised to eliminate typos and grammatical errors that I have found in several paragraphs.*

We thank the reviewer's this critical comment and apologize for our mistakes. We have corrected the typo's and grammatical errors and marked in red in the revised manuscript.

2) *The majority of the experiments were performed on a single cell model of resistance to cisplatin (CaSki cells and their derivative cisplatin-resistant P3 clone). Data on additional cell lines/resistance model could reinforce the significance of the observation described.*

We agree with the reviewer's opinion that our findings in this study need to be reproduced in additional cisplatin-resistant tumor model. To do this, **we performed additional experiments using two different types of cisplatin-resistant cervical cancer cell lines: SiHa CR and HeLa.** SiHa CR is newly established cisplatin-resistant tumor cells line, generated from SiHa P (a cisplatin-susceptible parental cell line of SiHa), and HeLa is commonly used in multiple drug resistance studies (Pan C et al., Nat Commun, 2021). Consistent with the results from CaSki CR cells, SiHa CR and HeLa displayed elevated autophagosome-mediated EGF secretion and EGFR signaling-dependent cisplatin resistance compared to control cells (Supplementary Fig. 2, 6 of the revised manuscript).

Importantly, we found that the NANOG-TRPV1-pEGFR axis was conserved, and TRPV1 inhibition successfully overcame the NANOG-mediated cisplatin resistance **in other cisplatin-resistant cervical cancer models we tested for revision** (Supplementary Fig. 7, 8 of the revised manuscript). Thus, these data support our conclusion that the NANOG-TRPV1-pEGFR axis could function as a central molecular pathway that confers resistance of tumor cells to cisplatin.

We appreciate the invaluable and helpful comment that improves the quality of our manuscript. We have included additional data and description in the revised manuscript as followed:

[On supplementary Fig. 2 of the revised manuscript]

[On page 5, 6, Supplementary Figure legend section of the revised manuscript]

Supplementary Fig. 2. The phenotypes of cisplatin-resistant tumor cells are critically dependent on EGFR signaling activation. **a** Flow cytometry analysis of the frequency of apoptotic (active caspase 3⁺) cells after treatment with or without cisplatin for 24 h. **b** The protein level of pEGFR, EGFR, and MCL1 expression were confirmed by western blots. **c** and **d** SiHa CR and HeLa cells were treated with gefitinib as the indicated dose. **e** and **f** SiHa CR and HeLa cells were transfected with siRNA targeting *GFP* or *EGFR*. **c** and **e** The protein levels of pEGFR, EGFR and MCL1 were determined by western blots. β-actin was used as an internal loading control. Numbers below the blot images indicate

the expression as measured by fold-change. **d** and **f** Flow cytometry analysis of the frequency of apoptotic (active caspase 3⁺) cells after incubation with or without cisplatin for 24 h. All experiments were performed in triplicate. The p-values by two-way ANOVA are indicated. The data represent the mean \pm SD. Source data are provided as a Source Data file. (*P \leq 0.05, **P \leq 0.01, ***P \leq 0.001. NS, not significant)

[On page 8, Results section of the revised manuscript]

To verify the reproducibility of the above results, we repeated experiments of Fig. 1c to 1g using two different types of cisplatin-resistant cervical cancer cell lines: SiHa CR and HeLa. SiHa CR is another newly established cisplatin-resistant tumor cell line, generated from SiHa P (a cisplatin-susceptible parental cell line of SiHa), and HeLa is commonly used in various drug resistance studies⁴². Consistent with the results from CaSki CR cells, SiHa CR displayed the resistance to cell death by cisplatin and the hyperactivated EGFR signaling compared to SiHa P cells (Supplementary Fig. 2a, b). Importantly, EGFR inhibition using gefitinib or *siEGFR* could reverse the cisplatin-resistant phenotypes of SiHa CR cells and HeLa cells (Supplementary Fig. 2c-f). Taken together, our data demonstrate that the hyperactivation of EGFR signaling pathway contributes to the resistance to cisplatin in cervical cancer cells.

[On supplementary Fig. 6 of the revised manuscript]

[On page 10, Supplementary Figure legend section of the revised manuscript]

Supplementary Fig. 6. Autophagosome formation is crucial for EGF secretion and EGFR signaling-mediated cisplatin resistance in SiHa CR cells. **a** In the cells treated with or without spautin-1, the protein levels of secreted EGF and internal LC3B, pEGFR, EGFR, and β -actin were

determined by western blots. β -actin was used as an internal loading control. Numbers below the blot images indicate the expression as measured by fold-change. **b** Flow cytometry analysis of the frequency of apoptotic (active caspase 3⁺) cells after incubation with or without cisplatin for 24 h. All experiments were performed in triplicate. The p-values by two-way ANOVA are indicated. The data represent the mean \pm SD. Source data are provided as a Source Data file. (* $P \leq 0.05$, ** $P \leq 0.01$, *** $P \leq 0.001$. NS, not significant)

[On page 10, Results section of the revised manuscript]

Colinearly, compared to SiHa P cells, SiHa CR cells also showed increase in autophagosome formation and EGF secretion (Supplementary Fig. 6a). In addition, the suppression of autophagosome formation by spautin-1 reduced EGF secretion, the level of pEGFR and cisplatin-resistant phenotypes in SiHa CR cells (Supplementary Fig. 6a, b). Given these data, we conclude that autophagosome-mediated EGF secretion induces the hyperactivation of EGFR signaling, thereby promoting chemoresistance to cell death by cisplatin in cervical cancer cells.

[On supplementary Fig. 7 of the revised manuscript]

[On page 11, Supplementary Figure legend section of the revised manuscript]

Supplementary Fig. 7. Knockdown of NANOG reduces cisplatin resistance by blocking autophagosome-mediated EGF secretion. **a** The protein levels of NANOG were determined by western blot. **b** and **c** SiHa CR or HeLa cells were transfected with siRNA targeting GFP or NANOG. **b** The levels of internal NANOG, LC3B, pEGFR, and EGFR and secreted EGF protein were confirmed by

western blots. **c** Flow cytometry analysis of the frequency of apoptotic (active caspase 3⁺) cells after incubation with or without cisplatin for 24 h. **a** and **b** β -actin was used as an internal loading control. Numbers below the blot images indicate the expression as measured by fold-change. All experiments were performed in triplicate. The p-values by two-way ANOVA are indicated. The data represent the mean \pm SD. Source data are provided as a Source Data file. (*P \leq 0.05, **P \leq 0.01, ***P \leq 0.001. NS, not significant)

[On page 10, 11, Results section of the revised manuscript]

Consistently, the increased expression of NANOG was also observed in another cisplatin-resistant tumor model (SiHa CR) (Supplementary Fig. 7a). Thus, these results demonstrate that cisplatin depletes tumor cells lacking NANOG while enriching tumor cells containing NANOG, suggesting that NANOG expression in tumor cells could confer a survival advantage under the selection pressure imposed by cisplatin.

To further investigate whether NANOG expression in tumor cells is responsible for the acquisition of cisplatin resistance by autophagy-mediated EGFR activation, we silenced the *NANOG* gene in cisplatin-resistant tumor cells using siRNAs. The knockdown of *NANOG* in the cisplatin-resistant tumor cells led to a significant decrease in autophagosome abundance, EGF secretion and the level of pEGFR (Fig. 3c and Supplementary Fig. 7b), suggesting an important role of NANOG in regulating the autophagy-mediated EGFR activation of cisplatin-resistant tumor cells. Notably, we found that *NANOG* knockdown reduced the resistance of cisplatin-resistant tumor cells to apoptotic cell death by cisplatin (Fig. 3d and Supplementary Fig. 7c).

[On supplementary Fig. 8 of the revised manuscript]

[On page 11, Supplementary Figure legend section of the revised manuscript]

Supplementary Fig. 8. TRPV1 inhibition reduces cisplatin resistance by blocking autophagosome-mediated EGF secretion. **a** The protein levels of TRPV1 were determined by western blot. **b** and **c** SiHa CR or HeLa cells were treated with DMSO or AMG9810. **b** The levels of internal LC3B, pEGFR, and EGFR and secreted EGF protein were confirmed by western blots. **c** Flow cytometry analysis of the frequency of apoptotic (active caspase 3⁺) cells after incubation with or without cisplatin for 24 h. **a** and **b** β -actin was used as an internal loading control. Numbers below the blot images indicate the expression as measured by fold-change. All experiments were performed in triplicate. The p-values by two-way ANOVA are indicated. The data represent the mean \pm SD. Source data are provided as a Source Data file. (*P \leq 0.05, **P \leq 0.01, ***P \leq 0.001. NS, not significant)

[On page 12, Results section of the revised manuscript]

Interestingly, TRPV1 was significantly overexpressed in **cisplatin-resistant tumor** cells compared to **cisplatin-susceptible tumor** cells (Fig. 4b and **Supplementary Fig. 8a**). To determine whether TRPV1 is functionally active in these cells, we examined TRPV1-induced electrical currents upon exposure to its agonist, capsaicin. TRPV1-specific inward currents were elicited by capsaicin and were abolished by the TRPV1-specific antagonist AMG9810 in CaSki CR cells, but not in CaSki P cells (Fig. 4c). Previous studies reported that TRPV1-mediated intracellular Ca²⁺ influx induced autophagy by promoting autophagosome formation^{50, 51}. Interestingly, TRPV1 inhibition reduced autophagosome formation, thereby impeding autophagy-driven EGFR signaling (Fig. 4d) as well as increasing the sensitivity of CaSki CR cells to cisplatin (Fig. 4e). **Moreover, the above results were also reproduced in SiHa CR or HeLa cells (Supplementary Fig. 8b, c).** Thus, our results suggest that TRPV1, which is functionally active as a Ca²⁺-permeable cation channel, **could confer** resistance to cisplatin by regulating **autophagosome formation** in cisplatin-resistant tumor cells.

3) Although the difference in EGFR activity score between low and high cisplatin resistant patients is statistically significant (fig 1b), there is a huge overlap in the values of the two cohorts. This suggests that the EGFR pathway might be hyperactivated only in a fraction of resistant tumors. This is true also per the autophagy signatures in figure 2a. This finding should be acknowledged by the authors in the results and in the discussion of the data. In addition, more details should be provided on the TCGA cohorts analyzed. Are patients in early or advanced stage of disease? Did they receive the same therapeutic regimen?

The reviewer raises an important point that the statistical difference in EGFR activity score between the two cohorts classified as having low or high CR signature might be due to the excessive values of a specific fraction of cohort. To address this point, we first investigated whether EGFR signaling is hyperactivated in cisplatin-resistant cervical cancer patients compared to cisplatin-sensitive

cervical cancer patients. **We newly analyzed the transcriptome data from CESC classified as responders (R, complete and partial response) or non-responders (NR, stable and progressive disease) to cisplatin.** Consistent with the results in our original manuscript, we found that EGFR activity score were statistically higher NR than in R (Fig. 1b of the revised manuscript). However, as the results in our original manuscript, an overlap of two distributions was also observed (proportion of overlap=35.8%).

We next assessed whether the statistical significance between the two groups was determined by a specific fraction of cohort. To do this, we performed two independent non-parametric statistical tests, such as the Wilcoxon Rank sum test and Kolmogorov-Smirnov test. Interestingly, we found that the Wilcoxon Rank sum test result (p-value=0.018) clearly indicated statistically significant differences in the mean of the two groups (Fig. 1a for the reviewer only). Additionally, two samples Kolmogorov-Smirnov test result (p-value=0.025) rejected the null hypothesis that the two groups were drawn from the same distribution, indicating statistically differences in the distribution of the two groups (Fig. 1b for the reviewer only). Thus, these results indicate that **the statistical significance between the two groups is determined by the differences in the mean and distribution of the two groups rather than the value of a specific fraction of cohort.**

Additionally, as the reviewer suggested, we have added clinical information of CESC cohort (Supplementary table 2 of the revised manuscript) and mentioned the source which the clinical data of CESC was obtained in the revised manuscript. We appreciate the reviewer's helpful comment.

[Fig. 1 for the reviewer only]

Wilcox test result

```

wilcoxon rank sum test with continuity correction

data: RSEGFR.activity_Mo and NRSEGFR.activity_Mo
W = 185, p-value = 0.01814
alternative hypothesis: true location shift is not equal to 0
  
```

Two sample kolmogorow-smirnow test

```

> ks.test(RSEGFR.activity_Mo, NRSEGFR.activity_Mo, alternative = 'greater')

Exact two-sample Kolmogorov-Smirnov test

data: RSEGFR.activity_Mo and NRSEGFR.activity_Mo
D^+ = 0.41212, p-value = 0.02526
alternative hypothesis: the CDF of x lies above that of y

Two samples are from different distribution
  
```

4) In figure 1c the quantification of the bands is not clear, all the bands are reported with a value of 1.0

We apologize for our mistakes. To correct it, we have revised the quantification value of the bands in the revised manuscript as followed:

[On Fig. 1c, 5g, and 5h of the revised manuscript]

5) In figure 1e and 1g the concentration of cisplatin is not shown. In addition, the method used for the measurement of apoptotic cell death should be reported in the figure legend.

We thank the reviewer's helpful comment. As the reviewer suggested, we have shown the concentrations of all agents we used in this study. In addition, we have revised the figure legend sections to report the method used for the measurement of apoptotic cell death in the revised manuscript as followed:

[On Fig. 1e, g of the revised manuscript]

[On page 34, Figure legend section of the revised manuscript]

e and g Flow cytometry analysis of the frequency of apoptotic (active caspase 3⁺) cells after incubation with or without cisplatin for 24 h.

6) *The authors state that both siEGFR 1 and 2 modulated cisplatin resistance in P3 cells. However, data on siEGFR 2 are not shown/described in the paper nor in the supplementary data.*

We appreciate the reviewer for pointing out our mistakes. We found that the data on *siEGFR* #2 was not shown on figure 1g by mistake. Thus, we have added the data on *siEGFR* #2 in the revised manuscript as followed:

[On Fig. 1g of the revised manuscript]

7) *The authors found that the increase in EGFR activation was due to increased autophagy-mediated release of EGF. In this respect, the quantification of EGF in the medium from P3 cells (fig. 2e) should be performed by using a quantitative method such as ELISA or similar approaches. Even more importantly, the increase in EGFR activation could be due to the increased secretion of a number of different EGFR ligands that are frequently expressed in cancer cells. Therefore, the levels of secretion of other EGF-like peptides should also be assessed.*

We agree with the reviewer's opinion that the levels of secreted EGFR ligands should be assessed and the quantification of secreted EGFR ligands should be performed by using ELISA. Previous studies indicated that EGFR activation is mediated by EGF and its related ligands, and that increased phosphorylation of EGFR through the autocrine growth factor loop plays an important role in human cancers (Schneider MR et al., J Cell Physiol, 2009).

In this regard, we assessed that levels of both intracellular and secreted high affinity EGFR ligands, such as EGF, TGF- α , HB-EGF, and BTC. **Interestingly, among the EGFR ligands, only EGF secretion was increased in CaSki CR cells compared to CaSki P cells in our experiment conditions (Supplementary Fig. 4a of the revised manuscript).** Indeed, we also found that the amount of secreted EGF was 4-fold higher in CaSki CR cells than in CaSki P cells using ELISA

assay (Supplementary Fig. 4b of the revised manuscript). Thus, these data suggest that EGF secretion could be a crucial cause that activates EGFR signaling pathway, thereby inducing cisplatin resistance in cisplatin-resistant cervical cancer cells.

We appreciate the invaluable comment that has helped us improve the quality of our manuscript. We have included these data and description in the revised manuscript as followed:

[On supplementary Fig. 4a, b of the revised manuscript]

[On page 8, Supplementary Figure legend section of the revised manuscript]

Supplementary Fig. 4. Secreted EGF plays crucial roles in cisplatin-resistant phenotypes in tumor cells. **a** The levels of internal EGF, HB-EGF, BTC, and TNF- α and secreted EGF, HB-EGF, BTC, TNF- α protein were determined by western blot analysis. **b** The amount of EGF secreted into the media was measured by ELISA.

[On page 9, Results section of the revised manuscript]

Since an increase of autophagosome abundance could trigger the secretion of intracellular molecule⁴³, we first investigated secretion levels of EGFR ligands. Previous studies indicated that EGFR activation is mediated by EGF and its related ligands, and that increased phosphorylation of EGFR through the autocrine growth factor loop plays an important role in human cancers⁴⁴. In this regard, we assessed those levels of both intracellular and secreted EGFR ligands in CaSki cells. Among the EGFR ligands, only EGF secretion was increased in CaSki CR cells compared to CaSki P cells (Supplementary Fig. 4a, b)

[On page 23, Materials and Methods section of the revised manuscript]

EGF assessment of medium supernatants

The experimental procedure has been described previously¹⁸. Briefly, the cells were grown in 6-well plates and incubated with medium containing 0.1% FBS for 48 h at 37°C. The supernatants were collected and centrifuged to remove cell debris. The EGF levels in the supernatants were measured by the Quantikine Human EGF ELISA kit (R&D Systems, Minneapolis, MN, USA, DEG00). For western blot analysis, the supernatants were further concentrated 30X by Centricon Plus-70 centrifugal Filter Units-3kDa (Millipore, UFC800324).

8) Similarly, the effects of NANOG signaling on different EGF-like peptides in addition to EGF should be assessed by using a quantitative approach (fig. 3c).

We thank the reviewer for raising this important suggestion. **As shown supplementary Fig. 4 of the revised manuscript, our data suggest that EGF could influence the NANOG-mediated cisplatin resistance rather than other EGFR ligands.** Furthermore, to verify the potential role of secreted EGF in the NANOG-mediated cisplatin resistance in tumor cells, we neutralized EGF with a specific monoclonal antibody in CaSki NANOG cells. We found that EGF neutralization decreased the NANOG-mediated cisplatin-resistant phenotype (Fig. 3g of the revised manuscript). Conversely, the treatment of recombinant EGF to NANOG-silenced CaSki CR cells increased resistance to apoptotic cell death by cisplatin (Fig. 3h of the revised manuscript). Thus, these results suggest that an immediate role for EGF in the NANOG-mediated cisplatin resistance.

We have added these data and description in the revised manuscript as followed:

[On Fig. 3g, h of the revised manuscript]

[On page 37, Figure legend section of the revised manuscript]

g and h. In the cells treated with or without anti-EGF or EGF, the frequency of apoptotic (active

caspace 3⁺) cells after incubation with or without cisplatin for 24 h were confirmed by flow cytometry.

[On page 11, Results section of the revised manuscript]

We further questioned that the role of secreted EGF in NANOG-mediated cisplatin resistance. To verify the potential role of EGF in the NANOG-mediated cisplatin resistance in tumor cells, we neutralized EGF with a specific monoclonal antibody in CaSki NANOG cells. We found that EGF neutralization decreased the NANOG-mediated cisplatin-resistant phenotype (Fig. 3g). Conversely, the treatment of recombinant EGF to NANOG-silenced CaSki CR cells increased resistance to apoptotic cell death by cisplatin (Fig. 3h), suggesting an immediate role for EGF in the NANOG-mediated cisplatin resistance. Taken together, our results demonstrate that NANOG is a critical mediator that could confer cisplatin resistance by regulating secretory autophagy-mediated EGFR activation.

9) The author should discuss why they focused on TRPV1 among the 21 upregulated genes and briefly discuss the possible role of other genes not investigated in details in this paper.

We thank the reviewer's this helpful comment. As the reviewer suggested, we have included additional information of the putative target genes reported to be related to induce chemotherapy-refractory phenotypes in the revised manuscript. Among the putative target genes, *ACSBG2*, *DUOX2*, and *TRPV1* were previously reported that could be involved in various chemotherapy-refractory phenotypes. For example, Acyl-CoA Synthetase Bubblegum Family Member 2 (*ACSBG2*) induces resistance to apoptotic cell death by increasing mitochondrial membrane lipids (Li YJ et al., Cell Rep. 2022;39:110870). In addition, Dual Oxidase 2 (*DUOX2*) induces epithelial mesenchymal transition through production of reactive oxygen species (Kang KA et al., Redox Biol 2018;17:224-235).

Although these putative targets have been reported to be related to chemotherapy-refractory phenotypes, their clinically available agents have not been developed. On the other hand, TRPV1 is not only involved in various chemotherapy-refractory phenotypes, but also clinically available agents have been developed. Thus, we focused on TRPV1 as an available target to overcome NANOG-mediated autophagic secretion.

To help the reader's understanding, we have included additional information on the reason why we focused on TRPV1 in revised manuscript as followed:

[On page 11, 12, Results section of the revised manuscript]

To investigate potential targets for impeding NANOG-driven autophagy in cisplatin-resistant tumor cells, we compared the expression of genes involved in EGFR hyperactivation among the genes upregulated by NANOG and highly expressed in chemotherapy-resistant cervical cancer patients (Fig.

4a). Among the putative target genes, *ACSBG2*, *DUOX2*, and *TRPV1* were previously reported that could be involved in various chemotherapy-refractory phenotypes. For example, Acyl-CoA Synthetase Bubblegum Family Member 2 (*ACSBG2*) could induce resistance to apoptotic cell death by increasing mitochondrial membrane lipids⁴⁶. In addition, Dual Oxidase 2 (*DUOX2*) induces epithelial mesenchymal transition through production of reactive oxygen species⁴⁷. Although these putative targets have been reported to be related to chemotherapy-refractory phenotypes, clinically available agents have not been developed. On the other hand, *TRPV1* is not only involved in various chemotherapy-refractory phenotypes, but also clinically available agents have been developed. Thus, we focused on *TRPV1* as an available target to overcome NANOG-mediated autophagic secretion.

10) *TRPV1* levels in cisplatin resistant cells should also be assessed with western blotting (Fig. 4b).

As the reviewer suggested, we assessed level of *TRPV1* expression using western blot analysis. Consistent with the immunofluorescence image, the level of *TRPV1* expression was increased in CaSki CR cells compared to CaSki P cells.

We have included these data in Figure section of the revised manuscript as followed:

[On Fig. 4b of the revised manuscript]

[On page 40, Figure legend section of the revised manuscript]

b The cells were stained with anti-*TRPV1* (red) antibodies and then visualized by confocal microscopy. DAPI was used to stain the nuclei. The images are representative of three separate experiments. Scale bar, 10 μ m. The graph depicts the experimental quantitation of *TRPV1*. The protein levels of *TRPV1* and β -actin were confirmed by western blots.

11) The TRPV1 inhibitor AMG9810 has been used at quite high concentration in different experiments (10 micromolar). The authors should show the effects of lower doses of AMG9810 on the activation of EGFR and secretion of EGF and other EGF-like peptides, in order to demonstrate a specific dose-dependent effect of the drug.

We appreciate the reviewer's insightful comment. As the reviewer suggested, we performed additional experiments by treating various concentrations of TRPV1 inhibitor, AMG9810. Interestingly, we found that TRPV1 inhibition by AMG9810 reduced autophagy-mediated EGF secretion and EGFR activation as well as the cisplatin-resistant phenotypes of CaSki CR cells in a dose-dependent manner (Fig. 4d, e of the revised manuscript). As shown supplementary Fig. 4 of the revised manuscript, since we found that among the EGFR ligands, only EGF secretion was increased in CaSki CR cells compared to CaSki P cells in our experiment conditions, we did not assess the secretion levels of other ligands. Thus, these results indicate that TRPV1 inhibitor could reduce resistance to cisplatin by suppressing autophagic EGF secretion in a dose-dependent manner.

We have included these data and description in the revised manuscript as followed:

[On Fig. 4d, e of the revised manuscript]

[On page 40, Figure legend section of the revised manuscript]

d and **e** CaSki P3 cells were treated with dimethyl sulfoxide (DMSO) or AMG9810, as indicated. **d** The protein levels of secreted EGF and internal LC3B, pEGFR, and EGFR were confirmed by western blots. **e**. Flow cytometry analysis of the frequency of apoptotic (active caspase 3⁺) cells after incubation with or without cisplatin for 24 h.

[On page 12, Results section of the revised manuscript]

Previous studies reported that TRPV1-mediated intracellular Ca²⁺ influx induced autophagy by

promoting autophagosome formation^{50, 51}. Interestingly, TRPV1 inhibition reduced autophagosome formation, thereby impeding autophagy-driven EGFR signaling (Fig. 4d) as well as increasing the sensitivity of CaSki CR cells to cisplatin **in a dose-dependent manner** (Fig. 4e).

12) The effects of MUT NANOG on TRPV1 were more significant in HEK293 as compared with CaSki cells. Is this due to the higher basal levels of TRPV1? This finding needs to be discussed.

We apologize for the confusion caused by our mistakes. To confirm the effects of NANOG MT on TRPV1 expression, we performed additional experiments. As shown below in Fig. 4h of the revised manuscript, NANOG MT did not increase TRPV1 expression in HEK293 cells as that much in CaSki cells at this time. We have recognized that previous TRPV1 western blot image contained black dots around the band of TRPV1 after taking a closer look at it. We have thought that those dot images could make confusion to reader.

Therefore, to correct our mistake, we have changed the old TRPV1 blot image with the new one in the revised manuscript as followed:

[On Fig. 4h of the revised manuscript]

13) The cohort of cervical cancer patients analyzed is too small to drive any firm conclusion (fig. 6). In particular, when combining the three biomarkers (NANOG/TRPV1/pEGFR), a subgroup includes only 9 patients. The authors need to significantly increase the number of patients if they want to support their hypothesis with clinical findings. In addition, the authors must acknowledge that increased survival observed in this retrospective analysis could be due to a prognostic and not predictive value of the investigated biomarkers. In fact, the predictive value of a biomarker can be only assessed within a randomized clinical trial. Finally, more information on the clinical and pathologic features of these patients should be provided at least in the supplementary material.

We concur that our IHC data needs to be reinforced by adding up more human clinical samples and will offer more concrete data to support our hypothesis. Unfortunately, although we agree with reviewer's opinion, there is a critical reason that we could not collect cisplatin-resistant human clinical samples.

Concurrent chemoradiotherapy is one of the standard-of-care treatments for locally advanced cervical cancer patients with stage 1B2 to IVA disease (usually applied to stage IIB and III patients). Given that the number of cervical cancer patients is rapidly decreasing in South Korea Like other advanced countries and more than 50% of patients with cervical cancers are usually diagnosed at stage I or IIA (they usually get radical hysterectomy instead of concurrent chemoradiotherapy) due to the Korean National Cancer Screening Program it is hard to collect large number of locally advanced cervical cancer tissue samples with chemoradiation data. Among the original human cervical TMAs which include 555 cervical neoplasia cases (low-grade CIN; 97, High-grade CIN; 280, and cancer; 178), only 77 cases are chemoradiation response data.

We had used those 77 cases for analysis and exclusively focused on whether the NANOG-TRPV1-pEGFR axis is associated with chemoradiation responsiveness and a patient survival in human cervical cancer. **Although we had used a small number of patients to verify our hypothesis, the results in this study were statistically significant and successfully supported our hypothesis of this study.** Therefore, we believe that these results indicate the NANOG-TRPV1-pEGFR axis links a potential association with a chemoradiation responsiveness and a patient survival in human cervical cancer.

Nevertheless, we also agree with the reviewer's opinion on the terminological difference between "predictive biomarker" and "prognostic biomarker". Therefore, **we have removed the term "predictive" from the manuscript.** In addition, as the reviewer commented, we have added a new Supplementary Table 1 for clinical pathological characteristics as followed:

[On page 14, Results section of the revised manuscript]

Thus, our results indicate that the **NANOG-TRPV1-pEGFR axis links a potential association with a chemoradiation responsiveness and a patient survival in human cervical cancer.**

[On page 24, 25, Materials and Methods section of the revised manuscript]

Immunohistochemistry

We used human cervical neoplasia tissue microarrays (TMAs), as previously reported²⁴. A total of 77 surgically resected primary cervical cancer patients (53 chemoradiosensitive and 24 chemoradioresistant) were analyzed in this study. **The patient's clinicopathological characteristics are summarized in Supplementary Table 1.** Some of the paraffin blocks for the chemoradioresistant patients

were provided by the Korea Gynecologic Cancer Bank through the Bio & Medical Technology Development Program of the Ministry of Education, Science and Technology, Korea (NRF-2017M3A9B8069610). Tissue samples and medical records were obtained with approval by the Institutional Review Board of Gangnam Severance Hospital (Seoul, KOR). Cancer patients who had a recurrence within one year after chemoradiotherapy were considered to have a “resistant response”.

[On page 22, Supplementary Table 1 of the revised manuscript]

Clinicopathological characteristics patients with cervical cancer

	No.	Chemoradiation response		p value
		Good No. (%)	Bad No. (%)	
Age				0.845
≤ 50	42	28 (52.8)	14 (58.3)	
> 50	35	25 (47.2)	10 (41.7)	
FIGO stage				<0.001
I	40	36 (67.9)	4 (16.7)	
II	32	17 (32.1)	15 (62.5)	
IV	5	0	5 (20.8)	
Tumor grade				0.293
Well	2	1 (1.9)	1 (4.2)	
Moderate	49	37 (69.8)	12 (50.0)	
Poor	25	14 (26.4)	11 (45.8)	
Unknown	1	1 (1.9)	0	
Cell type				0.316
SCC	60	40 (75.4)	20 (83.4)	
AD	13	11 (20.8)	2 (8.3)	
Others	4	2 (3.8)	2 (8.3)	
Tumor size				1.000
≤ 4 cm	43	30 (56.6)	13 (54.2)	
> 4 cm	34	23 (43.4)	11 (45.8)	
Lymphnode metastasis				0.111
Absent	36	29 (54.7)	7 (29.2)	
Present	25	15 (28.3)	10 (41.7)	
Unknown	16	9 (17.0)	7 (29.2)	
Lymphovascular invasion				0.014
Absent	20	18 (34.0)	2 (8.2)	
Present	36	25 (47.2)	11 (45.8)	
Unknown	21	10 (18.8)	11 (45.8)	
SCC antigen				
Negative	40	31 (58.5)	9 (37.5)	
Positive	27	15 (28.3)	12 (50.0)	
Unknown	10	7 (13.2)	3 (12.5)	
HPV				0.141
Negative	2	2 (3.8)	0	
Positive	42	32 (60.4)	10 (41.7)	
Unknown	33	19 (35.8)	14 (58.3)	
Recur				<0.001
Absent	45	45 (84.9)	0	
Present	32	8 (15.1)	24 (100.0)	
Status				<0.001
Alive	60	49 (92.5)	11 (45.8)	
Expire	17	4 (7.5)	13 (54.2)	

FIGO, International Federation of Gynecology and Obstetrics; SSC, squamous cell carcinoma; AD, adenocarcinoma; HPV, human papillomavirus

14) The authors claim that the mechanism they identified could involve different types of cancer. However, only one gastric cancer cell line was found to overexpress NANOG (supplementary figure 2), whereas no data are shown for other cancer cell lines. The authors should at least acknowledge that expression of NANOG is heterogeneous in these cancer types. In addition, additional information on the cohorts of patients included in these analyses should be provided (early vs advanced stage, therapeutic regimens etc).

The reviewer raises an important point regarding that the level of NANOG expression was not confirmed in other cancer cell lines other than gastric cancer cell lines. In our previous publications (Song KH et al., Cancer Research, 2017, Oh SJ et al., Cancer Research, 2018), we had assessed the levels of NANOG protein in a variety of human cancer cell lines. In this study, we intended to verify the functional effects of the NANOG-TRPV1-pEGFR axis in diverse types of cancer which cisplatin was used as the major chemotherapeutic agent. Therefore, we focused on lung and gastric cancer among the various cancer types. **Since it was confirmed that H1299 exhibited the highest NANOG expression among lung cancer in our previous two papers (Song KH et al., Cancer Research, 2017, Oh SJ et al., Cancer Research, 2018), we did not assess NANOG expression in lung cancer cell lines again in this study.** However, the expression of NANOG has not been validated in gastric cancer cell lines, we additionally assessed it in this study.

Additionally, we also agree with the reviewer's opinion that as the NANOG expression is heterogeneous in each cancer cell lines, the cisplatin resistance caused by the NANOG-TRPV1-pEGFR axis could not be conserved in all types of cancer cell lines. For this reason, to verify our hypothesis in this study, we selected a representative cell line with high NANOG expression, such as H1299 (lung cancer) and MKN28 (gastric cancer), respectively. Interestingly, we found that the NANOG-TRPV1-pEGFR axis was conserved, and TRPV1 inhibition overcame cisplatin resistance in the representative NANOG^{high} tumor cell lines (Supplementary Fig. 15, 16a, and 16b of the revised manuscript). Thus, these results suggest that the **functional properties of the NANOG-TRPV1-pEGFR axis are conserved across NANOG^{high} lung and gastric cancer cells**, and TRPV1 could be a potential target for controlling NANOG^{high} cisplatin-resistant cancer.

We appreciate the invaluable comment of the reviewer. To avoid the confusion and help reader's understanding, we have revised the Result section and included clinical information source of LUAD, LUSC, STAD, and STES cohorts as followed:

[On page 15, Results section of the revised manuscript]

TRPV1 is a therapeutic target for **cisplatin-resistant NANOG⁺ cancer cells**

To verify the functional effects of the NANOG-TRPV1-pEGFR axis in diverse types of cancer **which cisplatin was used as the major chemotherapeutic agent, we focused on lung and gastric cancer among the various cancer types.**

[On page 15, 16, Results section of the revised manuscript]

To further confirm whether a cisplatin resistance caused by the NANOG-TRPV1-pEGFR axis was conserved in NANOG^{high} lung or gastric cancer cells, we first tried to select NANOG^{high} lung or gastric cancer cell lines, respectively. In our previous studies, we profiled the levels of NANOG protein in a variety of human cancer cells^{27, 29}. Among lung cancer cells, we designated H1299 as a representative NANOG^{high} lung cancer cell line. In the case of gastric cancer cells, we newly screened the levels of NANOG protein and selected MKN28 as a representative NANOG^{high} gastric cancer cell (Supplementary Fig. 14). Interestingly, NANOG knockdown reduced TRPV1 expression in those tumor cells (Supplementary Fig. 15), indicating that the NANOG axis was conserved in all tested cells. Notably, TRPV1 inhibition by AMG9810 robustly dampened EGFR or AKT phosphorylation and downregulated MCL1 expression in the tumor cells (Supplementary Fig. 16a). Furthermore, treatment with AMG9810 led to a significantly increase in apoptotic cell death caused by cisplatin (Supplementary Fig. 16b). Thus, our results demonstrate that the functional properties of the NANOG-TRPV1-pEGFR axis are conserved across NANOG^{high} lung and gastric cancer cells, and TRPV1 could be a potential target for controlling NANOG^{high} cisplatin-resistant cancer.

[On page 26, Materials and Methods of the revised manuscript]

TCGA data collection and analysis

Gene expression data for CESC, LUAD, LUSC, STAD, and STES samples profiled by TCGA were collected from the Firehose BROAD GDAC data repository (<https://gdac.broadinstitute.org/>). Clinical data were also retrieved from the same source.

15) The authors claim that TRPV1 inhibition and cisplatin treatment might have a synergistic effect. However, in order to demonstrate synergism, the authors need to perform in vitro and/or in vivo experiments with combinations of the two drugs at different concentrations, according to specific models evaluating the interaction between different.

We agree with the reviewer's opinion. As the reviewer suggested, to demonstrate the synergistic effect of TRPV1 inhibitor (AMG9810) on sensitizing NANOG^{high} tumor cells to cisplatin treatment, we performed in vitro experiment with combinations of the two drugs at different concentrations (Fig. 8a, b of the revised manuscript). **Compared to cisplatin alone treatment, TRPV1 inhibitor treatment increased cisplatin-mediated cell death in dose-dependent manner** (Fig. 8a of the revised manuscript).

To quantify the synergistic effects of treatment of TRPV1 inhibitor with cisplatin, a combination score was calculated based on changes in the percentage of apoptotic cell death in cisplatin-treated tumor cells with or without TRPV1 inhibitor. Interestingly, we found that **treatment with**

a sublethal dose of TRPV1 inhibitor enhanced cisplatin-mediated cell death in a synergistic fashion (Fig. 8b of the revised manuscript). Thus, our data indicate that TRPV1 inhibition could be a promising therapeutic strategy to overcome the cisplatin-resistant phenotypes of NANOG^{high} tumor cells.

We appreciate the insightful and helpful comment that improves the quality of our manuscript. We have included these data and description in the revised manuscript as followed:

[On Fig. 8a, b of the revised manuscript]

[On page 45, Figure legend section of the revised manuscript]

Fig. 8. In vivo targeting of autophagy by TRPV1 inhibition renders tumors susceptible to cisplatin in a pre-clinical model. **a** CaSki CR cells were treated with cisplatin and AMG9810, as indicated. The frequency of apoptotic (active caspase 3⁺) cells was analyzed by flow cytometry. **b** The combination score was calculated based on changes in the percentage of apoptotic cells in cisplatin-treated tumor cells with or without AMG9810. Combination score = (% of active caspase 3⁺ tumor cells by cisplatin and AMG9810) / ((% of active caspase 3⁺ tumor cells by cisplatin)).

[On page 16, Results section of the revised manuscript]

Given our observation in vitro, we reasoned that the in vivo administration of TRPV1 inhibitor, AMG9810, could overcome resistance to cisplatin. To investigate the effect of AMG9810 on sensitizing NANOG^{high} tumor cells to cisplatin treatment, CaSki CR cells were co-treated with cisplatin and AMG9810 at various dose. Compared to cisplatin alone treatment, combined treatment of cisplatin with AMG9810 increased apoptotic cell death in dose-dependent manner (Fig. 8a). To quantify the synergistic effects of treatment of AMG9810 with cisplatin, a combination score was calculated based on changes in the percentage of apoptotic cell death in cisplatin-treated tumor cells with or without AMG9810. We found that treatment with a sublethal dose of AMG9810 enhanced the cisplatin-mediated cell death in a synergistic fashion (Fig. 8b). Given these data, we concluded that AMG9810 could be a promising therapeutic strategy to overcome the cisplatin-resistant phenotypes of NANOG^{high} tumor cells.

.16) Finally, the authors conclude that TRPV1 inhibition is a clinically available strategy to target cisplatin resistance. However, the clinical utility of this approach needs to be addressed in prospective clinical trials before making any conclusions.

We understand the reviewer's concern regarding that the clinical utility of our finding needs to be addressed in prospective clinical trials before making any conclusions. Although not only the importance of TRPV1 as a pharmacological target continues to grow (Li L et al., Int J Biol Sci, 2021) but also TRPV1 inhibitors have clinically used in various diseases including atopic dermatitis and neuropathic pain (Aghazadeh Tabrizi M et al., Med Res Rev, 2017, Koivisto AP et al., Nat Rev Drug Discov, 2021), the clinical trials of TRPV1 inhibitors for cancer treatment is yet to be studied.

In this study, to evaluate the preclinical therapeutic value of TRPV1 targeting, we inoculated the CaSki CR cells into NOD-SCID mice. Nine days after the tumor challenge, the mice were treated with cisplatin along with chitosan hydrogel containing AMG9810 (TRPV1 inhibitor) (Fig. 8c of the revised manuscript). **While cisplatin treatment alone had no effect on the growth of cisplatin-resistant tumors, combined treatment with cisplatin and AMG9810 retarded tumor growth** (Fig. 8d, e of the revised manuscript) and **prolonged the survival of the mice** (Fig. 8f of the revised manuscript).

Therefore, **we believe that our results in this study are enough to provide proof-of-concept evidence that TRPV1 inhibition could be a plausible strategy to control NANOG^{high} cisplatin-resistant cancer as well as chemotherapy-induced neuropathic pain (CIPN).**

Nevertheless, we agree with the reviewer's opinion. **Therefore, we have tone downed the expression that could cause misunderstandings** as followed:

We appreciate the reviewer's invaluable comment.

[On page 6, Introduction section of the revised manuscript]

Thus, we provide proof-of-concept evidence that TRPV1 inhibition **could be a plausible** strategy to control NANOG^{high} cisplatin-resistant cancer as well as CIPN.

[On page 16, Results section of the revised manuscript]

Given these data, we concluded that AMG9810 **could be a plausible** therapeutic strategy to overcome the cisplatin-resistant phenotypes of NANOG^{high} tumor cells.

[On page 17, Results section of the revised manuscript]

Taken together, we conclude that TRPV1 inhibition **could overcome** the cisplatin resistance of tumor cells and **represent** an attractive strategy for the control of human cancer as a synergistic agent

with cisplatin treatment.

[On page 19, Discussion section of the revised manuscript]

Thus, we provide proof-of-concept evidence that TRPV1 inhibition **could be** a clinically available strategy to control NANOG^{high} cisplatin-resistant cancer and CIPN.

Reviewer #2

The reasonably well written article presents results to link hyperactivation of EGFR, ion channel TRPV1, NANOG, autophagy-mediated EGF secretion via Ca²⁺ influx (the NANOG-TRPV1-pEGFR axis) in cisplatin resistance. Although abundant data is presented, the support for the somewhat complicate pathways is weak. The proposed links seem to be random rather than systematic, and likely not to be the main routes or mechanisms. The data presented are not of high quality or clarity. Although the experiments and rationales are based on previous publications by the authors, the current manuscript does not present a solid concept.

1) Hyperactivated EGFR confers resistance to cisplatin in cervical cancer: other mechanisms are likely, not limiting to EGFR hyperactivation.

We agree with the reviewer's opinion that there are various mechanisms which induce cisplatin resistance in cervical cancer other than EGFR hyperactivation. **We would like to clarify that the purpose of our studies is identifying the cause of emergence of cisplatin resistance in cervical cancer patients with EGFR hyperactivation, but not all cervical cancer patients.**

The cisplatin resistance in cervical cancer is complex and associated with the following features: 1) reduction in the intracellular accumulation of the platinum compounds (decrease in uptake, increase in efflux, and increased drug detoxification by cellular thiols); 2) increase in DNA damage repair (increased NER, loss of MMR, and increased TLS); 3) inactivation of apoptosis; 4) activation of EMT; 5) alteration in DNA methylation, microRNA profile, cancer stem cell characteristics, and expression of stress-response chaperones (Haiyan Zhu et al., Drug Des Devel Ther. 2016).

Among the diverse molecular mechanisms, the epidermal growth factor receptor (EGFR), an oncogenic receptor tyrosine kinase, hyperactivation has attracted attention, as it can trigger cisplatin resistance features above. In tumor cells, **the EGFR hyperactivation increases drug efflux, DNA damage repair, anti-apoptosis, and stem-like properties** (Sol-Bi Shin et al., Cancers. 2021, David J. Chen et al., Clinical Cancer Res. 2007, Suyeon Kim et al., Autophagy. 2020, Chien-Hui Weng et al., Oncogene. 2019). Notably, hyperactivated EGFR signaling was found in various cancer types that acquire cisplatin resistance (Rho JK et al., Oncol Res. 2011, Zhu L et al., J Exp Clin Cancer Res. 2018, Ma L et al., FEBS J. 2013). In this regard, results from recent studies have suggested that the **hyperactivation of EGFR signaling could be a one of major pathway that drives the adaptation of tumor cells to cisplatin, thereby contributing to the generation of cancer cells with better survival advantages.** Therefore, it is important to identify the cause of EGFR hyperactivation to overcome the clinical limitations of cisplatin treatment in cervical cancer patient with hyperactivated EGFR signaling.

In this study, we found that the emergence of cisplatin resistance was highly associated with EGFR hyperactivation on tumor cells (Fig. 1 of the revised manuscript), and the EGFR hyperactivation was arisen by a transcriptional increase in the pain-generating channel, TRPV1, via NANOG (Fig. 3, 4

of the revised manuscript). Furthermore, TRPV1 promoted autophagy-mediated EGF secretion via Ca²⁺ influx, which activated the EGFR-AKT signaling pathway and, consequentially, the acquisition of cisplatin resistance (Fig. 5 of the revised manuscript). Notably, TRPV1 inhibition using AMG9810, a potential pain-reliever, rendered tumors susceptible to cisplatin treatment and led to the successful control of the disease (Fig. 8 of the revised manuscript).

Thus, we believe that our findings in this study provide proof-of-concept evidence that TRPV1 inhibition could be a plausible strategy to control cisplatin resistance as well as CIPN at least for cervical cancer patients with hyperactivated EGFR signaling.

To clarify the purpose of our study, we have revised introduction section of the revised manuscript as followed:

[On page 4, Introduction section of the revised manuscript]

The cisplatin resistance in cervical cancer are complex and associated with the following features: 1) reduction in the intracellular accumulation of the platinum compounds (decrease in uptake, increase in efflux, and increased drug detoxification by cellular thiols); 2) increase in DNA damage repair; 3) inactivation of apoptosis; 4) activation of EMT; 5) alteration in DNA methylation, microRNA profile, cancer stem cell characteristics, and expression of stress-response chaperones¹⁵. Among the diverse molecular mechanisms, the epidermal growth factor receptor (EGFR), an oncogenic receptor tyrosine kinase, hyperactivation has attracted attention, as it can trigger cisplatin resistance features above. In tumor cells, the EGFR hyperactivation increases drug efflux, DNA damage repair, anti-apoptosis, and stem-like properties^{16, 17, 18, 19}. Notably, hyperactivated EGFR signaling was found in various cancer types that acquire cisplatin resistance^{20, 21, 22}. In this regard, results from recent studies have suggested that the hyperactivation of EGFR signaling could be a one of major pathway that drives the adaptation of tumor cells to cisplatin, thereby contributing to the generation of cancer cells with better survival advantages. Therefore, it is important to identify the cause of EGFR hyperactivation to overcome the clinical limitations of cisplatin treatment in cervical cancer patient with hyperactivated EGFR signaling.

2) Autophagosome mediates EGF secretion: the data is not sufficient and the conclusion seems not convincing.

We understand the reviewer's concern. To remedy the weak point of our results, we performed additional experiments to analyze the autophagy flux in cisplatin-susceptible and -resistant tumor cells. We first treated bafilomycin A1 (BafA), an inhibitor of V-ATPase which blocks the fusion of autophagosomes to lysosomes, to CaSki P and CR cells. While the protein abundance of LC3B-II was slightly increased by BafA in CaSki CR cells compared with CaSki P cells, there was no significant alteration in autophagy flux in CaSki CR cells compared to CaSki P cells (Supplementary Fig. 3 of the revised manuscript). These results indicate that **cisplatin-resistant cervical cancer cells exhibit an**

increase in autophagosome abundance rather than autophagy flux.

We next investigated which stages of the autophagy pathway is crucial for EGF secretion and EGFR signaling-mediated cisplatin resistance by using two other autophagy inhibitors, spautin-1 and BafA. Spautin-1 is a specific inhibitor of the PtdIns3K catalytic subunit (PIK3C3/VPS34) responsible for autophagosome formation. Interestingly, increased EGF secretion, pEGFR level and cisplatin-resistant phenotypes in CaSki CR cells were strongly suppressed by treatment with spautin-1 but not with BafA (Fig. 2g, h and Supplementary Fig. 5 of the revised manuscript). These results suggest that **autophagosome formation is important for EGF secretion and cisplatin resistance in CaSki CR cells, but late autophagy components involved in autophagosome lysosome fusion appear to be dispensable in the process.** Colinearly, compared to SiHa P cells, SiHa CR cells also showed increase in autophagosome formation and EGF secretion (Supplementary Fig. 6a of the revised manuscript). Notably, the suppression of autophagosome formation by spautin-1 reduced EGF secretion, the level of pEGFR and cisplatin-resistant phenotypes in SiHa CR cells (Supplementary Fig. 6a, b of the revised manuscript). Thus, these data support our hypothesis that autophagosome-mediated EGF secretion induces hyperactivation of EGFR signaling, thereby promoting resistance to cell death by cisplatin in cervical cancer cells.

We appreciate the invaluable comment that has helped us remedy the weakness of our manuscript. We have included these data and description in the revised manuscript as followed:

[On Supplementary Fig. 3 of the revised manuscript]

[On page 7, Supplementary Figure legend section of the revised manuscript]

Supplementary Fig. 3. Autophagy flux is not altered in cisplatin-resistant tumor cells. CaSki P or CR cells were starved in medium supplemented with 0.1% FBS for 48 h and then incubated with or without BafA (10nM) for 12 h. The protein levels of LC3B were confirmed by western blots. β -actin was used as an internal loading control. Numbers below the blot images indicate the expression as measured by fold-change. The graphs represent quantification for absolute level of LC3B-II (left, ***P

≤ 0.001, by two-way ANOVA) and autophagic flux (right, p = N.S by 2-tailed Student's t-test), respectively. Autophagic flux was calculated by dividing the value of LC3B-II in the presence of BafA by that absence BafA. All experiments were performed in triplicate. The p-values by two-way ANOVA are indicated. The data represent the mean ± SD. Source data are provided as a Source Data file. (*P ≤ 0.05, **P ≤ 0.01, ***P ≤ 0.001. NS, not significant)

[On page 9, Result section of the revised manuscript]

Interestingly, there was no significant alteration in autophagic flux in CaSki CR cells compared to CaSki P cells (Supplementary Fig. 3). Instead, the amount of LC3B-II escalated in parallel with increase in the amount of total cellular LC3B in CaSki CR cells (Fig. 2b). Therefore, our data indicate that cisplatin-resistant cervical cancer cells exhibit an increase in autophagosome abundance rather than autophagy flux.

[On Fig. 2g, h and Supplementary Fig. 5 of the revised manuscript]

[On page 35, 36, Figure legend section of the revised manuscript]

g In the cells treated with or without spautin-1, the protein levels of secreted EGF and internal LC3B, pEGFR, EGFR, and β-actin were determined by western blots. **h** Flow cytometry analysis of the frequency of apoptotic (active caspase 3⁺) cells after incubation with or without cisplatin for 24 h. **b, d** and **g** β-actin was used as the internal loading control.

[On page 9, Supplementary Figure legend section of the revised manuscript]

Supplementary Fig. 5. Increased EGF secretion in cisplatin-resistant tumor cells is not affected by treatment with BafA. In the cells treated with or without BafA, the protein levels of secreted

EGF and internal LC3B, pEGFR, EGFR, and β -actin were determined by western blots. β -actin was used as an internal loading control. Numbers below the blot images indicate the expression as measured by fold-change. This experiment was performed in triplicate.

[On Supplementary Fig. 6 of the revised manuscript]

[On page 10, Supplementary Figure legend section of the revised manuscript]

Supplementary Fig. 6. Autophagosome formation is crucial for EGF secretion and EGFR signaling-mediated cisplatin resistance in SiHa CR cells. **a** In the cells treated with or without spautin-1, the protein levels of secreted EGF and internal LC3B, pEGFR, EGFR, and β -actin were determined by western blots. β -actin was used as an internal loading control. Numbers below the blot images indicate the expression as measured by fold-change. **b** Flow cytometry analysis of the frequency of apoptotic (active caspase 3⁺) cells after incubation with or without cisplatin for 24 h. All experiments were performed in triplicate. The p-values by two-way ANOVA are indicated. The data represent the mean \pm SD. Source data are provided as a Source Data file. (* $P \leq 0.05$, ** $P \leq 0.01$, *** $P \leq 0.001$. NS, not significant)

[On page 9, 10, Result section of the revised manuscript]

We further investigated which stages of the autophagy pathway is crucial for EGF secretion and EGFR signaling-mediated cisplatin resistance by using two other autophagy inhibitors, spautin-1 and bafilomycin A1 (BafA). Spautin-1 is a specific inhibitor of the PtdIns3K catalytic subunit (PIK3C3/VPS34) responsible for autophagosome formation, whereas BafA is an inhibitor of V-ATPase, which blocks the fusion of autophagosomes to lysosomes⁴⁵. Interestingly, elevated EGF secretion and cisplatin resistance in CaSki CR cells was significantly reduced by treatment with spautin-1 but not with BafA (Fig. 2g, h and Supplementary Fig. 5), indicating that autophagosome formation is crucial for EGF

secretion and EGFR signaling-mediated cisplatin-resistant phenotype in CaSki CR cells, but late autophagy components involved in autophagosome-lysosome fusion appear to be dispensable in the process. Colinearly, compared to SiHa P cells, SiHa CR cells also showed increase in autophagosome formation and EGF secretion (Supplementary Fig. 6a). Notably, suppression of autophagosome formation by spautin-1 reduced EGF secretion, the level of pEGFR and cisplatin-resistant phenotype in SiHa CR cells (Supplementary Fig. 6a, b). Given these data, we conclude that autophagosome-mediated EGF secretion induces hyperactivation of EGFR signaling, thereby promoting resistance to cell death by cisplatin in cervical cancer cells.

3) NANOG activates TRPV1, which then stimulates Ca²⁺-AMPK, and then Autophagy to promote EGF secretion: the pathway seems random and unusual, and the support from the results is not convincing.

We understand the reviewer's concern that the molecular pathway that induces cisplatin resistance in cervical cancer cells is not convincing. Therefore, to remedy the weakness of our findings, we performed additional experiments. First, we assessed whether the NANOG-TRPV1-pEGFR axis-mediated cisplatin resistance is reproduced in additional cisplatin-resistant tumor model. To do this, **we used two different types of cisplatin-resistant cervical cancer cell line: SiHa CR and HeLa.** SiHa CR is newly established cisplatin-resistant tumor cells line, generated from SiHa P (a cisplatin-susceptible parental cell line of SiHa), and HeLa is commonly used in multiple drug resistance studies (Pan C et al., Nat Commun, 2021). Consistent with the results from CaSki CR cells, SiHa CR and HeLa displayed elevated autophagosome-mediated EGF secretion and EGFR signaling-dependent cisplatin resistance compared to control cells (Supplementary Fig. 2, 6 of the revised manuscript). In addition, we found that the NANOG-TRPV1-pEGFR axis was conserved, and TRPV1 inhibition successfully overcame the NANOG-mediated cisplatin resistance **in other cisplatin-resistant cervical cancer models** (Supplementary Fig. 7, 8 of the revised manuscript). Thus, these data indicate that **the NANOG-TRPV1-pEGFR axis is not a specific mechanism found only in one cisplatin-resistant tumor model.**

We next assessed whether TRPV1 potentiates the cisplatin resistance of NANOG⁺ tumor cells. To directly link TRPV1 to these phenotypes, **we used another NANOG-overexpressed cell line: HEK293 NANOG.** Consistent with the results from CaSki NANOG cells, TRPV1 inhibition using siRNA or inhibitor dampened the NANOG-mediated phenotypes (Supplementary Fig. 10 of the revised manuscript). Since previous reports suggested that TRP channels could promote autophagosomes formation and therapeutic resistance through the Ca²⁺-AMPK pathway (Arthm Kondratskyi et al., Front. Physiol. 2013), we downregulated CaMKK β expression to block Ca²⁺ signaling, and found that reduction of the NANOG-mediated phenotypes (Supplementary Fig. 11 of the revised manuscript). Thus, these results support our conclusion that **TRPV1-Ca²⁺-AMPK pathway plays a crucial role in secretory autophagy-dependent cisplatin resistance of NANOG⁺ tumor cells.**

Therefore, we believe that NANOG-TRPV1-pEGFR axis could function as a central molecular pathway to induce the cisplatin resistance in NANOG⁺ tumor cells. We have included these data and description in the revised manuscript as followed:

[On supplementary Fig. 2 of the revised manuscript]

[On page 5, 6, Supplementary Figure legend section of the revised manuscript]

Supplementary Fig. 2. The phenotypes of cisplatin-resistant tumor cells are critically dependent on EGFR signaling activation. **a** Flow cytometry analysis of the frequency of apoptotic (active caspase 3⁺) cells after treatment with or without cisplatin for 24 h. **b** The protein level of pEGFR, EGFR, and MCL1 expression were confirmed by western blots. **c** and **d** SiHa CR and HeLa cells were treated with gefitinib as the indicated dose. **e** and **f** SiHa CR and HeLa cells were transfected with siRNA

targeting *GFP* or *EGFR*. **c** and **e** The protein levels of pEGFR, EGFR and MCL1 were determined by western blots. β -actin was used as an internal loading control. Numbers below the blot images indicate the expression as measured by fold-change. **d** and **f** Flow cytometry analysis of the frequency of apoptotic (active caspase 3⁺) cells after incubation with or without cisplatin for 24 h. All experiments were performed in triplicate. The p-values by two-way ANOVA are indicated. The data represent the mean \pm SD. Source data are provided as a Source Data file. (*P \leq 0.05, **P \leq 0.01, ***P \leq 0.001. NS, not significant)

[On page 8, Results section of the revised manuscript]

To verify the reproducibility of the above results, we repeated experiments of Fig. 1c to 1g using two different types of cisplatin-resistant cervical cancer cell lines: SiHa CR and HeLa. SiHa CR is another newly established cisplatin-resistant tumor cell line, generated from SiHa P (a cisplatin-susceptible parental cell line of SiHa), and HeLa is commonly used in various drug resistance studies⁴². Consistent with the results from CaSki CR cells, SiHa CR displayed the resistance to cell death by cisplatin and the hyperactivated EGFR signaling compared to SiHa P cells (Supplementary Fig. 2a, b). Importantly, EGFR inhibition using gefitinib or *siEGFR* could reverse the cisplatin-resistant phenotypes of SiHa CR cells and HeLa cells (Supplementary Fig. 2c-f). Taken together, our data demonstrate that the hyperactivation of EGFR signaling pathway contributes to the resistance to cisplatin in cervical cancer cells.

[On supplementary Fig. 6 of the revised manuscript]

[On page 10, Supplementary Figure legend section of the revised manuscript]

Supplementary Fig. 6. Autophagosome formation is crucial for EGF secretion and EGFR signaling-mediated cisplatin resistance in SiHa CR cells. **a** In the cells treated with or without spautin-1, the protein levels of secreted EGF and internal LC3B, pEGFR, EGFR, and β -actin were

determined by western blots. β -actin was used as an internal loading control. Numbers below the blot images indicate the expression as measured by fold-change. **b** Flow cytometry analysis of the frequency of apoptotic (active caspase 3⁺) cells after incubation with or without cisplatin for 24 h. All experiments were performed in triplicate. The p-values by two-way ANOVA are indicated. The data represent the mean \pm SD. Source data are provided as a Source Data file. (* $P \leq 0.05$, ** $P \leq 0.01$, *** $P \leq 0.001$. NS, not significant)

[On page 10, Results section of the revised manuscript]

Colinearly, compared to SiHa P cells, SiHa CR cells also showed increase in autophagosome formation and EGF secretion (Supplementary Fig. 6a). In addition, the suppression of autophagosome formation by spautin-1 reduced EGF secretion, the level of pEGFR and cisplatin-resistant phenotypes in SiHa CR cells (Supplementary Fig. 6a, b). Given these data, we conclude that autophagosome-mediated EGF secretion induces the hyperactivation of EGFR signaling, thereby promoting chemoresistance to cell death by cisplatin in cervical cancer cells.

[On supplementary Fig. 7 of the revised manuscript]

[On page 11, Supplementary Figure legend section of the revised manuscript]

Supplementary Fig. 7. Knockdown of NANOG reduces cisplatin resistance by blocking autophagosome-mediated EGF secretion. **a** The protein levels of NANOG were determined by western blot. **b** and **c** SiHa CR or HeLa cells were transfected with siRNA targeting GFP or NANOG. **b** The levels of internal NANOG, LC3B, pEGFR, and EGFR and secreted EGF protein were confirmed by western blots. **c** Flow cytometry analysis of the frequency of apoptotic (active caspase 3⁺) cells after incubation with or without cisplatin for 24 h. **a** and **b** β -actin was used as an internal loading control. Numbers below the blot images indicate the expression as measured by fold-change. All experiments

were performed in triplicate. The p-values by two-way ANOVA are indicated. The data represent the mean \pm SD. Source data are provided as a Source Data file. (*P \leq 0.05, **P \leq 0.01, ***P \leq 0.001. NS, not significant)

[On page 10, 11, Results section of the revised manuscript]

Consistently, the increased expression of NANOG was also observed in another cisplatin-resistant tumor model (SiHa CR) (Supplementary Fig. 7a). Thus, these results demonstrate that cisplatin depletes tumor cells lacking NANOG while enriching tumor cells containing NANOG, suggesting that NANOG expression in tumor cells could confer a survival advantage under the selection pressure imposed by cisplatin.

To further investigate whether NANOG expression in tumor cells is responsible for the acquisition of cisplatin resistance by autophagy-mediated EGFR activation, we silenced the *NANOG* gene in cisplatin-resistant tumor cells using siRNAs. The knockdown of *NANOG* in the cisplatin-resistant tumor cells led to a significant decrease in autophagosome abundance, EGF secretion and the level of pEGFR (Fig. 3c and Supplementary Fig. 7b), suggesting an important role of NANOG in regulating the autophagy-mediated EGFR activation of cisplatin-resistant tumor cells. Notably, we found that *NANOG* knockdown reduced the resistance of cisplatin-resistant tumor cells to apoptotic cell death by cisplatin (Fig. 3d and Supplementary Fig. 7c).

[On supplementary Fig. 8 of the revised manuscript]

[On page 12, Supplementary Figure legend section of the revised manuscript]

Supplementary Fig. 8. TRPV1 inhibition reduces cisplatin resistance by blocking autophagosome-mediated EGF secretion. **a** The protein levels of TRPV1 were determined by western blot. **b** and **c** SiHa CR or HeLa cells were treated with DMSO or AMG9810. **b** The levels of

internal LC3B, pEGFR, and EGFR and secreted EGF protein were confirmed by western blots. **c** Flow cytometry analysis of the frequency of apoptotic (active caspase 3⁺) cells after incubation with or without cisplatin for 24 h. **a** and **b** β -actin was used as an internal loading control. Numbers below the blot images indicate the expression as measured by fold-change. All experiments were performed in triplicate. The p-values by two-way ANOVA are indicated. The data represent the mean \pm SD. Source data are provided as a Source Data file. (*P \leq 0.05, **P \leq 0.01, ***P \leq 0.001. NS, not significant)

[On page 12, Results section of the revised manuscript]

Interestingly, TRPV1 was significantly overexpressed in **cisplatin-resistant tumor** cells compared to **cisplatin-susceptible tumor** cells (Fig. 4b and **Supplementary Fig. 8a**). To determine whether TRPV1 is functionally active in these cells, we examined TRPV1-induced electrical currents upon exposure to its agonist, capsaicin. TRPV1-specific inward currents were elicited by capsaicin and were abolished by the TRPV1-specific antagonist AMG9810 in CaSki CR cells, but not in CaSki P cells (Fig. 4c). Previous studies reported that TRPV1-mediated intracellular Ca²⁺ influx induced autophagy by promoting autophagosome formation^{50, 51}. Interestingly, TRPV1 inhibition reduced autophagosome formation, thereby impeding autophagy-driven EGFR signaling (Fig. 4d) as well as increasing the sensitivity of CaSki CR cells to cisplatin (Fig. 4e). **Moreover, the above results were also reproduced in SiHa CR or HeLa cells (Supplementary Fig. 8b, c)**. Thus, our results suggest that TRPV1, which is functionally active as a Ca²⁺-permeable cation channel, **could confer** resistance to cisplatin by regulating **autophagosome formation** in cisplatin-resistant tumor cells.

[On supplementary Fig. 10 of the revised manuscript]

[On page 14, Supplementary Figure legend section of the revised manuscript]

Supplementary Fig. 10. TRPV1 is required for NANOG-mediated refractory phenotypes. **a** and **b** HEK293 NANOG cells were transfected with siRNA targeting *GFP* or *TRPV1*. **c** and **d** HEK293

NANOG cells were treated with DMSO or AMG9810. **a** and **c** The protein levels of TRPV1, LC3B, pEGFR, EGFR, pAKT, and AKT were analyzed by western blots. β -actin was used as an internal loading control. Numbers below the blot images indicate the expression as measured by fold-change. **b** and **d** Flow cytometry analysis of the frequency of apoptotic (active caspase 3⁺) cells after incubation with or without cisplatin for 24 h. All experiments were performed in triplicate. The p-values by two-way ANOVA are indicated. The data represent the mean \pm SD. Source data are provided as a Source Data file. (*P \leq 0.05, **P \leq 0.01, ***P \leq 0.001. NS, not significant)

[On page 13, Results section of the revised manuscript]

We then questioned whether TRPV1 potentiates the refractory phenotypes of NANOG⁺ tumor cells. To directly link TRPV1 to these phenotypes, we silenced TRPV1 expression in CaSki NANOG or HEK293 NANOG cells (Fig. 5a and Supplementary Fig. 10a). Notably, *TRPV1* knockdown decreased autophagy-mediated activation of EGFR-AKT pathway (Fig. 5a and Supplementary Fig. 10a) and increased sensitivity to cisplatin in CaSki NANOG and HEK293 NANOG cells (Fig. 5b and Supplementary Fig. 10b). Consistent with the above results, we found that the NANOG-mediated phenotypes were dampened by TRPV1 inhibition using AMG9810 (Fig. 5c, d and Supplementary Fig. 10c, d), indicating TRPV1 plays a crucial role in NANOG-mediated resistance to cisplatin.

[On supplementary Fig. 11 of the revised manuscript]

[On page 15, Supplementary Figure legend section of the revised manuscript]

Supplementary Fig. 11. Ca²⁺-AMPK pathway is required for NANOG-mediated EGFR activation. HEK293 NANOG cells were transfected siRNA targeting *GFP* or *CaMKKβ*. The levels of CaMKKβ, pAMPK, AMPK, LC3B, pEGFR, and EGFR were confirmed by western blot analysis. β -actin was used as an internal loading control. Numbers below the blot images indicate the expression as measured by fold-change. This experiment was performed in triplicate.

[On page 13, 14, Results section of the revised manuscript]

Indeed, intracellular Ca²⁺ signaling blockade by CaMKK β knockdown reduced the NANOG-mediated phenotypes (Fig. 5g, h and Supplementary Fig. 11). Taken together, our findings indicate that the TRPV1 channel plays an important role in the secretory autophagy-dependent refractory phenotypes of NANOG⁺ tumor cells.

4) The description of experiments/figures are overly brief in paces preventing proper evaluation. Fig. 1: (b) CESC cohort: no information? What is it? (c) p-EGFR P0 P3, 1.0, 1.0 ? (e) The frequency of apoptotic cells?

We appreciate the reviewer for pointing out our mistakes. We have added clinical information of CESC cohort (Supplementary table 2) and mentioned the source which the clinical data of CESC was obtained in the revised manuscript. In addition, we have corrected our mistakes, such as the error of quantification value below the western blot images and lack of experimental description, as followed:

[On page 13, Materials and Methods of the revised manuscript]

TCGA data collection and analysis

Gene expression data for CESC, LUAD, LUSC, STAD, and STES samples profiled by TCGA were collected from the Firehose BROAD GDAC data repository (<https://gdac.broadinstitute.org/>). Clinical data were also retrieved from the same source.

[On Fig. 1c, 5g, and 5h of the revised manuscript]

[On page 34, Figure legend section of the revised manuscript]

e and g Flow cytometry analysis of the frequency of apoptotic (active caspase 3⁺) cells after incubation with or without cisplatin for 24 h.

5) Many results present values of little differences? For an example: Fig. 1b: the values of the two groups are; largely overlapping.

We understand the reviewer's concern that the statistical difference in EGFR activity score between the two cohorts classified as having low or high chemotherapy-resistant gene signature might be due to the excessive values of a specific fraction of cohort. To address this point, we first investigated whether EGFR signaling is hyperactivated in cisplatin-resistant cervical cancer patients compared to cisplatin-sensitive cervical cancer patients. We newly analyzed the transcriptome data from CESC classified as responders (R, complete and partial response) or non-responders (NR, stable and progressive disease) to cisplatin. Consistent with the results in our original manuscript, we found that EGFR activity score were statistically higher NR than in R (Fig. 1b of the revised manuscript). However, as the results in our original manuscript, an overlap of two distributions was also observed (proportion of overlap=35.8%).

We next assessed whether the statistical significance between the two groups was determined by a specific fraction of cohort. To do this, we have performed two independent non-parametric statistical tests, such as the Wilcoxon Rank sum test and Kolmogorov-Smirnov test. Interestingly, we found that the Wilcoxon Rank sum test result (p-value=0.018) clearly indicated statistically significant differences in the mean of the two groups (Fig. 1a for the reviewer only). Additionally, two samples Kolmogorov-Smirnov test result (p-value=0.025) rejected the null hypothesis that the two groups were drawn from the same distribution, indicating statistically differences in the distribution of the two groups (Fig. 1b for the reviewer only). Thus, these results indicate that **the statistical significance between the two groups is determined by the differences in the mean and distribution of the two groups rather than the value of a specific fraction of cohort.**

We appreciate the reviewer's this helpful suggestion.

[Fig. 1 for the reviewer only]

Wilcoxon test result

```
Wilcoxon rank sum test with continuity correction
data: RSEGFR.activity_Mo and NRSEGFR.activity_Mo
W = 185, p-value = 0.01814
alternative hypothesis: true location shift is not equal to 0
```

Two sample kolmogorow-smirnow test

```
> ks.test(RSEGFR.activity_Mo, NRSEGFR.activity_Mo, alternative = 'greater')
Exact two-sample Kolmogorov-Smirnov test
data: RSEGFR.activity_Mo and NRSEGFR.activity_Mo
D* = 0.41212, p-value = 0.02526
alternative hypothesis: the CDF of x lies above that of y
Two samples are from different distribution
```

Reviewer #3

The authors have shown that cisplatin resistance is highly associated with hyperactivation of EGFR on tumor cells and that EGFR hyperactivation rises by a transcriptional increase in TRPV1 via NANOG. In addition, the authors have shown that TRPV1 promoted autophagy-mediated EGF secretion with calcium influx, which activated EGFR-AKT signaling pathways. Furthermore, the authors have shown that TRPV1 inhibition using AMG9810 increased the effectiveness of cisplatin treatment.

I enjoyed reading the rationale and the introductory part of the manuscript. If there is a way to inhibit tumor growth and, at the same time, alleviate the chemotherapy-induced neuropathic pain, that will help the patients, as that is one of the major limitations of cisplatin treatment.

However, there are several places in the manuscript that need further clarification to improve the paper's overall significance.

1) Define secretory autophagy and degradative autophagy. It was unclear what the difference was and how it promoted EGF secretion by NANOG. The papers have many references to previous work, but it is not clear how it ties all together to the context of this.

We appreciate the reviewer's this insightful comment. The proteins secreted through so-called "conventional mechanisms" are characterized by the presence of an N-terminal signaling peptide, needed for access to the endoplasmic reticulum and the Golgi apparatus, required for further secretion. However, some relevant cytosolic proteins, lack of this signal peptides, are secreted by different unconventional or "non-canonical" processes. One form of this unconventional secretion is named secretory autophagy because it is specifically associated with the autophagy process. It is defined by ATG proteins that regulate the biogenesis of the autophagosome, its representative organelle. The canonical degradative autophagy involves the fusion of the autophagosomes with lysosomes for content degradation, **whereas the secretory autophagy bypasses this degradative process to allow the secretion** (Caludio Daniel Gonzalez et al., Front. Endocrinol. 2020). Therefore, **it is possible to experimentally distinguish the secretory autophagy and degradative autophagy by using different inhibitors that block the specific each step of autophagy process.**

To address the reviewer's comment, we further investigated which stages of the autophagy pathway is crucial for EGF secretion and EGFR signaling-mediated cisplatin resistance by using two other autophagy inhibitors, spautin-1 and bafilomycin A1 (BafA). Spautin-1 is a specific inhibitor of the PtdIns3K catalytic subunit (PIK3C3/VPS34) responsible for autophagosome formation, whereas BafA is an inhibitor V-ATPase which blocks the fusion of autophagosomes to lysosomes. Interestingly, increased EGF secretion, pEGFR level and cisplatin-resistant phenotypes in CaSki CR cells were strongly suppressed by treatment with spautin-1 but not with BafA (Fig. 2g, h and Supplementary Fig. 5 of the revised manuscript). These results suggest that **autophagosome formation is important for EGF secretion and cisplatin resistance in CaSki CR**

cells, but late autophagy components involved in autophagosome lysosome fusion appear to be dispensable in the process. Notably, the above results were reproduced in newly established cisplatin-resistant cervical cancer model (SiHa CR) (Supplementary Fig. 6 of the revised manuscript). These data support our hypothesis that autophagosome-mediated EGF secretion induces hyperactivation of EGFR signaling, thereby promoting resistance to cell death by cisplatin in cervical cancer cells.

We appreciate the invaluable and helpful comment that improves the quality of our manuscript. We have included these data and description in the revised manuscript as followed:

[On Fig. 2g, h and Supplementary Fig. 5 of the revised manuscript]

[On page 35, 36, Figure Legend section of the revised manuscript]

g In the cells treated with or without spautin-1, the protein levels of secreted EGF and internal LC3B, pEGFR, EGFR, and β -actin were determined by western blots. **h** Flow cytometry analysis of the frequency of apoptotic (active caspase 3⁺) cells after incubation with or without cisplatin for 24 h. **b**, **d** and **g** β -actin was used as the internal loading control.

[On page 9, Supplementary Figure Legend section of the revised manuscript]

Supplementary Fig. 5. Increased EGF secretion in cisplatin-resistant tumor cells is not affected by treatment with BafA. In the cells treated with or without BafA, the protein levels of secreted EGF and internal LC3B, pEGFR, EGFR, and β -actin were determined by western blots. β -actin was used as an internal loading control. Numbers below the blot images indicate the expression as measured by fold-change. This experiment was performed in triplicate.

[On Supplementary Fig. 6 of the revised manuscript]

[On page 10, Supplementary Figure legend section of the revised manuscript]

Supplementary Fig. 6. Autophagosome formation is crucial for EGF secretion and EGFR signaling-mediated cisplatin resistance in SiHa CR cells. **a** In the cells treated with or without spautin-1, the protein levels of secreted EGF and internal LC3B, pEGFR, EGFR, and β -actin were determined by western blots. β -actin was used as an internal loading control. Numbers below the blot images indicate the expression as measured by fold-change. **b** Flow cytometry analysis of the frequency of apoptotic (active caspase 3⁺) cells after incubation with or without cisplatin for 24 h. All experiments were performed in triplicate. The p-values by two-way ANOVA are indicated. The data represent the mean \pm SD. Source data are provided as a Source Data file. (* $P \leq 0.05$, ** $P \leq 0.01$, *** $P \leq 0.001$. NS, not significant)

[On page 9, 10, Result section of the revised manuscript]

We further investigated which stages of the autophagy pathway is crucial for EGF secretion and EGFR signaling-mediated cisplatin resistance by using two other autophagy inhibitors, spautin-1 and bafilomycin A1 (BafA). Spautin-1 is a specific inhibitor of the PtdIns3K catalytic subunit (PIK3C3/VPS34) responsible for autophagosome formation, whereas BafA is an inhibitor of V-ATPase, which blocks the fusion of autophagosomes to lysosomes⁴⁵. Interestingly, elevated EGF secretion and cisplatin resistance in CaSki CR cells was significantly reduced by treatment with spautin-1 but not with BafA (Fig. 2g, h and Supplementary Fig. 5), indicating that autophagosome formation is crucial for EGF secretion and EGFR signaling-mediated cisplatin-resistant phenotype in CaSki CR cells, but late autophagy components involved in autophagosome-lysosome fusion appear to be dispensable in the process. Colinearly, compared to SiHa P cells, SiHa CR cells also showed increase in autophagosome formation and EGF secretion (Supplementary Fig. 6a). In addition, suppression of autophagosome formation by spautin-1 reduced EGF secretion, the level of pEGFR and cisplatin-resistant phenotype in SiHa CR cells (Supplementary Fig. 6a, b). Given these data, we conclude that autophagosome-

mediated EGF secretion induces hyperactivation of EGFR signaling, thereby promoting resistance to cell death by cisplatin in cervical cancer cells.

2) How are low and high chemotherapy-resistant tumors defined? What is the cut-off mark? The gene signatures were generated from the breast cancer model. Are the same gene signatures apply to cervical?

We understand the reviewer's concern that the CR signature generated from the breast cancer model could predict responsiveness to cisplatin of cervical cancer patients. Although the CR signature was generated from the breast cancer model, it could also be applied to cervical cancer because the genes constituting the CR signature could be expressed in other tissue including the cervix.

Nevertheless, we agree with the reviewer's comment that it is necessary to verify that the CR signature is applicable in cervical cancer patients. To address the reviewer's comment, **we newly analyzed the transcriptome data from CESC classified as responders (R, complete and partial response) or non-responders (NR, stable and progressive disease) to cisplatin.** The CESC patient's information is retrieved from the Firehose BROAD GDAC data repository (<http://gdac.broadinstitute.org/>) and have been summarized in Supplementary Table 2 of the revised manuscript. **Importantly, we found that the CR signature was statistically increased in NR compared to R (Supplementary Fig. 1a of the revised manuscript).** Thus, these results indicate that **although the CR signature was generated from the breast cancer model, it could predict responsiveness to cisplatin in cervical cancer patients and serve as an indicator for discriminating cisplatin-resistant patients.**

To avoid this confusion and clarify that the hyperactivated EGFR signaling is associated with responsiveness to cisplatin in cervical cancer patients, we have replaced the old transcriptome data which classified as low levels or high levels of CR signature expression with the new one which classified as R or NR to cisplatin.

Additionally, we previously predicted responsiveness to cisplatin of cervical cancer patients in low or high levels of CR signature expression. The 25th and 75th percentiles were used as cutoffs values for the CR signature expression (CR signature low < 25th; CR signature high > 75th).

We appreciate the invaluable and helpful comment that improves the quality of our manuscript. We have included these data and description in the revised manuscript as followed:

[On Supplementary Fig. 1a of the revised manuscript]

[On page 4, Supplementary Figure legend section of the revised manuscript]

Supplementary Fig. 1. EGFR signaling is associated with chemotherapy resistance in cervical cancer patients. **a** Comparison of the level of chemotherapy-resistant (CR) gene signature in responder (R, $n = 55$) and non-responder (NR, $n = 12$) of cisplatin-treated CESC cohort. **b**. Correlation between expression level of CR gene signature and EGFR activity score in CESC cohort. **c**. Comparison of the level of EGFR activity scores in the CESC cohort with low levels (low, $n = 90$) and high levels (high, $n = 90$) of CR gene signature. The 30th and 70th percentiles were used as cutoffs values for the CR gene signature ($\text{CR gene signature}^{\text{high}} > 70\text{th}$; $\text{CR gene signature}^{\text{low}} < 30\text{th}$). The two-tailed p-values and correlation were assessed using the unpaired Student's t test and the Pearson's correlation. The data represent the mean \pm SD. Source data are provided as a Source Data file. (* $P \leq 0.05$, ** $P \leq 0.01$, *** $P \leq 0.001$. NS, not significant)

[On Fig. 1b of the revised manuscript]

[On page 33, 34, Figure legend section of the revised manuscript]

b Comparison of the level of EGFR activity scores in responder (R, $n = 55$) and non-responder (NR, $n = 12$) of cisplatin-treated CESC cohort.

[On page 7, Result section of the revised manuscript]

We further questioned whether hyperactivated EGFR signaling could confer resistance to cisplatin-mediated chemotherapy. By analyzing the EGFR activity scores in transcriptomic data on patients with TCGA cervical cancer patients classified as responders (R, complete or partial response) or non-responders (NR, stable or progressive disease) to cisplatin treatment, we found that EGFR activity scores were statistically higher in NR than in R (Fig. 1b). We also found that the above results were reproduced in TCGA cervical cancer patients classified as having low or high chemotherapy-resistant (CR) gene signature⁴⁰ (Supplementary Fig. 1). These results suggest that hyperactivated EGFR signaling may drive resistance to cisplatin-mediated chemotherapy, thereby leading to poor clinical outcomes in patients with cervical cancer.

3) Please elaborate on why LC3B-I and LC3B-II are tested and how they are different. In some setting, the upper band (LC3B-I) is used to come up with a conclusion, but in some settings, it is the lower band (LC3B-II) that is used (i.e. Fig 5c).

We appreciate the reviewer's insightful comment. The microtubule-associated protein light chain 3B (LC3B) is now widely used to monitor autophagy. It is known to exist in two forms: LC3B-I, which is found in the cytoplasm, and LC3B-II, which is membrane-bound and is converted from LC3B-I (termed LC3B lipidation) to initiate formation and lengthening of the autophagosome. **Because the amount of LC3B-II is clearly correlated with the number of autophagosomes, the detection of LC3B conversion (LC3B-I to LC3B-II) by immunoblot analysis is widely used to monitor autophagy.**

Previous studies demonstrated that depending on the serum concentration or incubation time, autophagosome abundance represented by the lipidated form (LC3B-II) of LC3B could be different in the same cell lines (Noboru Mizuhima et al., Autophagy, 2007, Suyeon Kim et al., Autophagy 2020). To confirm the reviewer's comment, we newly performed the experiments to assess the LC3B change by TRPV1 inhibition in CaSki *NANOG* cells. As in LC3B image of other experiments in the revised manuscript, inhibition of TRPV1 in CaSki *NANOG* cells mainly reduced the lipidated form (LC3B-II) of LC3B. **It seems that the difference of LC3B image shown Fig. 5c of our original manuscript was due to do not meet the sufficient serum-starvation condition unlike other experimental condition.**

Therefore, to help the reader's understanding and correct our mistake, we have revised the result section to elaborate on why LC3B-I and LC3B-II are tested, and changed the old LC3B western

blot image with the new one in the revised manuscript as followed:

[On page 8, 9, Result section of the revised manuscript]

Therefore, we hypothesized that autophagy-mediated EGF secretion might induce cisplatin resistance by activating EGFR signaling in cervical cancer. We first confirmed the level of autophagosome formation in cisplatin-resistant tumor cells. We found that autophagosome abundance was significantly increased in CaSki CR cells compared to CaSki P cells (Fig. 2a). Because the amount of LC3B-II (lipidated form of LC3B-I) is clearly correlated with the number of autophagosome, we also assessed the absolute level of LC3B-II (normalized by β -actin) by immunoblotting. Relative to CaSki P cells, CaSki CR cells exhibited escalated LC3B-II levels (Fig. 2b).

[On Fig. 5C of the revised manuscript]

4) The authors used a sub-section titled “TRPV1 promotes NANOG-mediated resistance to cisplatin through the Ca²⁺-AMPK pathway”; however, they have not tested any effect of TRPV1 overexpression in sensitive cells. It is not the resistance, but the sensitivity is being tested here.

We agree with the reviewer’s opinion. As the reviewer suggested, we performed additional experiments to assess the role of TRPV1 in cisplatin resistance. To do this, we transfected the *TRPV1* gene in cisplatin-susceptible CaSki P cells. The overexpression of TRPV1 in CaSki P cells led to a significant increase in autophagosome abundance, EGF secretion, the level of pEGFR as well as resistance to cisplatin (Fig. 7a-c of the revised manuscript). Interestingly, we could not observe any changes in the levels of NANOG upon TRPV1 overexpression (Supplementary Fig. 12 of the revised manuscript). These results suggest that ectopic expression of TRPV1 could induce the chemoresistance of tumor cells to cisplatin.

Furthermore, we found that these increased phenotypes mediated by TRPV1 were reversed

upon inhibition of TRPV1 channel activity (Fig. 7a-c of the revised manuscript). Thus, our results indicate that an important role of TRPV1 channel activity on those properties.

Notably, *TRPV1*-transfected cells had markedly higher levels of EGF secretion, compared with empty vector-transfected cells or AMG9810-treated CaSki *TRPV1* cells (Fig. 7c of the revised manuscript). The neutralization of secreted EGF by antibody led to a significant decrease in the levels of pEGFR in CaSki *TRPV1* cells (Fig. 7d of the revised manuscript), suggesting a direct role for EGF in the TRPV1-mediated EGFR signal pathway. Furthermore, EGFR knockdown reduced the resistance of TRPV1-transfected tumor cells to cisplatin (Fig. 7e, f of the revised manuscript). Thus, our results indicate that TRPV1 by itself was sufficient to induce resistance to cisplatin of tumor cells via EGF-EGFR signaling axis.

We have included these data and description in the revised manuscript as followed:

[On Fig. 7 of the revised manuscript]

[On page 44, Figure legend section of the revised manuscript]

Fig. 7. TRPV1 promotes cisplatin resistance through EGF-EGFR axis. a-c CaSki *no insert* and *TRPV1* cells were treated with DMSO or AMG9810, as indicated. a The protein levels of TRPV1, LC3B, pEGFR, and EGFR were confirmed by western blots. b The frequency of apoptotic (active

caspace 3⁺) cells were analyzed by flow cytometry. **c** The amount of EGF secreted into the media was measured by ELISA. **d** CaSki *TRPV1* cells were treated with IgG or anti-EGF. Levels of pEGFR and EGFR were confirmed by western blots. **e** and **f** CaSki *TRPV1* cells were transfected with siRNA targeting *GFP* or *EGFR*. **e** The protein levels of pEGFR and EGFR were confirmed by western blot analysis. **f** Flow cytometry analysis of the frequency of apoptotic (active caspase 3⁺) cells after incubation with or without cisplatin for 24 h. **a**, **d**, and **e** β -actin was used as the internal loading control. Numbers below the blot images indicate the expression as measured by fold-change. All experiments were performed in triplicate. The p-values by two-way ANOVA (**b** and **f**) or one-way ANOVA (**c**) are indicated. The data represent the mean \pm SD. Source data are provided as a Source Data file. (*P \leq 0.05, **P \leq 0.01, ***P \leq 0.001. NS, not significant)

[On Supplementary Fig. 12 of the revised manuscript]

[On page 16, Supplementary Figure legend section of the revised manuscript]

Supplementary Fig. 12. TRPV1 overexpression does not affect levels of NANOG. CaSki P cells were transfected with empty vector (*no insert*) or *TRPV1*. The levels of NANOG protein were confirmed by western blots. β -actin was used as an internal loading control. Numbers below the blot images indicate the expression as measured by fold-change. This experiment was performed in triplicate.

[On page 14, 15, Result section of the revised manuscript]

TRPV1 promotes resistance to cell death by cisplatin via EGF-EGFR signaling pathway

Given the important role of TRPV1 in cisplatin-resistant phenotypes of NANOG⁺ tumor cells, we questioned that TRPV1 expression alone could induce these phenotypes. We found that transfection of CaSki P cells with *TRPV1* not only elevated autophagosome formation shown by the markedly increased LC3B-I to LC3B-II transition but also induced phosphorylation of EGFR (Fig. 7a). Furthermore, we found that *TRPV1* overexpression in CaSki P cells induced a resistance to apoptotic cell death by cisplatin (Fig. 7b). Interestingly, we could not observe any changes in the levels of NANOG upon *TRPV1* overexpression (Supplementary Fig. 12). These results suggest that ectopic expression of TRPV1 could induce the chemoresistance of tumor cells to cisplatin.

Notably, the inhibition of TRPV1 activity using AMG9810 failed to enhance the EGF secretion, activation of EGFR signaling as well as cisplatin-resistant phenotypes (Fig. 7a-c). These data indicate a key role of TRPV1 channel activity on those properties. Importantly, *TRPV1*-transfected cells had markedly higher levels of EGF secretion, compared with empty vector-transfected cells or AMG9810-treated CaSki *TRPV1* cells (Fig. 7c). The neutralization of secreted EGF by antibody led to a significant decrease in the levels of pEGFR in CaSki *TRPV1* cells (Fig. 7d), suggesting a direct role for EGF in the TRPV1-mediated EGFR signaling pathway. Furthermore, EGFR knockdown reduced the resistance of TRPV1-transfected tumor cells to cisplatin (Fig. 7e, f). Thus, our results indicate that TRPV1 by itself was sufficient to induce the resistance to cisplatin of tumor cells via EGF-EGFR signaling pathway.

5) The authors concluded that TRPV1 inhibition could reverse the cisplatin resistance (Figure 7). However, no experiment was done to directly prove that. To test reversal, the tumors have to form first, and then treat the tumors with TRPV1 inhibitors to reverse, meaning to shrink the tumor. What was actually done is the prevention of tumor progression, not the reversal of cisplatin resistance. The title is, therefore, misleading.

We agree with the reviewer's comment that our findings in this study did not meet the conclusion which TRPV1 inhibition could reverse the cisplatin resistance.

To avoid the confusion, we have replaced "reverse" with "overcome" and have tone downed the expression that could cause misunderstandings as followed:

[On page 1, Title of the revised manuscript]

TRPV1 inhibition **overcomes cisplatin resistance by blocking autophagy-mediated hyperactivation of EGFR signaling pathway**

[On page 6, Introduction section of the revised manuscript]

Thus, we provide proof-of-concept evidence that TRPV1 inhibition **could be a plausible** strategy to control NANOG^{high} cisplatin-resistant cancer as well as CIPN.

[On page 16, Result section of the revised manuscript]

TRPV1 inhibition **sensitizes NANOG^{high} tumor cells to cisplatin**

Given our observation in vitro, we reasoned that the in vivo administration of TRPV1 inhibitor, AMG9810, should overcome resistance to cisplatin.

[On page 16, Results section of the revised manuscript]

Given these data, we concluded that AMG9810 could be a plausible therapeutic strategy to overcome the cisplatin-resistant phenotypes of NANOG^{high} tumor cells.

[On page 17, Result section of the revised manuscript]

Taken together, we conclude that TRPV1 inhibition could overcome the cisplatin resistance of tumor cells and represent an attractive strategy for the control of human cancer as a synergistic agent with cisplatin treatment.

[On page 19, Discussion section of the revised manuscript]

Thus, we provide proof-of-concept evidence that TRPV1 inhibition could be a clinically available strategy to control NANOG^{high} cisplatin-resistant cancer and CIPN.

6) *Minor comments:*

6-1) *Figure 1e. What is the concentration of cisplatin?*

We thank the reviewer's helpful comment. To help the reader's understanding, we have marked the concentration of all chemicals we used in Figure section of revised manuscript as followed:

[On Fig. 1e, g of the revised manuscript]

6-2) *Please indicate in the Methods section how the cisplatin was dissolved. Depending on the solvent, the efficacy of cisplatin changes and will be beneficial to the readers.*

We agree with the reviewer's opinion. As the reviewer suggested, we have additionally indicated the solvent of chemicals in revised manuscript as followed:

[On page 20, Materials and Methods section of the revised manuscript]

Chemical reagents

The following chemical reagents were used in this study: AMG9810 (Sigma-Aldrich, St. Louis, MO, USA, A2731), capsaicin (Sigma-Aldrich, M2028), gefitinib (Selleckchem, Houston, TX, USA, S1025), cisplatin (Selleckchem, S1166), spautin-1 (Selleckchem, S78880), and bafilomycin A1 (Sigma-Aldrich, B1793-2UG). **AMG9810, gefitinib, cisplatin, spautin-1, or bafilomycin A1 were dissolved in dimethyl sulfoxide (DMSO). Capsaicin was dissolved in ethanol.**

[On page 25, Materials and Methods section of the revised manuscript]

Tumor treatment experiments

NOD/SCID mice were inoculated subcutaneously with 1×10^6 CaSki CR cells per mouse. Nine days following the tumor challenge, AMG9810 (0.05 mg/kg)-loaded chitosan hydrogel or cisplatin (2.5 mg/kg) was administered intratumorally or intraperitoneally, respectively. **The cisplatin is prepared in DMSO, diluted in phosphate buffered saline (PBS).**

6-3) Figure 1g. Data for *siEGFR #2* is missing.

We appreciate the reviewer for pointing out our mistakes. To correct our mistake, we have added data for *siEGFR #2* in the revised manuscript as followed:

[On Fig. 1g of the revised manuscript]

6-4) Figure 4. The authors indicate that “NANOG MT had no profound impact on either TRPV1 mRNA or protein levels”, but it looks significantly downregulated. I do not see bands for TRPV1 in NANOG MT in HEK293 cells, but the value says 1.0. I think there is an error here.

The reviewer raises an important comment that NANOG MT might reduce TRPV1 expression in HEK293 cells. We apologize for the confusion mediated by our mistakes. To confirm the effects of NANOG MT on TRPV1 expression, we performed additional experiments. As shown below in Fig. 4h of the revised manuscript, NANOG MT did not increase TRPV1 expression in HEK293 cells as that much in CaSki cells at this time. We recognized that previous TRPV1 western blot image contained black dots around the TRPV1 band after taking a closer look at it. We have thought that those dot images could make confusion to reader.

Therefore, to correct our mistake, we have changed the old TRPV1 blot image with the new one in the revised manuscript as followed:

[On Fig. 4h of the revised manuscript]

6-5) Figure 5g. Please correct the wrong values.

We apologize for our mistakes. To correct it, we have revised the quantification value of the bands in the revised manuscript as followed:

[On Fig. 5g of the revised manuscript]

6-6) Figure 6. The sentence “The high expression of NANOG and TRPV1 was correlated with chemoradiation resistant tumors, whereas the high expression of pEGFR tended to correlate with chemoradiation resistance” is confusing. Although Fig 6b shows that it is not significant, the immunohistochemical staining looks significant. Could you please clarify the discrepancies in the conclusion?

We appreciate the reviewer’s comment and apologize for the confusion. To avoid the confusion, we have replaced the old pEGFR IHC image with the new one that could be representative of the mean of pEGFR IHC score in the revised manuscript as followed:

[On Fig. 6a of the revised manuscript]

6-7) Supplementary Figure 5 is confusing. ABCC5 is usually upregulated in cisplatin-resistant cells but it was downregulated in P3. What was the rationale for silencing TRPV1 in P3 cells that has low ABCC5 and ABCC6 transporters?

We thank the reviewer for pointing out our mistakes and apologize for the confusion. While we took a close look at the data of Supplementary Fig. 5a cautiously in our original manuscript, we recognized that the ratio of relative mRNA expression was wrongly calculated. Therefore, we repeated the experiment to assess ABCC2, ABCC5, and ABCC6 expression in cisplatin-sensitive and -resistant cells again. As shown below in Supplementary Fig. 17a of the revised manuscript, we found that ABCC5

and ABCC6 was overexpressed in CaSki CR cells compared to CaSki P cells, and confirmed the ratio of relative ABCC5 and ABCC 6 mRNA expression has been reproduced consistently.

To correct our mistake, we have changed the previously performed data with the new one in the revised manuscript as followed:

[On Supplementary Fig. 17a of the revised manuscript]

Reviewer #4

Oh et al., have submitted a manuscript entitled “TRPV1 inhibition reverses cisplatin resistance by blocking autophagy-mediated hyperactivation of EGFR signal pathway” for consideration of publication. This study describes a novel mechanism where EGFR is activated through an autophagy-mediated pathway leading to cisplatin resistance in cervical cancers. Even though the evidence in the cisplatin-resistant cell line is convincing, there is a lack of subsequent results supporting this mechanism in cervical cancer. Thus, there are not enough results presented to warrant publication. List below are my specific comments.

1) The generation of a cisplatin resistant cell line (P3) gives evidence that EGFR activation and Mcl-1 levels are increased. This needs to be demonstrated in at least two independent cervical cancer cells. It might just be a selection for cisplatin resistant specific to only this cell line.

We agree with the reviewer’s comment that our findings in this study need to be reproduced in additional cisplatin-resistant tumor models. To do this, **we performed additional experiments using two different types of cisplatin-resistant cervical cancer cell lines: SiHa CR and HeLa.** SiHa CR is a newly established cisplatin-resistant tumor cell line, generated from SiHa P (a cisplatin-susceptible parental cell line of SiHa), and HeLa is commonly used in multiple drug resistance studies (Pan C et al., Nat Commun, 2021). Consistent with the results from CaSki CR cells, SiHa CR and HeLa displayed elevated autophagosome-mediated EGF secretion and EGFR signaling-dependent cisplatin resistance compared to control cells (Supplementary Fig. 2, 6 of the revised manuscript).

Importantly, we found that the NANOG-TRPV1-pEGFR axis was conserved, and TRPV1 inhibition successfully overcame the NANOG-mediated cisplatin resistance **in other cisplatin-resistant cervical cancer models** (Supplementary Fig. 7, 8 of the revised manuscript). Thus, these data support our conclusion that the NANOG-TRPV1-pEGFR axis could function as a central molecular pathway that confers resistance of tumor cells to cisplatin.

We appreciate the invaluable comment that has helped us improve the quality of our manuscript. We have included additional data and description in the revised manuscript as follows:

[On supplementary Fig. 2 of the revised manuscript]

[On page 5, 6, Supplementary Figure legend section of the revised manuscript]

Supplementary Fig. 2. The phenotypes of cisplatin-resistant tumor cells are critically dependent on EGFR signaling activation. **a** Flow cytometry analysis of the frequency of apoptotic (active caspase 3⁺) cells after treatment with or without cisplatin for 24 h. **b** The protein level of pEGFR, EGFR, and MCL1 expression were confirmed by western blots. **c** and **d** SiHa CR and HeLa cells were treated with gefitinib as the indicated dose. **e** and **f** SiHa CR and HeLa cells were transfected with siRNA targeting *GFP* or *EGFR*. **c** and **e** The protein levels of pEGFR, EGFR and MCL1 were determined by western blots. β-actin was used as an internal loading control. Numbers below the blot images indicate the expression as measured by fold-change. **d** and **f** Flow cytometry analysis of the frequency of apoptotic (active caspase 3⁺) cells after incubation with or without cisplatin for 24 h. All experiments were performed in triplicate. The p-values by two-way ANOVA are indicated. The data represent the mean ± SD. Source data are provided as a Source Data file. (*P ≤ 0.05, **P ≤ 0.01, ***P ≤ 0.001. NS, not significant)

[On page 8, Results section of the revised manuscript]

To verify the reproducibility of the above results, we repeated experiments of Fig. 1c to 1g using two different types of cisplatin-resistant cervical cancer cell lines: SiHa CR and HeLa. SiHa CR is another newly established cisplatin-resistant tumor cell line, generated from SiHa P (a cisplatin-susceptible parental cell line of SiHa), and HeLa is commonly used in various drug resistance studies⁴². Consistent with the results from CaSki CR cells, SiHa CR displayed the resistance to cell death by

cisplatin and the hyperactivated EGFR signaling compared to SiHa P cells (Supplementary Fig. 2a, b). Importantly, EGFR inhibition using gefitinib or *siEGFR* could reverse the cisplatin-resistant phenotypes of SiHa CR cells and HeLa cells (Supplementary Fig. 2c-f). Taken together, our data demonstrate that the hyperactivation of EGFR signaling pathway contributes to the resistance to cisplatin in cervical cancer cells.

[On supplementary Fig. 6 of the revised manuscript]

[On page 10, Supplementary Figure legend section of the revised manuscript]

Supplementary Fig. 6. Autophagosome formation is crucial for EGF secretion and EGFR signaling-mediated cisplatin resistance in SiHa CR cells. **a** In the cells treated with or without spautin-1, the protein levels of secreted EGF and internal LC3B, pEGFR, EGFR, and β-actin were determined by western blots. β-actin was used as an internal loading control. Numbers below the blot images indicate the expression as measured by fold-change. **b** Flow cytometry analysis of the frequency of apoptotic (active caspase 3⁺) cells after incubation with or without cisplatin for 24 h. All experiments were performed in triplicate. The p-values by two-way ANOVA are indicated. The data represent the mean ± SD. Source data are provided as a Source Data file. (*P ≤ 0.05, **P ≤ 0.01, ***P ≤ 0.001. NS, not significant)

[On page 10, Results section of the revised manuscript]

Concurrently, compared to SiHa P cells, SiHa CR cells also showed increase in autophagosome formation and EGF secretion (Supplementary Fig. 6a). In addition, the suppression of autophagosome formation by spautin-1 reduced EGF secretion, the level of pEGFR and cisplatin-resistant phenotypes in SiHa CR cells (Supplementary Fig. 6a, b). Given these data, we conclude that autophagosome-mediated EGF secretion induces the hyperactivation of EGFR signaling, thereby promoting chemoresistance to cell death by cisplatin in cervical cancer cells.

[On supplementary Fig. 7 of the revised manuscript]

[On page 11, Supplementary Figure legend section of the revised manuscript]

Supplementary Fig. 7. Knockdown of *NANOG* reduces cisplatin resistance by blocking autophagosome-mediated EGF secretion. **a** The protein levels of *NANOG* were determined by western blot. **b** and **c** SiHa CR or HeLa cells were transfected with siRNA targeting *GFP* or *NANOG*. **b** The levels of internal *NANOG*, LC3B, pEGFR, and EGFR and secreted EGF protein were confirmed by western blots. **c** Flow cytometry analysis of the frequency of apoptotic (active caspase 3⁺) cells after incubation with or without cisplatin for 24 h. **a** and **b** β-actin was used as an internal loading control. Numbers below the blot images indicate the expression as measured by fold-change. All experiments were performed in triplicate. The p-values by two-way ANOVA are indicated. The data represent the mean ± SD. Source data are provided as a Source Data file. (*P ≤ 0.05, **P ≤ 0.01, ***P ≤ 0.001. NS, not significant)

[On page 10, 11, Results section of the revised manuscript]

Consistently, the increased expression of *NANOG* was also observed in another cisplatin-resistant tumor model (SiHa CR) (Supplementary Fig. 7a). Thus, these results demonstrate that cisplatin depletes tumor cells lacking *NANOG* while enriching tumor cells containing *NANOG*, suggesting that *NANOG* expression in tumor cells could confer a survival advantage under the selection pressure imposed by cisplatin.

To further investigate whether *NANOG* expression in tumor cells is responsible for the acquisition of cisplatin resistance by autophagy-mediated EGFR activation, we silenced the *NANOG* gene in cisplatin-resistant tumor cells using siRNAs. The knockdown of *NANOG* in the cisplatin-resistant tumor cells led to a significant decrease in autophagosome abundance, EGF secretion and the level of

pEGFR (Fig. 3c and Supplementary Fig. 7b), suggesting an important role of NANOG in regulating the autophagy-mediated EGFR activation of **cisplatin-resistant tumor cells**. Notably, we found that *NANOG* knockdown reduced the resistance of **cisplatin-resistant tumor cells** to apoptotic cell death by cisplatin (Fig. 3d and Supplementary Fig. 7c).

[On supplementary Fig. 8 of the revised manuscript]

[On page 12, Supplementary Figure legend section of the revised manuscript]

Supplementary Fig. 8. TRPV1 inhibition reduces cisplatin resistance by blocking autophagosome-mediated EGF secretion. **a** The protein levels of TRPV1 were determined by western blot. **b** and **c** SiHa CR or HeLa cells were treated with DMSO or AMG9810. **b** The levels of internal LC3B, pEGFR, and EGFR and secreted EGF protein were confirmed by western blots. **c** Flow cytometry analysis of the frequency of apoptotic (active caspase 3⁺) cells after incubation with or without cisplatin for 24 h. **a** and **b** β-actin was used as an internal loading control. Numbers below the blot images indicate the expression as measured by fold-change. All experiments were performed in triplicate. The p-values by two-way ANOVA are indicated. The data represent the mean ± SD. Source data are provided as a Source Data file. (*P ≤ 0.05, **P ≤ 0.01, ***P ≤ 0.001. NS, not significant)

[On page 12, Results section of the revised manuscript]

Interestingly, TRPV1 was significantly overexpressed in **cisplatin-resistant tumor cells** compared to **cisplatin-susceptible tumor cells** (Fig. 4b and Supplementary Fig. 8a). To determine whether TRPV1 is functionally active in these cells, we examined TRPV1-induced electrical currents upon exposure to its agonist, capsaicin. TRPV1-specific inward currents were elicited by capsaicin and were abolished by the TRPV1-specific antagonist AMG9810 in CaSki CR cells, but not in CaSki P cells (Fig. 4c). Previous studies reported that TRPV1-mediated intracellular Ca²⁺ influx induced autophagy

by promoting autophagosome formation^{50, 51}. Interestingly, TRPV1 inhibition reduced autophagosome formation, thereby impeding autophagy-driven EGFR signaling (Fig. 4d) as well as increasing the sensitivity of CaSki CR cells to cisplatin (Fig. 4e). **Moreover, the above results were also reproduced in SiHa CR or HeLa cells (Supplementary Fig. 8b, c).** Thus, our results suggest that TRPV1, which is functionally active as a Ca²⁺-permeable cation channel, **could confer** resistance to cisplatin by regulating **autophagosome formation** in cisplatin-resistant tumor cells.

2) In figure 1, Gefitinib was used to show increased cell death in P3 cells. As a control, P0 cells needs to be shown.

As the reviewer suggested, we repeated the experiments assessing the effects of gefitinib or *siEGFR* on both CaSki P and CR cells. Inhibition or knockdown of EGFR in CaSki CR cells down-regulated MCL1 expression and re-sensitized to cisplatin (Fig. 1d-g of the revised manuscript). In contrast, Inhibition or knockdown of EGFR did not alter MCL1 expression and susceptibility to cisplatin of CaSki P cells (Fig. 1d-g of the revised manuscript).

We have included these data and description in the revised manuscript as followed:

[On Fig. 1c-g of the revised manuscript]

[On page 8, Results section of the revised manuscript]

To determine the roles of hyperactivated EGFR in the cisplatin-resistant phenotype of CaSki CR cells, we inhibited EGFR in CaSki CR cells using small-molecule inhibitor of EGFR, such as gefitinib (Fig. 1d). Compared to the control cells, gefitinib-treated CaSki CR cells were more susceptible to apoptosis induced by cisplatin (Fig. 1e). Consistently, the knockdown of *EGFR* using two kinds of small interfering RNAs (siRNAs) (*siEGFR* #1 or #2) reversed the cisplatin-resistant phenotypes of CaSki CR cells (Fig. 1f, g). **In contrast, EGFR inhibition did not alter susceptibility to cisplatin of CaSki P cells (Fig. 1d-g).**

3) In figure 2, the chemotherapy resistant signature is not convincing. The variability is very high and expression at the gene level often does not correlated with protein level. More evidence needs to be presented that cisplatin resistance patients have altered autophagy function.

We understand the reviewer's concern that the chemotherapy-resistant (CR) signature is not convincing evidence for discriminating cisplatin-resistant patients, and more evidence needs to be present that cisplatin-resistant patients have altered autophagy function. As the reviewer's comment, the expression at the gene level often does not correlated with protein level. To compensate this problem, many researchers have used gene signature which is consisted of multiple genes related to specific cellular phenotypes rather than used a single gene expression.

To verify that the CR signature could serve as an indicator for discriminating cisplatin-resistant cervical cancer patients, **we newly analyzed the transcriptome data from CESC classified as responders (R, complete and partial response) or non-responders (NR, stable and progressive disease) to cisplatin.** The CESC patient's information is retrieved from the Firehose BROAD GDAC data repository (<http://gdac.broadinstitute.org/>) and have been summarized in Supplementary Table 2 of the revised manuscript. **Importantly, we found that the CR signature was statistically increased in NR compared to R (Supplementary Fig. 1a of the revised manuscript).** Thus, these results indicate that **CR signature could predict responsiveness to cisplatin in cervical cancer patients and serve as an indicator for discriminating cisplatin-resistant patients.**

Nevertheless, **we have decided to remove the Fig. 2a of our original manuscript** because the autophagy signature we used in our original manuscript was not specific to the functionality of secretory autophagy but represented that of autophagy itself. Although we have searched many reports to find an accurate indicator or gene signature which can predict the functionality of secretory autophagy in cancer patients, unfortunately, we could not find it. Therefore, to avoid this confusion and clarify our results, we have removed the data on the signature of autophagy in the revised manuscript.

We appreciate the invaluable and helpful comment that improves the quality of our manuscript.

4) Fig 2 B shows changes in LC3B punctation was demonstrated using anti-LC3B antibodies. This was not very convincing as the punctation is very weak in P3. Transfection of GFP-LC3B would be a better method to validate whether autophagy is altered in P3 cells.

We thank the reviewer for this helpful comment. As the reviewer suggested, we seriously considered using GFP-LC3B. However, even GFP is frequently used as a reporter system, it has a risk of being easily quenched by the materials used in immunofluorescence analysis. For the reason, we have used PE-conjugated secondary antibody to assess LC3B punctation.

When PE-conjugated secondary antibody was used under the same experimental conditions, clear immunofluorescence images with more enriched autophagosome formation were obtained in CaSki CR cells compared to CaSki P cells.

Therefore, to clarify our results, we have replaced the old LC3B immunofluorescence images with the new one in the revised manuscript as followed:

[On Fig. 2a of the revised manuscript]

5) In figure 2 E, EGF was shown in the medium which was reduced by knocking out ATG7. There needs to be demonstrated that EGF release by cells lead to the hyperactivation of EGFR in P3 cells. This could be accomplished in several different way such as immunodepleting EGF in the media or knocking down EGF in the cells.

We agree with the reviewer's comment that there are needs to be demonstrated that EGF release by cells leads to the hyperactivation of EGFR in CaSki CR cells. To assess the reviewer's comment, we neutralized EGF by its specific monoclonal antibody in tumor cells. We found that neutralization of EGF led to a significant decrease in the phosphorylated levels of EGFR and resistance to apoptotic cell death by cisplatin (Supplementary Fig. 4c, d of the revised manuscript). Thus, our data suggest that a direct role for EGF in the hyperactivated EGFR signaling-mediated cisplatin resistance.

We have included these data and description in the revised manuscript as followed:

[On Supplementary Fig. 4c, d of the revised manuscript]

[On page 8, Supplementary Figure legend section of the revised manuscript]

c The levels of pEGFR, EGFR, and MCL1 protein were confirmed by western blots. β -actin was used as an internal loading control. Numbers below the blot images indicate the expression as measured by fold-change. **d** The frequency of apoptotic (active caspase 3⁺) cells were analyzed by flow cytometry. All experiments were performed in triplicate. The p-values by two-tailed Student's t-test (**b**) or two-way ANOVA (**d**) are indicated. The data represent the mean \pm SD. Source data are provided as a Source Data file. (*P \leq 0.05, **P \leq 0.01, ***P \leq 0.001. NS, not significant)

[On page 9, Results section of the revised manuscript]

Notably, the neutralization of secreted EGF by its specific monoclonal antibody led to a significant decrease in the phosphorylation levels of EGFR and the resistance to apoptotic cell death imposed by cisplatin in CaSki CR cells (Supplementary Fig. 4c, d), suggesting a direct role for EGF in the hyperactivated EGFR signaling-mediated cisplatin resistance.

6) Whenever demonstrating difference in LC3-II levels, autophagy flux needs to be determined. This requires the use of an autophagy inhibitor that prevents the formation of autolysosomes such as chloroquine.

We agree with the review's opinion. Many inhibitors, such as chloroquine (CQ) and bafilomycin A1 (BafA), have been used frequently to assess the autophagy flux in cells. We used BafA to verify the autophagy flux in this study because CQ could affect not only autophagy but also other cellular processes, such as proliferation and apoptosis. To verify the autophagy flux in cisplatin-susceptible and -resistant cells, we treated BafA, an inhibitor of V-ATPase which blocks the fusion of autophagosomes to lysosomes, to CaSki P and CR cells. While the protein abundance of LC3B-II was slightly increased

by BafA in CaSki CR cells compared with CaSki P cells, there was no significant alteration in autophagy flux in CaSki CR cells compared to CaSki P cells (Supplementary Fig. 3 of the revised manuscript). Thus, these results indicate that **cisplatin-resistant cervical cancer cells exhibit an increase in autophagosome abundance rather than autophagy flux.**

We have included these data and description in the revised manuscript as followed:

[On Supplementary Fig. 3 of the revised manuscript]

[On page 7, Supplementary Figure legend section of the revised manuscript]

Supplementary Fig. 3. Autophagy flux is not altered in cisplatin-resistant tumor cells.

CaSki P or CR cells were starved in medium supplemented with 0.1% FBS for 48 h and then incubated with or without BafA (10nM) for 12 h. The protein levels of LC3B were confirmed by western blots. β-actin was used as an internal loading control. Numbers below the blot images indicate the expression as measured by fold-change. The graphs represent quantification for absolute level of LC3B-II (left, *** $P \leq 0.001$, by two-way ANOVA) and autophagic flux (right, $p = N.S$ by 2-tailed Student's t-test), respectively. Autophagic flux was calculated by dividing the value of LC3B-II in the presence of BafA by that absence BafA. All experiments were performed in triplicate. The p-values by two-way ANOVA are indicated. The data represent the mean \pm SD. Source data are provided as a Source Data file. (* $P \leq 0.05$, ** $P \leq 0.01$, *** $P \leq 0.001$. NS, not significant)

[On page 9, Result section of the revised manuscript]

Interestingly, there was no significant alteration in autophagic flux in CaSki CR cells compared to CaSki P cells (Supplementary Fig. 3). Instead, the amount of LC3B-II escalated in parallel with increase in the amount of total cellular LC3B in CaSki CR cells (Fig. 2b). Therefore, our data indicate that cisplatin-resistant cervical cancer cells exhibit an increase in autophagosome abundance rather than autophagy flux.

7) In figure 3, the knockdown of NANOG reduces EGF in the media and increase cisplatin induced cell death in P3 cells. To demonstrate this is due to EGFR signaling, EGF should be added to the media and changes in cell death measured. In P0 cells overexpressing NANOG, EGF should be knockdown and effects on cisplatin induced cell death determined.

We agree with the reviewer's comment. As the reviewer suggested, we neutralized EGF with a specific monoclonal antibody in CaSki NANOG cells. The EGF neutralization decreased the NANOG-mediated cisplatin-resistant phenotype (Fig. 3g of the revised manuscript). Conversely, the treatment of recombinant EGF to NANOG-silenced CaSki CR cells increased resistance to apoptotic cell death by cisplatin (Fig. 3h of the revised manuscript). Thus, these results suggest that an immediate role for EGF in the NANOG-mediated cisplatin resistance.

We appreciate the insightful comment that has helped us improve the quality of our manuscript. We have included these data and description in the revised manuscript as followed:

[On Fig. 3g, h of the revised manuscript]

[On page 37, Figure legend section of the revised manuscript]

g and h. In the cells treated with or without anti-EGF or EGF, the frequency of apoptotic (active caspase 3⁺) cells after incubation with or without cisplatin for 24 h were confirmed by flow cytometry.

[On page 11, Results section of the revised manuscript]

We further questioned that the role of secreted EGF in NANOG-mediated cisplatin resistance. To verify the potential role of EGF in the NANOG-mediated cisplatin resistance in tumor cells, we neutralized EGF with a specific monoclonal antibody in CaSki NANOG cells. The EGF neutralization decreased the NANOG-mediated cisplatin-resistant phenotype (Fig. 3g). Conversely, the treatment of

recombinant EGF to NANOG-silenced CaSki CR cells increased resistance to apoptotic cell death by cisplatin (Fig. 3h), suggesting an immediate role for EGF in the NANOG-mediated cisplatin resistance. Taken together, our results demonstrate that NANOG is a critical mediator that could confer cisplatin resistance by regulating secretory autophagy-mediated EGFR activation.

8) It would be important to use AMG9810 in combination with EGFR inhibitors to demonstrate whether these treatments would be synergistic?

We thank the reviewer's this insightful comment. As the reviewer suggested, we performed the additional experiments to assess the effect of co-treatment of AMG9810 and gefitinib in cisplatin-resistant tumor cells. As shown below Fig. 2 for the reviewer only, co-treatment of AMG9810 and gefitinib could not induce synergistic effects. **This result is thought to be occurred as AMG9810 and gefitinib targets a crucial molecular pathway that confers resistance of tumor cell to cisplatin, such as EGFR signaling, in our CR models.**

Although experimental condition is different from the reviewer's question, we assessed a synergistic effect when AMG9810 in combination with cisplatin. To do this, we co-treated with AMG9810 with cisplatin at various dose. Compared to cisplatin alone treatment, AMG9810 treatment increased cisplatin-mediated cell death in dose-dependent manner (Fig. 8a of the revised manuscript).

To quantify the synergistic effects of treatment of AMG9810 with cisplatin, a combination score was calculated based on changes in the percentage of apoptotic cell death in cisplatin-treated tumor cells with or without AMG9810. Interestingly, we found that treatment with a sublethal dose of AMG9810 enhanced cisplatin-mediated cell death in a synergistic fashion (Fig. 8b of the revised manuscript). Thus, our data indicate that TRPV1 inhibition could be a plausible therapeutic strategy to overcome the cisplatin-resistant phenotypes of tumor cells.

[On Fig. 2 for the reviewer only]

[On Fig. 8a, b of the revised manuscript]

9) Besides P3 cells, it would be important to show that upregulation of NANOG in another cell lines sensitizes these tumors to combined cisplatin and AMG9810.

We appreciate the reviewer's this helpful suggestion. As the reviewer suggested, we performed additional experiments to verify that TRPV1 inhibition using AMG9810 could overcome resistance to cisplatin of other NANOG^{high} tumor cells. Similar to the results in the CaSki NANOG cells, **treatment of *siTRPV1* or AMG9810 reduced autophagosome abundance and EGFR signaling in the HEK293 NANOG cells compared to control** (Supplementary Fig. 10a, c of the revised manuscript). Notably, we found that **TRPV1 inhibition using *siRNA* or inhibitor reduced NANOG-mediated cisplatin resistant phenotypes** (Supplementary Fig. 10b, d of the revised manuscript).

In addition, **the above results were also reproduced in four-independent NANOG^{high} tumor cells, such as SiHa CR, HeLa, H1299 and MKN28 (Supplementary Fig. 8, 16 of the revised manuscript)**. These results indicate that TRPV1 plays a crucial role in NANOG-mediated resistance to cisplatin.

We have included these data and description in the revised manuscript as followed:

[On Supplementary Fig. 10 of the revised manuscript]

[On page 14, Supplementary Figure legend section of the revised manuscript]

Supplementary Fig. 10. TRPV1 is required for NANOG-mediated refractory phenotypes. **a** and **b** HEK293 NANOG cells were transfected with siRNA targeting *GFP* or *TRPV1*. **c** and **d** HEK293 NANOG cells were treated with DMSO or AMG9810. **a** and **c** The protein levels of TRPV1, LC3B, pEGFR, EGFR, pAKT, and AKT were analyzed by western blots. β -actin was used as an internal loading control. Numbers below the blot images indicate the expression as measured by fold-change. **b** and **d** Flow cytometry analysis of the frequency of apoptotic (active caspase 3⁺) cells after incubation with or without cisplatin for 24 h. All experiments were performed in triplicate. The p-values by two-way ANOVA are indicated. The data represent the mean \pm SD. Source data are provided as a Source Data file. (* $P \leq 0.05$, ** $P \leq 0.01$, *** $P \leq 0.001$. NS, not significant)

[On page 13, Results section of the revised manuscript]

We then questioned whether TRPV1 potentiates the refractory phenotypes of NANOG⁺ tumor cells. To directly link TRPV1 to these phenotypes, we silenced TRPV1 expression in CaSki NANOG and HEK293 NANOG cells (Fig. 5a and Supplementary Fig. 10a). Notably, *TRPV1* knockdown decreased autophagy-mediated activation of EGFR-AKT pathway (Fig. 5a and Supplementary Fig. 10a) and increased sensitivity to cisplatin in CaSki NANOG and HEK293 NANOG cells (Fig. 5b and Supplementary Fig. 10b). Consistent with the above results, we found that the NANOG-mediated phenotypes were dampened by TRPV1 inhibition using AMG9810 (Fig. 5c, d and Supplementary Fig. 10c, d), indicating TRPV1 plays a crucial role in NANOG-mediated resistance to cisplatin.

[On supplementary Fig. 8 of the revised manuscript]

[On page 12, Supplementary Figure legend section of the revised manuscript]

Supplementary Fig. 8. TRPV1 inhibition reduces cisplatin resistance by blocking autophagosome-mediated EGF secretion. **a** The protein levels of TRPV1 were determined by

western blot. **b** and **c** SiHa CR or HeLa cells were treated with DMSO or AMG9810. **b** The levels of internal LC3B, pEGFR, and EGFR and secreted EGF protein were confirmed by western blots. **c** Flow cytometry analysis of the frequency of apoptotic (active caspase 3⁺) cells after incubation with or without cisplatin for 24 h. **a** and **b** β -actin was used as an internal loading control. Numbers below the blot images indicate the expression as measured by fold-change. All experiments were performed in triplicate. The p-values by two-way ANOVA are indicated. The data represent the mean \pm SD. Source data are provided as a Source Data file. (* $P \leq 0.05$, ** $P \leq 0.01$, *** $P \leq 0.001$. NS, not significant)

[On page 12, Results section of the revised manuscript]

Interestingly, TRPV1 inhibition reduced autophagosome formation, thereby impeding autophagy-driven EGFR signaling (Fig. 4d) as well as increasing the sensitivity of CaSki CR cells to cisplatin (Fig. 4e). Moreover, the above results were also reproduced in SiHa CR or HeLa cells (Supplementary Fig. 8b, c). Thus, our results suggest that TRPV1, which is functionally active as a Ca²⁺-permeable cation channel, could confer resistance to cisplatin by regulating autophagosome formation in cisplatin-resistant tumor cells.

[On supplementary Fig. 16 of the revised manuscript]

10) It is unclear who TRPV1 leads to autophagy mediated secretion of EGF.

Previous studies reported that TRPV1 is not only involved in pain sensations but also other cellular process, such as autophagy (Chenfei Wang et al., Front. Pharmacol. 2022, Mingliang Chen et al., Signal transduction and targeted therapy. 2021). Interestingly, during the last years, it has become evident that the role of autophagy is not restricted to degradation alone but also mediated unconventional forms of secretion (Tom G. Keulers et al., Front. Oncol. 2016). Indeed, recent studies have demonstrated that autophagy induces cancer therapeutic resistance by promoting secretion of intracellular molecules, such as growth factors and chemokines (Suyeon Kim et al., Autophagy. 2020,

Silvina Odete Bustos et al., *Cancers*. 2022). Therefore, we hypothesized that TRPV1 might induce EGF secretion by regulating secretory autophagy.

In this study, we found that EGF secretion was increased in cisplatin-resistant tumor cells compared with cisplatin-susceptible cells (Supplementary Fig. 4a, b, and 6a of the revised manuscript), and that elevated EGF secretion was regulated by secretory autophagy (Fig. 2d, e, g, and Supplementary Fig. 5, 6a of the revised manuscript).

Furthermore, we found that TRPV1 inhibition reduced autophagosome abundance and EGF secretion in cisplatin-resistant tumor cells (Fig. 4d, and Supplementary Fig. 8b of the revised manuscript). Conversely, TRPV1 overexpression in cisplatin-susceptible cells induced autophagosome abundance and EGF secretion (Fig. 7a, c of the revised manuscript).

Therefore, we believe that **these results support our hypothesis that the TRPV1 promotes EGF secretion by regulating secretory autophagy.**

[On supplementary Fig. 4a, b, and 6a of the revised manuscript]

Supplementary Fig. 4

Supplementary Fig. 6

[On Fig. 2d, e, g and Supplementary Fig. 5 of the revised manuscript]

Fig. 2d

Fig. 2e

Fig. 2g

Supplementary Fig. 5

[On Fig. 4d and Supplementary Fig. 8b of the revised manuscript]

Fig. 4d

Supplementary Fig. 8b

[On Fig. 7a, c of the revised manuscript]

11) It is unclear how this will be different than blocking autophagy mediated cell survival function by providing cells with nutrients such as amino acids.

The ability of cells to respond to changes in nutrient availability is essential for the maintenance of metabolic homeostasis and viability. One of the key cellular responses to nutrient withdrawal is the upregulation of autophagy. Recently, there has been a rapid expansion in knowledge of the molecular mechanisms involved in the regulation of autophagy induction in response to depletion of key nutrients, such as amino acids, ATP, and oxygen levels (Ryan C Russell et al., Cell Research. 2013). Among key nutrients to regulate autophagy, amino acids stimulate insulin-mechanistic target of rapamycin (mTOR)-mediated signal transduction which controls the major metabolic pathways. The mTOR activation by amino acids suppresses autophagosome formation and/or maturation by phosphorylating ULK1 and ATG13, which inhibits the ULK1 complex (Alfred J. Meijer et al., Amino Acids. 2015).

As the reviewer commented, theoretically, **the nutrients provision to tumors might be a strategy to reduce the autophagy-mediated therapeutic refractoriness**. However, **the nutrients provision is not suitable as a clinically applicable strategy** because it could promote oncogenic phenotypes, such as hyperplasia by the activation of mTOR pathway.

In contrast, TRPV1 inhibition could block autophagy-mediated therapeutic refractoriness without promoting oncogenic phenotypes. In this study, we found that TRPV1 promoted autophagy-mediated EGFR hyperactivation and, consequentially, the acquisition of cisplatin resistance. Notably, TRPV1 inhibition rendered tumors susceptible to cisplatin treatment and led to the successful control of the disease.

Thus, we believe that **our findings in this study could provide the evidence to support that TRPV1 inhibition could be a clinically plausible strategy to not only block autophagy-mediated therapeutic refractoriness but also alleviate, at the same time, chemotherapy-induced neuropathic pain.**

We appreciate the review's invaluable comment.

Reviewers' Comments:

Reviewer #1:

Remarks to the Author:

The authors addressed the points that I raised. The manuscript is significantly improved. Nicola Normanno

Reviewer #2:

Remarks to the Author:

The manuscript presents abundant data, mostly Western blots, to suggest an elaborate and innovative lineal pathway, NANOG - TRPV1 - Ca²⁺ - EGF secretion - pEGFR - AKT in cisplatin resistance.

The authors made good efforts to improve the resubmission. The revision adds many additional data of similar quality to address the previous reviews. This reviewer doesn't have specific criticism but think that such average data and unusual/innovative conclusions are difficult to be believed by critical readers.

Overall, it is conceptually difficult to accept the components in the lineal pathway that seem to be very random. If more concrete results to support each of the several claims, these will be merit of forming several solid publications.

Reviewer #4:

Remarks to the Author:

The authors have sufficiently answered all my concerns about the experimental design. There needs to be further discussion on several important issue raised by the manuscript. The authors need to recognize that EGFR is only one of four EGFR family members and even though EGF binds to EGFR, it affects activation of ErbB2, 3 and 4. This needs to be discussed. Another important issue this manuscript describes is the autophagy secretory pathway versus its degradatory pathway. As the authors know, autophagy flux will lead to survival under stress condition and when over activated or dysregulated will lead to cell death. This needs to be discussed how these different functions of autophagy can exist in cancer cells.

Otherwise this manuscript is very important to expand our understanding of autophagy functions.

Reviewer #5:

Remarks to the Author:

The authors have generally addressed the previous comments. However, there are still a few issues that need to be addressed.

1. For comments #2 (How are low and high chemotherapy-resistant tumors defined? What is the cut-off mark? The gene signatures were generated from the breast cancer model. Are the same gene signatures apply to cervical?), the answer from the authors is insufficient, and this one needs further clarification. In the revised article, the author used 70th and 30th for CR gene signature high and low, respectively. However, it is unclear to this reviewer that how this cut-offs were chosen. If the authors use a more objective cut-offs (such as 50th, 50th), would the results remain the same? Also, on a separate but related issue, Figure 1B shows there are 12 patients in the NR group and 55 patients in the R group. However, a quick look into TCGA CESC dataset by this reviewer, it seems there are 15 patients in NR group and 58 patients in R group. The patients summary provided by the author (Table S2) confirms this. It seems some patients were excluded for analysis. What is the rationale for that? The authors must provide reasonable rationale if certain patients were excluded. Otherwise, this will diminish the confidence in this analysis which is supposed to be unbiased.

2. For minor comments #2 (Please indicate in the Methods section how the cisplatin was dissolved. Depending on the solvent, the efficacy of cisplatin changes and will be beneficial to the readers.), the authors stated that cisplatin was dissolved in DMSO. This is an incorrect way to prepare cisplatin and would significantly weaken some of the conclusions. It is well known that DMSO inactivates platinum compounds such as cisplatin and carboplatin (Cancer Res. 2014 Jul 15;74(14):3913-22). Using DMSO to prepare cisplatin would significantly reduce the efficacy of cisplatin and may also have unwanted other effects. Just an example, in Figure 5B, 10 μ M of cisplatin which is quite high,, only induced 4%-8% of apoptotic cells, which is quite low, suggesting the cisplatin used in the experiment may not be fully effective due to inactivation by DMSO. The correct way to dissolve cisplatin is using PBS or water.

Dear Reviewer,

We appreciate the detailed and invaluable comments that have been raised in response to our manuscript (NCOMMS-21-51679B) entitled “**TRPV1 inhibition overcomes cisplatin resistance by blocking autophagy-mediated hyperactivation of EGFR signaling pathway**”. We appreciate the detailed and relevant comments from the reviewers. Your assistance has been invaluable and has helped us improve the quality of our manuscript. We have fully addressed each reviewer’s questions and comments and amended the manuscript content accordingly. We have also provided point-by-point responses to the critiques raised by the reviewers. All the reviewer questions are in **bold and italic text** while our responses are in regular or bold text (if necessary). In the revised manuscript, the changes are marked in **RED**. Once again, we appreciate you for your time, consideration, and invaluable guidance.

Sincerely,

Tae Woo Kim, PhD
Professor, Department of Biomedical Science
Korea University College of Medicine
Phone: +82-2-2286-1301
Fax: +82-2-923-0480
E-mail: twkim0421@korea.ac.kr

Responses to reviewers:

Reviewer #1

The authors addressed the points that I raised. The manuscript is significantly improved.

We appreciate the reviewer's positive conclusion.

Reviewer #2

The manuscript presents abundant data, mostly Western blots, to suggest an elaborate and innovative lineal pathway, NANOG - TRPV1 - Ca²⁺ - EGF secretion – pEGFR - AKT in cisplatin resistance. The authors made good efforts to improve the resubmission. The revision adds many additional data of similar quality to address the previous reviews. This reviewer doesn't have specific criticism but think that such average data and unusual/innovative conclusions of are difficult to be believed by critical readers.

Overall, it is conceptually difficult to accept the components in the lineal pathway that seem to be very random. If more concrete results to support each of the several claims, these will be merit of forming several solid publications.

We understand the reviewer's concern regarding the lineal NANOG-TRPV1-pEGFR axis in cisplatin resistance. In this study, we demonstrated that the cause of TRPV1 overexpression in cisplatin-resistant cancer was due to direct transcriptional regulation by NANOG, but other possibilities may exist. Studies have reported that NANOG also works together with other stemness factors, such as KLF4, MYC, OCT4, or SOX2, to control target genes that have important functions in embryonic stem cells and, plausibly, in tumor cells (Ben-Porath I et al., Nat Genet. 2008, Oh et al., J Clin Invest. 2022). From this perspective, it is possible that TRPV1 expression by other stemness factors also contribute to tumor progression and therapeutic resistance. Additionally, TRPV1 was found to be aberrantly expressed by various factors in tumors, such as inflammation, tissue damage, and hypoxic conditions (Li L et al., Int J Bio Sci. 2021, Zhai K et al., Int J Mol Sci. 2020), suggesting that TRPV1 expression in tumor cells could be mediated by various tumor microenvironmental factors as well as stemness factors besides NANOG.

Furthermore, we found that TRPV1 expression by itself was sufficient to induce cisplatin resistance in its channel activity-dependent manner. Notably, after *EGFR* knockdown, the cisplatin-resistant phenotypes of *TRPV1*-transduced P0 cell was almost entirely lost, highlighting that the EGFR

pathway can function as a primary route through which TRPV1 promotes these phenotypes. Although it will be important in future studies to assess the precise underlying mechanisms by which TRPV1 regulates autophagy-dependent EGFR activation, this connection immediately hints at several potentially promising therapeutic targets to control cisplatin-resistant cancer in the clinic.

We appreciate the invaluable comment that has helped us improve the quality of our manuscript. We have revised the discussion section in the revised manuscript as follows:

[On page 17, 18, Discussion section of the revised manuscript]

The regulatory mechanism of TRPV1, particularly in the course of acquiring cisplatin resistance, has not yet been extensively studied. *In this study, we noted that the cause of TRPV1 overexpression in cisplatin-resistant cancer was due to direct transcriptional regulation by NANOG, but other possibilities may exist. NANOG also works together with other stemness factors, such as KLF4, MYC, OCT4, or SOX2, to control target genes that have important functions in embryonic stem cells and, plausibly, in tumor cells^{27, 53}. From this perspective, it is possible that TRPV1 expression by other stemness factors also contribute to tumor progression and therapeutic resistance. Additionally, TRPV1 was found to be aberrantly expressed by various factors in tumors, such as inflammation, tissue damage, and hypoxic conditions³³, suggesting that TRPV1 expression in tumor cells could be mediated by various tumor microenvironmental factors as well as several different stemness factors besides NANOG. Furthermore, we found that TRPV1 expression by itself was sufficient to induce cisplatin resistance in its channel activity-dependent manner. Notably, after EGFR knockdown, the cisplatin-resistant phenotypes of TRPV1-transduced P0 cell was almost entirely lost, highlighting that the EGFR pathway can function as a primary route through which TRPV1 promotes these phenotypes. Although it will be important in future studies to assess the precise underlying mechanisms by which TRPV1 regulates autophagy-dependent EGFR activation, this connection immediately hints at several potentially promising therapeutic targets to control cisplatin-resistant cancer in the clinic.*

Reviewer #4

The authors have sufficiently answered all my concerns about the experimental design. There needs to be further discussion on several important issues raised by the manuscript.

1) The authors need to recognize that EGFR is only one of four EGFR family members and even though EGF binds to EGFR, it affects activation of ErbB2, 3 and 4. This needs to be discussed.

The reviewer raises an important issue regarding that TRPV1-mediated EGF secretion may induce the resistance to cisplatin treatment through activation of other ErbB family members, such as ErbB2, ErbB3, and ErbB4. As the reviewer mentioned, when EGFR binds to its ligand, such as EGF, it undergoes a conformational change that allows it to dimerize with other EGFR molecules or with other ErbB family members. For example, the dimerization of EGFR with ErbB2 leads to the activation of both receptors, which results in initiation of intracellular signaling pathways that regulate cellular processes, such as proliferation, differentiation, and survival (Freed DM et al., Cell. 2017, Che JC et al., Physiol Rev. 2016). Interestingly, Shiqi Ma et al. reported that EGFR/ErbB2 heterodimer can induce the resistance to tyrosine kinase inhibitors by preventing apoptosis (Shiqi Ma et al., Oncogene. 2021).

We believe that there is a possibility that TRPV1-mediated EGF secretion may induce cisplatin resistance through activation of other ErbB family members, and further exploration should be progressed to define the possibility in future studies.

Thus, we have included additional description in the revised manuscript as followed:

[On page 18, Discussion section of the revised manuscript]

With regard to the cisplatin resistance induced by autophagy-dependent EGFR activation, this raises a question: Is EGFR alone capable of promoting these phenotypes? When EGFR binds to its ligand, such as EGF, it undergoes a conformational change that allows it to dimerize with other EGFR molecules or with other ErbB family members, such as ErbB2, ErbB3, or ErbB4. For example, the dimerization of EGFR with ErbB2 leads to the activation of both receptors, which results in initiation of intracellular signaling pathways that regulate cellular processes, such as proliferation, differentiation, and survival^{54, 55}. Interestingly, Shiqi Ma et al. reported that EGFR/ErbB2 heterodimer can induce the resistance to tyrosine kinase inhibitors by preventing apoptosis⁵⁶. Therefore, we believe that there is a possibility that TRPV1-mediated EGF secretion may induce cisplatin resistance through activation of other ErbB family members, and further exploration should be progressed to define the possibility in future studies.

2) Another important issue this manuscript describes is the autophagy secretory pathway versus its degradatory pathway. As the authors know, autophagy flux will lead to survival under stress condition and when over activated or dysregulated will lead to cell death. This needs to be discussed how these different functions of autophagy can exist in cancer cells.

Otherwise this manuscript is very important to expand our understanding of autophagy functions.

We appreciate the reviewer's insightful comment. Studies have demonstrated that autophagy

plays a dual role in cancer by suppressing the growth of tumors or the progression of cancer development, which seems to be dependent on unknown characteristics of various cancer types (Dong Wook Shin. *Biomol Ther.* 2020). The exact mechanisms underlying the paradoxical functions of autophagy in cancer cells are complex and not fully understood. Accumulating evidence has implicated that a combination of genetic and environmental factors determines the outcomes of autophagy activation in cancer cells (Su Min Lim et al., *Cell & Bioscience.* 2021). In this study, we unexpectedly found that cisplatin-resistant cancer cells exhibited an increase in autophagosome abundance which can accelerate autophagy flux. Interestingly, however, it also increased the secretion of EGF, which activated the EGFR-AKT signaling and, consequentially, the acquisition of cisplatin resistance. Notably, we found that the NANOG-TRPV1 axis was one of major molecular pathway inducing autophagosome abundance in cisplatin-resistant cancer cells. Although how paradoxical functions of autophagy can coexist in those cancer cells remain as a critical question to be answered in future studies, at least our results propose that the NANOG-TRPV1 axis could be used as one of molecular markers to predict the outcome of autophagosome abundance in cisplatin-resistant cancer cells.

We have included additional description in the revised manuscript as followed:

[On page 19, Discussion section of the revised manuscript]

Studies have demonstrated that autophagy plays a dual role in cancer by suppressing the growth of tumors or the progression of cancer development, which seems to be dependent on unknown characteristics of various cancer types⁶⁰. The exact mechanisms underlying the paradoxical functions of autophagy in cancer cells are complex and not fully understood. Accumulating evidence has implicated that a combination of genetic and environmental factors determines the outcomes of autophagy activation in cancer cells⁶¹. Here, we unexpectedly found that cisplatin-resistant cancer cells exhibited an increase in autophagosome abundance which can accelerate autophagy flux. Interestingly, however, it also increased the secretion of EGF, which activated the EGFR-AKT signaling and, consequentially, the acquisition of cisplatin resistance. Notably, we found that the NANOG-TRPV1 axis could be a major molecular pathway inducing autophagosome abundance in cisplatin-resistant cancer cells. Although how paradoxical functions of autophagy can coexist in those cancer cells remain as a critical question to be answered in future studies, at least our results propose that the NANOG-TRPV1 axis could be used as one of molecular markers to predict the outcome of autophagosome abundance in cisplatin-resistant cancer cells.

Reviewer #5 - Replacement for Reviewer #3 - (Remarks to the Author):

The authors have generally addressed the previous comments. However, there are still a few issues that need to be addressed.

1) For comments #2 (How are low and high chemotherapy-resistant tumors defined? What is the cut-off mark? The gene signatures were generated from the breast cancer model. Are the same gene signatures apply to cervical?), the answer from the authors is insufficient, and this one needs further clarification. In the revised article, the author used 70th and 30th for CR gene signature high and low, respectively. However, it is unclear to this reviewer that how this cut-off was chosen. If the authors use a more objective cut-offs (such as 50th, 50th), would the results remain the same? Also, on a separate but related issue, Figure 1B shows there are 12 patients in the NR group and 55 patients in the R group. However, a quick look into TCGA CESC dataset by this reviewer, it seems there are 15 patients in NR group and 58 patients in R group. The patient summary provided by the author (Table S2) confirms this. It seems some patients were excluded for analysis. What is the rationale for that? The authors must provide reasonable rationale if certain patients were excluded. Otherwise, this will diminish the confidence in this analysis which is supposed to be unbiased.

We thank the reviewer's insightful comments. As suggested, we newly analyzed the level of EGFR activity scores from CESC classified as having low levels (low, n = 150) and high levels (high, n =150) of CR gene signature (Cutoffs values for the CR gene signature: 50th, 50th). As with previous results (Cutoffs values for the CR gene signature: 30th, 70th) (Supplementary Fig. 1c of the revised manuscript), we found that EGFR activity scores were statistically higher in CR high than in CR low patients (Figure for the reviewer only). Although the difference in EGFR activity score between CR low (< median) and CR high (> median) patients is statistically significant, there is an overlap in the values of the two cohorts. To show an obvious difference in EGFR activity score between the two cohorts, we used 70th and 30th for CR gene signature high and low, respectively.

In addition, the reviewer raises a concern that it seems some patients were excluded for analysis. The NR group is consisted of 12 patients treated with cisplatin alone (cisplatin) and 3 patients treated with cisplatin plus other drugs (cisplatin/other). As above, the R group is also consisted of 55 patients treated with cisplatin alone and 3 patients treated with cisplatin plus other drugs. To rule out the effects of other drugs, we analyzed patients who was treated only cisplatin in each group. To avoid the confusion and help reader's understanding, we have revised the Result section in the revised manuscript as followed:

[Figure for the reviewer only]

[On page 6, Results section of the revised manuscript]

We further questioned whether hyperactivated EGFR signaling could confer resistance to cisplatin-mediated chemotherapy. By analyzing the EGFR activity scores in transcriptomic data on patients with TCGA cervical cancer patients classified as responders (R, complete or partial response) or non-responders (NR, stable or progressive disease) to cisplatin **alone** treatment, we found that EGFR activity scores were statistically higher in NR than in R (Fig. 1b).

2) For minor comments #2 (Please indicate in the Methods section how the cisplatin was dissolves. Depending on the solvent, the efficacy of cisplatin changes and will be beneficial to the readers.), the authors stated that cisplatin was dissolved in DMSO. This is an incorrect way to prepare cisplatin and would significantly weaken some of the conclusions. It is well known that DMSO inactivates platinum compounds such as cisplatin and carboplatin (Cancer Res. 2014 Jul 15;74(14):3913-22). Using DMSO to prepare cisplatin would significantly reduce the efficacy of cisplatin and may also have unwanted other effects. Just an example, in Figure 5B, 10uM of cisplatin which is quite high, only induced 4%-8% of apoptotic cells, which is quite low, suggesting the cisplatin used in the experiment may not be fully effective due to inactivation by DMSO. The correct way to dissolve cisplatin is using PBS or water.

We appreciate the reviewer for pointing out our mistake. We found that the solvent to dissolve cisplatin was erroneously stated in our manuscript as DMSO. We previously used DMF (dimethylformamide) rather than DMSO as the solvent for cisplatin in our experiments. Although DMF is an organic solvent like DMSO, it is a solvent that can maintain cisplatin activity for a long time unlike DMSO (Yong Weon Yi et al., DNA Repair (Amst). 2011).

Additionally, the reviewer raises an important point that the percentage of apoptotic cells was

small even though we used quite high dose of cisplatin. In cisplatin-mediated cell cytotoxicity experiments, we have treated 10uM cisplatin for 24 hours. Under these conditions, we found that 15-20% apoptotic cell death was induced in cisplatin-sensitive CaSki (CaSki P) cells, whereas 3-5% apoptotic cell death was induced in cisplatin-resistant CaSki (CaSki CR) cells or CaSki NANOG cells. When the time of cisplatin treatment was increased, apoptotic cell death in those cell lines dramatically increased, but the fold change in apoptotic cell death between those cell lines was decreased. For this reason, to show a clear difference in cisplatin-mediated cell death between the experimental groups, we treated cisplatin for 24 hours. We believe that the reason why the apoptotic cell death was low at high concentrations of cisplatin is due to the cisplatin treatment time, not cisplatin activity.

To correct our mistake, we have revised the material and method section in the revised manuscript as followed:

[On page 20, Material and Methods section of the revised manuscript]

The following chemical reagents were used in this study: AMG9810 (Sigma-Aldrich, St. Louis, MO, USA, A2731), capsaicin (Sigma-Aldrich, M2028), gefitinib (Selleckchem, Houston, TX, USA, S1025), cisplatin (Selleckchem, S1166), spautin-1 (Selleckchem, S78880), and bafilomycin A1 (Sigma-Aldrich, B1793-2UG). **The chemical reagents we used were dissolved in below solvents. AMG9810, gefitinib, spautin-1, or bafilomycin A1: Dimethyl sulfoxide (DMSO); Cisplatin: Dimethyl formamide (DMF); Capsaicin: ethanol.**

[On page 25, Material and Methods section of the revised manuscript]

NOD/SCID mice were inoculated subcutaneously with 1×10^6 CaSki CR cells per mouse. Nine days following the tumor challenge, AMG9810 (0.05 mg/kg)-loaded chitosan hydrogel or cisplatin (2.5 mg/kg) was administered intratumorally or intraperitoneally, respectively. The cisplatin is prepared in **DMF**, diluted in phosphate buffered saline (PBS).